

# The use of O₂ 1.27 μm absorption band revisited for GHG monitoring from space and application to MicroCarb

Jean-Loup Bertaux[1,4], Alain Hauchecorne[1], Franck Lefèvre[1,6], François-Marie Breon[5], Laurent Blanot[3], Denis Jouglet[2], Pierre Lafrique[2], Pavel Akaev[4],

[1]LATMOS/IPSL, UVSQ Université Paris-Saclay, Sorbonne Université, CNRS, 11 Boulevard d'Alembert, 78280 Guyancourt, France
[2]Centre National d'Etudes Spatiales, CST, 18 av. Edouard Belin, 31401 Toulouse, France
[3]ACRI-ST, 11 Boulevard d'Alembert, 78280 Guyancourt, France
[4]Laboratory for planetary and exoplanetary atmospheres, IKI/RAN, Moscow, Russia
[5]LSCE, CEA/CNRS/ Université de Versailles Saint-Quentin, 91191 Gif-sur-Yvette, France
[6]LATMOS/IPSL, Sorbonne Université, UVSQ, CNRS, Paris, France

*Correspondence to*: Jean-Loup Bertaux (jean-loup.bertaux@latmos.ipsl.fr)

**Abstract.**

Monitoring $CO_2$ from space is essential to characterize the spatio/temporal distribution of this major greenhouse gas, and quantify its sources and sinks. The mixing ratio of $CO_2$ to dry air can be derived from the $CO_2/O_2$ column ratio. The $O_2$ column is usually derived form its absorption signature on the solar reflected spectra over the $O_2$ A-band (i.e. OCO-2, Tanso/Gosat, Tansat). As a result of atmospheric scattering, the atmospheric path length varies with the aerosols load, their vertical distribution, and their optical properties. The spectral distance between the $O_2$ A-band (0.76 μm) and the $CO_2$ absorption band (1.6 μm) results in significant uncertainties due to the varying spectral properties of the aerosols over the globe.

There is another $O_2$ absorption band at 1.27 μm with weaker lines than in the A-band. As the wavelength is much nearer to the $CO_2$ and $CH_4$ bands, there is less uncertainty when using it as a proxy of the atmospheric path length to the $CO_2$ and $CH_4$ bands. This $O_2$ band is used by the TCCON network implemented for the validation of space-based GHG (Green House Gases) observations. However, this absorption band is contaminated by the spontaneous emission of the excited molecule $O_2^*$, which is produced by the photo-dissociation of $O_3$ molecules in the stratosphere and mesosphere. From a satellite looking nadir, this emission has a similar shape as the absorption signal that is used.

In the frame of the CNES MicroCarb project, scientific studies have been performed in 2016-2018 to explore the problems associated to this $O_2^*$ airglow contamination and methods to correct it. A theoretical synthetic spectrum of the emission was derived from a new approach, based on $A_{21}$ Einstein coefficients information contained in the line-by-line HITRAN 2016 data base. The shape of our synthetic spectrum is fully validated when compared to $O_2^*$ airglow spectra observed by SCIAMACHY/ENVISAT in limb viewing.

We have designed an inversion scheme of SCIAMACHY limb viewing spectra, allowing to determine the vertical distribution of the Volume Emision Rate of the $O_2^*$ airglow. The VER profiles and corresponding integrated nadir intensities were both compared to a model of the emission based on the chemical-transport model REPROBUS. The airglow intensities depend mostly on the Solar Zenith Angle (both in model and data) and the model underestimate the observed emission by ~15%. This is fully confirmed with SCIAMACHY nadir viewing measurements over the oceans: in such conditions, we have disentangled and retrieved the nadir $O_2^*$ emission in spite of the moderate spectral resolving power (~860).



It is shown that with the MicroCarb spectral resolution power (25,000) and SNR, the contribution of the $O_2^*$ emission at 1.27 µm to the observed spectral radiance in nadir viewing may be disentangled from the lower atmosphere/ground absorption signature with a great accuracy. Simulations with 4ARCTIC radiative transfer inversion tool have shown that the $CO_2$ mixing ratio may be retrieved with the accuracy required for quantifying

the $CO_2$ natural sources and sinks (pressure level error $\leq$ 1 hPa, $X_{CO2}$ accuracy better than 0.4 ppmv) with only the $O_2$ 1.27 µm band. As a result of these studies (at an intermediate phase), it was decided to include this band (B4) in the MicroCarb design, while keeping the $O_2$ A band for reference (B1). Our approach is very similar (likely identical), to the approach of Sun et al. (2018) who also analysed the potential of the $O_2$ 1.27 µm band and concluded favourably for GHG monitoring from space. We advocate for the inclusion of this $O_2$ band on

other GHG monitoring future space missions, such as GOSAT-3 and EU/ESA $CO_2$-M missions, for a better GHG retrieval.

## 1. Introduction

Carbon Dioxyde ($CO_2$) is recognized as the main driver of climate change. Its evolution in time is therefore scrutinized with attention. The atmospheric fraction is the ratio of the atmospheric increase to the anthropogenic

emission. On decadal time scales, this ratio has been close to 0.5 since the beginning of continuous measurements of atmospheric concentration in the late 50s, despite an increase of the anthropogenic emissions by a factor of 5 (Le Quéré et al., 2018). An atmospheric fraction lower than 1 is explained by the existence of natural sinks that are fuelled by the increasing amount of $CO_2$ in the atmosphere. The current global carbon budget indicates that the ocean and land surface contribute roughly equally to the sink. There is little doubt that

the oceanic sink will continue in the future despite a solubility decrease induced by raising temperature, while the fate of the land sink is more uncertain (Ciais et al., 2014). There is a lack of understanding of the vegetation dynamic, and its response to increasing $CO_2$ and changing climate. In fact, there is no consensus whether the land sink is mostly in the tropics, mid-latitudes or boreal regions. This lack of understanding of the vegetation processes limits our ability to anticipate the carbon budget and thus the rate of climate change.

There is therefore a strong need for a better understanding of the carbon cycle and the processes that control the exchanges of carbon between the atmosphere, the vegetation and the soil. This understanding can be obtained through a continuous monitoring of the $CO_2$ fluxes at the land-atmosphere interface and the analysis of its response to inter-annual climate anomalies. This objective suggests the development of a satellite monitoring system as recognized by the scientific community and several space agencies (CEOS, 2018).

The first satellites to be launched with the aim of monitoring the $CO_2$ cycle were GOSAT (JAXA) and OCO (NASA). The latter was unfortunately lost at launched, and a very similar satellite, OCO-2, was build and launched. These have been followed by TANSAT (China). All instruments rely on a similar method to estimate the $CO_2$ concentration from space: High resolution spectra of the reflected sunlight are acquired over several bands centred on clusters of $CO_2$ and $O_2$ absorption lines. The depths of the lines are sensitive to the number of

molecules along the sunlight atmospheric path. The so-called differential absorption method makes it possible to infer the amount of absorbing gas along the line of sight, using some ancillary information on the temperature profile. $CO_2$ is the target component of the atmosphere and $O_2$ is used as a normalization component to link the $CO_2$ estimated number of molecule to a mixing ratio. Note that the sunlight atmospheric path length is linked to the surface pressure, but also to the presence of light-scattering particles (aerosols and clouds) in the atmosphere.



Because oxygen is well mixed in the atmosphere, it is adequate for the normalization of the measurement to estimate a mixing ratio.

The instruments currently in orbit focus on the $CO_2$ absorption bands at 1.6 and 2.0 µm, and the $O_2$ absorption band at 0.76 µm. The use of the oxygen band poses several challenges: (i) there is still significant uncertainty on

the radiative transfer modelling within this band; and (ii) its central wavelength is notably different from that of the $CO_2$ bands so that the spectral variations of the atmospheric scatterer optical properties may lead to different optical paths.

An alternative could be the use of the $O_2$ absorption band around 1.27 µm. It is much closer in wavelength to the $CO_2$ absorption bands which reduces the uncertainties linked to the spectral variations of the atmospheric

path. In addition, the absorption lines are weaker than those in the 0.76 µm band so that the radiative transfer modelling may be more accurate. In fact, the 1.27 µm band is the one used for the processing of TCCON spectra for the estimate of column mixing ratio. This band was not selected for current flying $CO_2$ monitoring mission because it is affected by airglow, a light emitted by oxygen molecules in the high atmosphere. Airglow has a spectrum that is very similar to the oxygen absorption spectrum used to estimate the sunlight atmospheric

path.

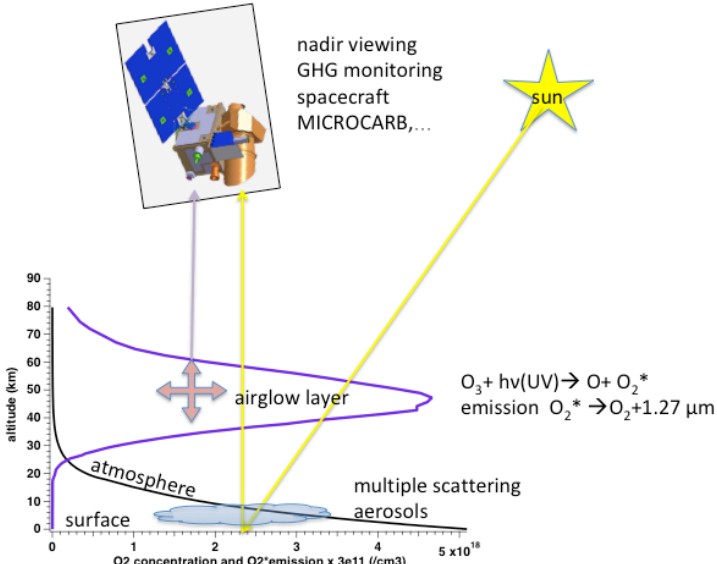

**Figure 1. Sketch of a space instrument and platform to monitor GHG gases, including $CO_2$. The $O_2$ concentration (black curve) and the $O_2^*$ Volume Emission Rate of the 1.27 µm airglow (blue curve) are plotted as a function of altitude. The optical path of nadir viewing observations are crossing inevitably the airglow layer, which emission is**

**superimposed on the spectrum of solar radiation scattered by the surface + aerosols + atmosphere system. The $O_2$ absorption at 1.27 µm is mainly produced in the lower atmosphere, while the airglow is in the range ~30-70 km of altitude.**

Previous studies (Kuang et al., 2002) conducted during the preparation phase of the OCO mission (Crisp et al.

2004) indicated that the contribution of airglow could not be corrected with the desired accuracy. Conversely, similar studies performed during the design phase of the CNES-MicroCarb mission indicated that the airglow



could be distinguished from the oxygen absorption spectrum, provided that the instrument achieve a high spectral resolution. These unpublished studies led to the addition of a fourth band, centred at 1.27 µm, in the MicroCarb optical concept. The MicroCarb mission shall then be the first $CO_2$ monitoring mission to test the potential of the 1.27 µm band, rather than the 0.76 µm band, for the estimate of $CO_2$ column concentrations from

5       space. Note that the instrument does carry the classical $O_2$ band at 0.76 µm both for safety and for comparison purposes. Recently, an independent study (Sun et al., 2018) confirms the MicroCarb analysis. This paper shows that, indeed, airglow has a spectral signature that is different from that of the oxygen absorption and can therefore be distinguished from the signature of oxygen absorption. It argues for the inclusion of the 1.27 µm band in the design of future $CO_2$ monitoring mission.

In this paper, we describe the analysis of the airglow signature that has been conducted in the context of the MicroCarb preparation.

When describing the choices made to define the OCO investigation to determine from nadir viewing observations the $CO_2$ vertical columns and mixing ratios with associated $O_2$ columns, Kuang et al. (2002) recognized the virtues of the $O_2$ band at 1.27 µm (nearest to the $CO_2$ bands), but discarded its use because it is

contaminated by the intense $O_2$ airglow day side emission when looking nadir from an orbiter (Figure 1). They quoted Noxon (1982) as having shown that the emission is not only intense, but variable. In fact, Noxon (1982) analysed spectra of this emission collected from 60 flights of a KC-135 aircraft over 10 years and a variety (latitude and seasons) of observing conditions, including two solar eclipses. He reported that there were no secular variation (within 30%), and also that the variations with latitude (obtained along a single flight) were

very smooth. This smoothness is comforted by the present study of both the SCIAMACHY data set and the airglow model that we made, plugged to a CTM model of ozone (not a climatology).

We also note that the TCCON ground-based spectrometer array, observing the sun, uses this 1.27 µm band to derive the $CO_2$/dry air mixing ratio (because the $O_2$/dry air mixing ratio is fixed = 0.2095) rather than the A band, which can also be measured by some TCCON spectrometers. The argument is that the depth of the $O_2$

lines at 1.27 µm has the same order of magnitude as those of $CO_2$, while the A band (760 nm) absorption lines are much stronger. The use of spectral bands with similar absorption depth may reduce small systematic errors (e. g. detector linearity failure) for atmospheric quantities that are based on measurement ratios. We argue that the same argument can be used for observations from space, although other problems are added (Ring effect of filling the line bottoms, polarization...).

Given the level of accuracy which is needed for a useful retrieval of $CO_2$, it may not be possible to rely only on an *a priori* model of the $O_2$ airglow to subtract it from a nadir-viewing spectrum which contains both the absorption spectrum of $O_2$ and the emission spectrum at 1.27 µm, intimately intricated. An exception may be for high surface reflectance scenes, such as glint viewing, when the transmitted-reflected signal gets much larger than the airglow emission. Indeed, for typical scenes, the amplitude of the reflected and airglow spectra are

similar, with nearly identical spectral variations. There are nonetheless some differences that make it possible to disentangle one from the other. First, there is the Collision Induced Absorption (CIA) which is present in absorption, but not in emission, since it is proportional to the square of the $O_2$ density, and therefore confined to lowest altitudes. Second, the transmittance Tr=exp(-τ) saturates at high optical thicknesses τ>1,while the emission does not .Third, individual rotational lines are subjected to pressure broadening, also proportional to

the square of the air density. Therefore, the emission lines occurring at high altitudes are much thinner than the



same absorption lines built in the lower atmosphere. These effects are illustrated on Fig. 2. O'Brien and Rayner (2002) have proposed to discriminate the emission from the absorption by recording one single line at very high spectral resolution (resolving power of 400,000), with an imager and three very narrow filters, which positions are indicated on Fig. 2. One difficulty with this scheme is that the photon flux collected in those three narrow bands is very small and corresponding SNR strongly reduced, rending unpractical this proposal. By contrast, the size of a pixel element of MicroCarb (corresponding to a resolving power 25,000) is also indicated for comparison. The whole spectrum is recorded, and the shoulders of the absorption line contributes to the disentangling of emission and absorption in a retrieval exercise, with 1024 pixels distributed along the $O_2$ band.

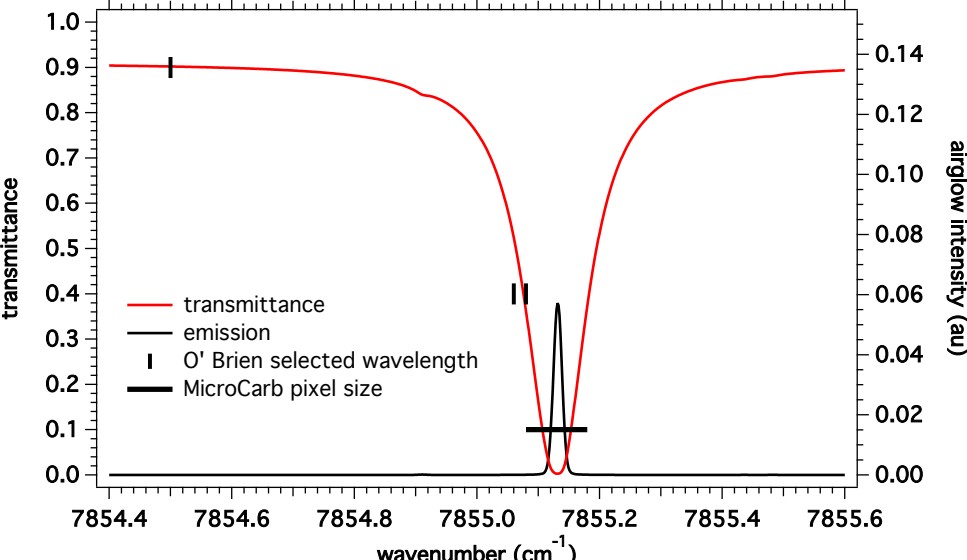

**Figure 2. The transmittance within an individual $O_2$ line (red) is much larger than its counterpart in emission (black line). The channels recommended by O'Brien and Rayner (2002), of width 0.02 cm-1, are represented: one outside an $O_2$ line for the continuum, the other two on the side of an absorption line, but still outside the airglow emission line. The transmittance was calculated with HITRAN at nadir at highest spectral resolution. The black line represents the MicroCarb pixel size giving a resolving power 25,000.**

This paper is organized as follows. In Section 2 a brief review of previous observations of the $O_2$ (0,0) airglow emission at 1.27 µm is presented first. Then the details of the spectral structure of this band are described, together with a method to compute accurately the shape of the airglow spectrum. In Section 3 we describe how the SCIAMACHY observations of this airglow at the limb have been processed in order to derive VER (Volume Emission Rate) vertical profiles, and how a synthetic spectrum may be derived from combining the VER profile and our spectroscopic studies. Our model spectral shapes are validated by a comparison with SCIAMACHY observed shapes. In Section 4 we compared the airglow total nadir intensities and VER profiles derived from SCIAMACHY limb observations with our REPROBUS airglow model (described in details in Appendix A). A deficit of airglow from the model is found. Another comparison of the ozone predicted by REPROBUS with GOMOS/ENVISAT observations indicates a deficit in ozone which, when accounted for, would narrow the discrepancy between SCIAMACHY and the airglow model. The Microcarb space mission with its instrument is



briefly described in Section 5, while in Section 6 are detailed the accuracy and bias results of the $O_2$ column retrievals in some typical situations. It is shown that the $O_2$ airglow emission may be extracted form nadir viewing SCIAMACHY observations over the oceans, where the reflectance is minimal, in spite of its moderate spectral resolution. In section 7 are examined briefly some other cases where absorption measurements could be contaminated by airglow emission. Finally some conclusions are drawn, with a prospective on future GHG monitoring space missions.

## 2. Observations and spectroscopy of $O_2$ airglow (0,0) band at 1.27 μm

### 2.1 Observations of the airglow of the $O_2$* emission at 1.27 μm

The aeronomical emission at 1.27 μm was first observed in 1956, in the "dayglow" (daytime aeronomical emissions) from instruments on board soviet stratospheric balloons (up to 30 km altitude) (Gopshtein and Kushpil, 1964), but its origin was not understood at that time. Noxon and Vallance Jones (1962) recorded spectra from a KC-135 plane flying at 13 km of altitude and described the origin of the emission in the form of the electronic transition of the oxygen molecule from an excited state to the fundamental, with the emission of a photon in one of the rotating branches of the (0,0) transition that form the entire "1.27 μm atmospheric IR band":

$$O_2\left(a^1\Delta_g\right) \longrightarrow O_2\left(X^3\Sigma_g^-\right) + h\nu(1.27\ \mu m) \tag{1}$$

The recorded intensity was very large (more than 10 MegaRayleigh), but atmospheric absorption in the very same band absorbs most of it before reaching the ground. The much fainter emission from the (0,1) transition of the same electronic state at 1.58 μm had been observed earlier from the ground, because it is not attenuated by $O_2$ absorption (most $O_2$ molecules are in the V''= 0 vibration level at atmospheric temperatures).

There are various ways to produce an $O_2$ molecule in its excited state $a^1\Delta_g$, which are detailed in Appendix A. Once it is produced, it remains there with a long lifetime, about 75 minutes, and is spontaneously de-excited by emitting a photon, or also by a collision without a photon ("quenching"). The most important mechanism of production of these excited molecules is the photolysis of ozone by solar UV:

$$O_3 + h\nu(\lambda \le 310\ nm) \longrightarrow O\left(\ ^1D\right) + O_2\left(a^1\Delta_g\right) \tag{2}$$

which therefore occurs during the day, but can be observed more easily from the ground at dusk with a high intensity of 30 MegaRayleigh.

At night, the emission falls to 100 KiloRayleigh, but this time the origin of the molecules $a^1\Delta_g$ is mainly due to the recombination of oxygen atoms:

$$O + O + M \longrightarrow O_2\left(a^1\Delta_g\right) + M \tag{3}$$

### 2.1.1. Observations of the 1.27 μm from the ground



One difficulty for ground-based observations of the 1.27 µm emission is that most of the emission is absorbed by $O_2$ before reaching the ground, letting 4-10 % coming down to the ground. The second difficulty is that there is a strong solar light scattered component by the atmosphere/dust. It is (now) clear what should be the optimal observing conditions:

- look near the zenith to minimize $O_2$ absorption in the lower atmosphere. From high altitude the $O_2$ absorption will be a little bit reduced.

- look just after the sunset, when the sun is still illuminating the ozone layer producing this emission, between 30 and 80-90 km, but not illuminating the lower atmosphere, to avoid Rayleigh/Mie scattering which produces a strong sky background signal.

The first observation form the ground was reported by Lowe et al. (1969), followed by Baker et al. (1975) and Pendleton et al. (1996), all fulfilling the above-mentioned optimal conditions.

### 2.1.2. Space Observations of the 1.27 µm

With a sounding rocket Evans et al. (1968) were able to reconstruct for the first time the vertical distribution of the emission at 1.27 µm, by inverting the brightness integral. Their VER (Volume Emission Rate) profile showed that emissivity is highest at about 50 km (~ $10^7$ photons cm$^{-3}$ s$^{-1}$), and zero or low below 30 km (due to quenching and screening of solar UV by ozone). A secondary maximum at about 85 km is due to the presence of a layer of mesospheric ozone well documented by GOMOS/ENVISAT in star occultation mode of observation

(Kyrölä et al., 2018).

From space, the following observations at the limb should be noted:

- the SME satellite (Solar Mesospheric Explorer), Thomas et al. (1984)

- the IR SABER photometer on board the NASA TIMED aeronomy mission, (Mlynczak et al., 2007).

- the OSIRIS spectrometer (Optical Spectrograph and InfraRed Imager System) on the ODIN satellite (Llewellyn

et al. 2004).

- the SCIAMACHY experiment on board ENVISAT (2002-2012), which we analyse in Section 3 and 4.

The objective of observations is generally to derive the concentration of O and $O_3$ in the upper atmosphere.

### 2.1.3. Observations on Venus and Mars

The emission of Venus at 1.27 µm was discovered by Pierre Connes (Connes et al., 1979), then studied in particular by Dave Crisp et al. (1996). It is due to the recombination reaction (3) O+O+M. On Mars, both types of emissions (2) and (3) exist. The ozone emission was discovered by Pierre Connes (Noxon et al., 1976), and recombination emission (3) by the OMEGA experiment aboard Mars Express (Bertaux et al., 2012).

### 2.2. Spectroscopy: The various electronic states of the $O_2$ molecule

On Fig. 3 (from Khomich et al., 2008) are represented the various electronic states of the di-oxygen molecule $O_2$, and the names of the transitions between them. The $O_2$ molecule, being composed of two identical atoms, is said to be *homopolar*. The fundamental state $X\,^3\Sigma_g^-$ is put at a reference energy level=0. The number 3 indicates a

triplet state, which may be decomposed in 3 sub-states with very nearby energies. The descending arrows on Fig.



indicate transitions to a lower level, corresponding to the emission of one photon. The reverse process corresponds to the absorption of one photon. We describe briefly below three of the transitions indicated on Fig. 3 which are the most relevant to GHG gases, since $O_2$ is used as a reference to determine the mixing ratio of $CO_2$. We can make use of the energy/wavelength conversion:

$$E(ev) = h\nu = \frac{hC}{\lambda} = \frac{1242.26}{\lambda}\,(nm) \qquad (4)$$

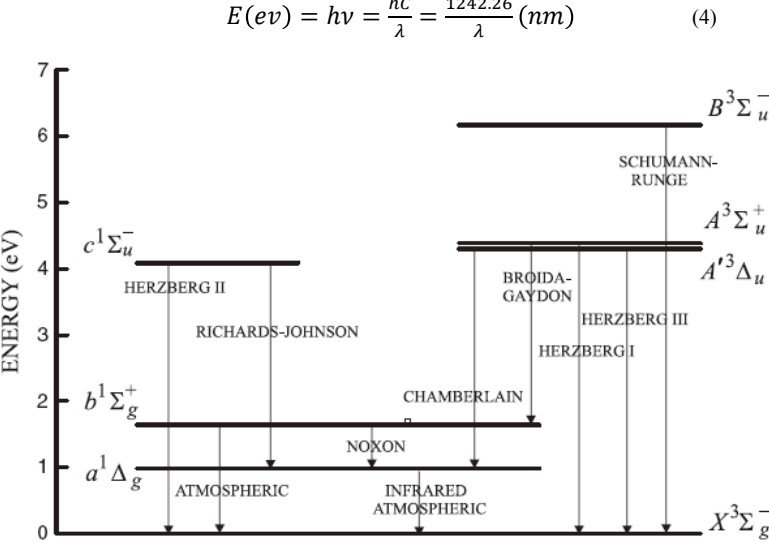

**Figure 3.** **Energy levels of the various electronic states of the di-oxygen molecule $O_2$, and name of electronic transitions between them (reprinted by permission from Springer Nature Customer Service Centre GmbH: Springer**

**Verlag, Processes Responsible for the Occurrence of the Airglow by Vladislav Yu. Khomich, Anatoly I. Semenov, Nicolay N. Shefov, copyright 2008). The present study is relevant to the so-called « infrared atmospheric » transition.**

1.The « Schumann-Runge » band is in the far UV. The UV solar flux dissociates also $O_2$ molecules, producing O

atoms, which may recombine with $O_2$ to form $O_3$, which absorbs in the nearer UV. All together, $O_2$ and $O_3$ are protecting life (and DNA molecules) from harmful solar UV.

2. The « atmospheric » band is the transition ($b\ ^1\Sigma_g^+ \rightarrow X\ ^3\Sigma_g^-$), around 760 nm also called A-band from Fraunhofer early nomenclature, or "atmospheric band", heavily used in GHG studies.

3. The « infra-red atmospheric » band is the transition ($a\ ^1\Delta_g \rightarrow X\ ^3\Sigma_g^-$), around 1270 nm or 1.27 µm in the near

infra-red, sometimes called $O_2$ IR band, or ($^1$Delta band) (according to Gordon et al., 2010). This band is the subject of the present study.

Because the $O_2$ molecule is homopolar, it has no electric dipolar moment and in principle electronic transitions are forbidden. The electronic transitions can only happen thanks to the existence of a magnetic dipole moment (M1) and/or a quadrupolar electric moment (E2). As a consequence, spontaneous transitions from a given

electronic state down to the fundamental state $X\ ^3\Sigma_g^-$ are unlikely, therefore the lifetime of such a state is rather long: 13 s for the atmospheric A-band at 760 nm, transition ($b\ ^1\Sigma_g^+ \rightarrow X\ ^3\Sigma_g^-$) (Mlynczak and Solomon, 1993) and 75 minutes for the $O_2$ IR band at 1.27 µm, transition ($a\ ^1\Delta_g \rightarrow X\ ^3\Sigma_g^-$) (Lafferty et al. 1998). Therefore, the $O_2$



molecule excited at level $a\ ^1\Delta_g$ (often represented by $O_2^*$ or $^1\Delta\ O_2$ in the following) which results from ozone photo-dissociation has plenty of time to reach thermal equilibrium with ambient gas, and the various states of vibration-rotation will be populated according to a Boltzmann law (therefore, depending on the local temperature T) modulated by the rotation quantum number J, with a statistical weight 2J+1 as described later below.

For the $O_2$ IR band at 1.27 μm, the transition ($a\ ^1\Delta_g \rightarrow X\ ^3\Sigma_g^-$) is mainly due to a magnetic dipole (M1). There is, however, also a system of absorption lines due to the electric quadrupole (E2), identified for the first time both in atmospheric absorption spectra (looking at the sun) and in laboratory experiment (Gordon et al., 2010). The overall absolute strength of this (E2) system is about 215 times weaker than the (M1) system (Gordon et al., 2010).

**2.3. Computation of a synthetic spectrum of $O_2^*$ emission from HITRAN 2016**

The aim of this section is to describe a way to compute what could be the emission spectrum of the emission of $O_2^*$ in the dayglow, relevant to nadir GHG daylight observations. In an early phase of the present studies in

2016, we made the following crude approximation. We computed, from HITRAN 2016 (Gordon et al., 2017) the local high resolution spectrum of absorption by $O_2$ molecule at a variety of altitudes. Then we assumed that the emission spectrum *shape* (but not magnitude) was identical to the absorption spectrum. In the following, we detail how we can compute more accurately a theoretical spectrum of the local emission of $O_2^*$ molecule, from the data that are present in HITRAN 2016 line-by-line informations and some additional considerations.

**2.3.1 Computation of the total emission rate of $O_2^*$ molecule coming from ozone dissociation**

As will be described with more details later on, the Reprobus CTM model is used to compute the 3D distribution of ozone as a function of space and time, based on a set of chemical reactions, solar effect, and the actual

meteorological fields of winds, temperature and pressure procured by ECMWF.

Then an additional model computes the photo-dissociation of ozone with the UV solar spectrum of the day, yielding to a 3D distribution of the concentration $[O_2^*]$ of species $O_2^*$ (electronic state $^1\Delta_g$), in units of $cm^{-3}$. As said above, the lifetime of this electronic state is about 75 minutes, corresponding to a spontaneous emission rate $A_{global}$ of $1/75*60 = 2.22 \times 10^{-4}\ s^{-1}$. For low altitudes, one must take into account the quenching of this excited state

by collisions with all other gases (including $O_2$), mainly at the lowest altitudes (<50 km). Ignoring the quenching, the rate of emitted photon, the Volume Emission Rate VER in units of photons/ ($cm^3$ s) may be computed form:

$$VER = -\frac{d[O_2^*]}{dt} = A_{global}[O_2^*] \qquad (5)$$

**2.3.2 Computation of the airglow detailed line-by-line intensity**

The same principle may be applied to the detailed emission spectrum, by computing a VER for each transition line $L_i$ of the 1.27 μm band, line-by-line. This can be done from the data contained in the HITRAN data base for this $O_2$ electronic transition. All the existing lines (within a certain wavelength interval) in absorption do exist

also in emission, and only those do exist. In equation (5), $A_{global}$ must be replaced by the Einstein coefficient of



spontaneous emission $A(L_i)$ which is present in the HITRAN table for each transition (web site HITRANonline, https://hitran.org). In the present study we have used only the $^{16}O^{16}O$ main isotope of $O_2$. For higher degrees of refinement that may be necessary in view of the high accuracy required on $X_{CO2}$, other $O_2$ isotopes should be considered.

The concentration $[O_2*(E_i)]$ of the $O_2*$ molecule must be computed in all possible energy levels $E_i$. $E_i$ depends on the rotational number J' (J' for upper state, J'' for lower state which here is irrelevant), and the vibration number (mainly V'=0, but some with V'=1). The concentration $[O_2*(E_i)]$ will depend on the rotational statistical weight 2J'+1 and the temperature T, and is also proportional to $[O_2*]$.

$$VER(L_i) = A(L_i)\left[O_2^*(E_{i)}\right] \qquad (6)$$

**2.3.3 Computing the distribution of $O_2*$ molecules among the various energy levels.**

In their 2006 paper, Simeckova et al. (2006) describe « the calculation of the statistical weights and the Einstein
A -coefficients for the 39 molecules and their associated isotopologues/isotopomers currently present in the line-by-line portion of the HITRAN database ». This is all that is needed to calculate second members of equation (6).
In an approximation of a two level system (upper m and lower n levels are denoted as 2 and 1 respectively), we have the well-known equations linking the Einstein A- and B-coefficients

$$g_1 B_{12} = g_2 B_{21} \qquad (7)$$
$$A_{21} = 8\pi h \nu^3 B_{21} \qquad (8)$$

where $A_{21}$(spontaneous emission) is in $s^{-1}$, and $B_{12}$ (absorption) and $B_{21}$ (stimulated emission) are in $cm^3$ $(J\ s^2)^{-1}$,
and $g_1$ and $g_2$ are the statistical weights of the levels 1 and 2, respectively.
We start from equation (17) of Simeckova et al. (2006) with molecules in the upper level, to describe their relative distribution according to their energy level and temperature. If N is the total number of molecules per unit volume at the temperature T, the population $N_2$ of the energy level $E_2$ is equal to:

$$N_2 = \frac{g_2\ N}{Q_{tot}(T)}\ e^{-c_2 E_2/T} \qquad (9)$$

where $Q_{tot}(T)$ is the total internal partition sum of the absorbing gas at the temperature T, and $E_2$ is the energy of the upper state in units of wavenumber $(cm^{-1})$ . $c_2$ is the second radiation constant, $c_2=hc/k_B$, where c is the speed of light, $h$ is the Planck constant, and $k_B$ =1.38065 x$10^{-16}$ erg $K^{-1}$ is the Boltzmann constant.
Since the sum of all values of $N_{2i}$ is equal to the total number of molecules N, an expression of $Q_{tot}(T)$ may be derived (where index i refers to each value of $E_2$):

$$Q_{tot}(T) = \sum_i g_i\ e^{-c_2 E_{2i}/T} \qquad (10)$$



For the lower level, we may find the value of $Q_{tot}(T)$ in Table 1 of the paper of Simeckova et al. (2006). For instance, $Q_{tot}(T=296 \text{ K})=215.77$ for the main oxygen isotopologue $^{16}O^{16}O$. The temperature 296 K is a reference temperature for the HITRAN data base. For our purpose, we have to find $Q_{tot}(T)$ for the upper level of the transition, from a summation like in equation (6). The summation must be not on all the transitions, but on all energy levels, because from a given energy level having a certain population $N_{2i}$, there are several transitions going down to the lower level with different $A_{21}$. Once we have $Q_{tot}(T)$, the total internal partition sum, then we may compute all values of $N_{2i}$, for the required temperature, from the distribution of the excited molecules between the various energy levels from equation (9).

The energy levels should be measured for convenience from the lowest energy level of the upper state $E_0$, $E=E_i-E_0$. We found for the upper level $Q_{tot}(T=296 \text{ K})=147.196$ for 296 K, and $Q_{tot}(T=200 \text{ K})= 100.143$ for the upper level. On Fig. 4 are represented both the exponential term E and the statistical weight, product of the exponential term by $2J'+1$. Only the V'=0 are kept here, because levels V'=1 are weakly populated, though they are present in the line list that are extracted from HITRAN line-by-line data base in our selected wavelength interval of interest (transition (1,1).

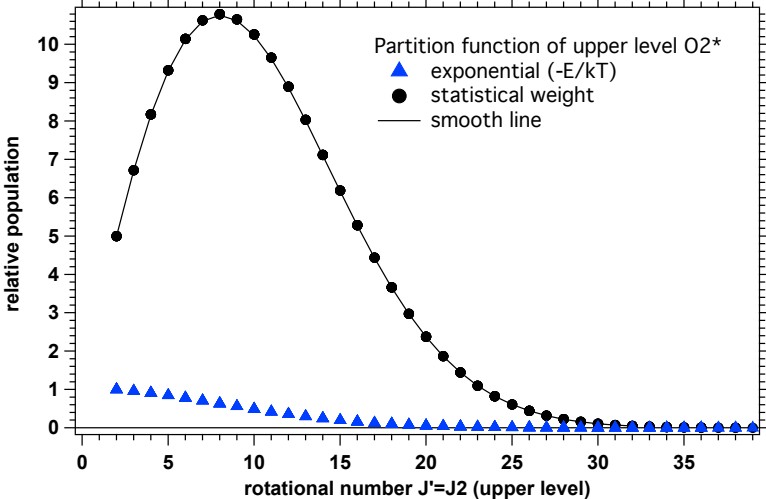

**Figure 4. Partition function of the $O_2^*$ molecule as a function of rotation state J'. Blue triangles: exponential term for population, =1 for the lowest; Black circles: the population computed for all HITRAN transitions with V'=0, V''=0. There are 5, 7, or 8 values (and transitions) for each black circle on the figure. The V''=1, V'=1 transitions (1,1) are not shown. The first point for J'=2 is obviously 5=2J'+1, and exp(0)=1.**

On Fig. 5 are represented the various energy levels (in cm$^{-1}$) of the upper state, the $O_2^*$ molecule. The energy is counted above the lowest energy level of the fundamental state $X\,^3\Sigma_g^-$.



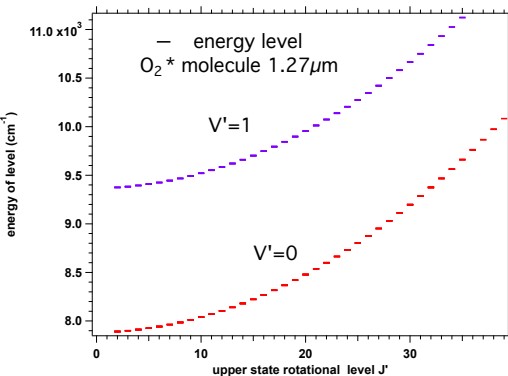

**Figure 5. Energy levels (in cm⁻¹) of the excited molecule $O_2^*$ as a function of rotational and vibrational quantum numbers J' and V'.**

5   **2.3.4. Computing the total decay rate of the excited molecule $O_2^*$**

The actual distribution $Q(J',T)/Q_{tot}(T)$ of the excited molecule $O_2^*$ is a function of the rotational sate J' and of the temperature, such that the sum on all J' is equal to one:

$$\frac{\sum Q(J',T)}{Q_{tot}(T)} = 1 \qquad (11)$$

Each particular rotational level has its own decay rate, or Einstein probability of spontaneous emission, called $A_{21}$, which is given in HITRAN tables of line-by-line lists. The total (average) decay rate is obtained by summing all $A_{21}$ on all transitions weighted by the relative population:

$$A_{21tot} = \sum A_{21}(J') \frac{Q(J',T)}{Q_{tot}(T)} \qquad (12)$$

Since there is the emission of one photon around 1.27 μm for each decay of one excited $O_2^*$ molecule, $A_{21tot}$ is the sum of all rates of all transitions going down, which is the total emission rate, the total number of photons emitted in the whole band per second by one single molecule of $O_2^*$.

20   Then we must multiply by the number density of $O_2^*$ to get the volume emission rate in photons per (cm³ s).

We found that the total decay rate is $A_{21tot}=2.29\ 10^{-4}\ s^{-1}$. We may compute the lifetime $1/A_{21tot}=4365\ s\sim 73$ mn, slightly different from the 75 mn quoted by Lafferty et al. (1998). The excited molecule $O_2^*$ will, in average, stay excited for more than one hour (in the absence of quenching, de-excitation by collisions without the emission of one photon, not addressed here).

25   It must be realized that it is experimentally very difficult to measure directly such a long lifetime. Instead, because the values of $A_{21}$ are connected to the values of absorption coefficients $B_{21}$, it is easier to measure the absorption of $O_2$ molecules, and then make the appropriate calculations to derive the $A_{21}$ values, according to principles explained in Simeckova et al. (2006), which have been used to fill the HITRAN line-by-line lists with $A_{21}$ rates for each transition.





**2.3.5. Computing the emission spectrum of the excited molecule $O_2*$**

The emission rate per $O_2*$ molecule $\varepsilon(k)$ of a transition k is obtained by multiplying the Einstein Coefficient $A_{21}(k)$ by the relative population of the upper level:

$$\varepsilon(k) = A_{21}(k) \frac{Q(J'(k),T)}{Q_{tot}(T)} \qquad (13)$$

When plotted as a function of wavelength of transition k it represents the emission spectrum for one molecule, per molecule and per second. It is displayed on Fig. 6 for 296 K (right scale) and compared to the absorption line
intensity SS which are found in the line-by-line HITRAN line list (left scale). The distribution of the lines in three branches is clearly observed (Q branch: J'-J''= 0; P branch: J'-J''= -1; R branch: J'-J''= +1). The transitions from V'=1 to V''=1 are very weak and near the zero line.

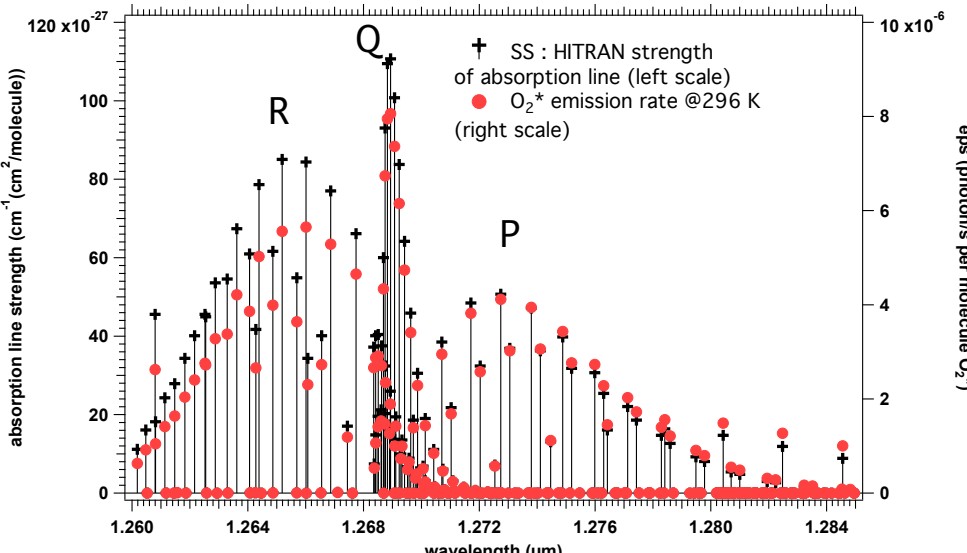

**Figure 6: Emission rate per excited molecule $O_2*$ (eps=$\varepsilon$(k), in photons/s per molecule $O_2*$, left scale) compared to the**
**absorption line strengths SS at 296 K (in units of cm$^{-1}$(cm$^2$/ molecule). See text for more explanations.**

Therefore, a spectrum of the local emission in the band could be computed, by describing each emission transition by a gaussian with an appropriate width (associated to the temperature), adding all transitions to form a full spectrum, and multiplying by the actual density of $O_2*$ molecules.
However, we have implemented another method, to take advantage of LBLRTM software (Line By Line Radiative Transfer Model, Clough and Iacono, 1995) which computes for the terrestrial atmosphere absorption spectra (either local, or integrated over one LOS, line-of-sight) from HITRAN data base. Indeed, with the adequate scaling of both right and left scale of Fig. 6, it is noted that the strength of absorption lines are just above the emission rates on the left side of the graph (short wavelength), while it is the reverse on the right side.





As we shall see below, there is a theoretical reason for this progressive change of the ratio of emission to absorption strength.

### 2.3.6. Theoretical computation of the ratio emission/absorption

By reporting the expression of $A_{21}$ from equation (20) from Simeckova et al. (2006) (which links $A_{21}$ to the HITRAN line strength SS) into the expression of emission in eq. (13), we could find a very simple result on the ratio of emission $\varepsilon$ to absorption line strength SS for each line:

$$\frac{\varepsilon}{SS} = K \; \frac{v_0^2}{\exp\left(\frac{c_2 v_0}{T}\right)-1} \qquad (14)$$

with the constant K being:

$$K = 8\pi c \left(\frac{Q_{tot}^{lo}}{Q_{tot}^{up}}\right) \exp\left(\frac{c_2 E\_0}{T_0}\right) \qquad (15)$$

15 in which $E\_0 = 7892.02$ cm$^{-1}$ is the energy of the lowest level J'=2, V'=0 for the $O_2$* excited molecule as found in HITRAN data. On Fig. 7 are plotted both the ratio of our calculated emission to the HITRAN line strength SS for all transitions within our spectral interval, (black dots), and the analytical formula (13) applied to all transitions of HITRAN Table. Both share the same scale on the left: there is a perfect coincidence, which validates our derivation of equation (14).

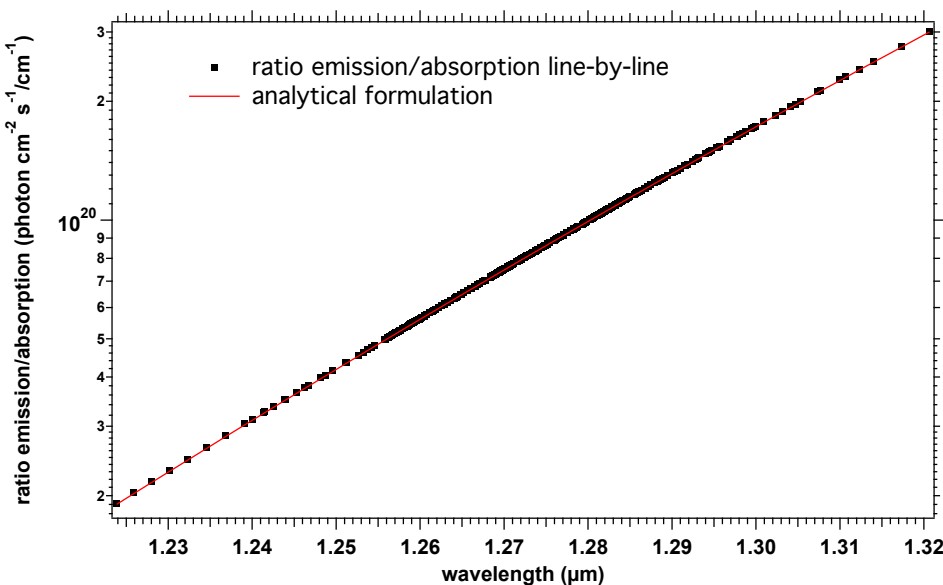

**Figure 7: Both the ratio of our calculated emission rate to the HITRAN line strength SS (black dots), and the analytical formula (14) applied to all 375 transitions of HITRAN Table (between 1.2238 and 1.32068 µm) are plotted using the same scale on the left, in units of photon cm$^{-2}$ s$^{-1}$ /cm$^{-1}$.**





**2.3.7. Practical method to produce a synthetic emission spectrum**

In order to simulate a local emission spectrum, we could use the method exposed in 2.3.5, giving the emissivity per molecule $O_2^*$ for each line, and then distribute this emissivity over a gaussian attached to each line, and add

spectrally all line contributions. We have developed another method, capitalizing on the capabilities of LBLRTM software (Clough and Iacono 1995). With LBLRTM, we may compute a local absorption spectrum of $O_2$ (for instance, computing the vertical atmospheric transmission between 67 and 68 km of altitude, $Tr(\lambda)$). This transmission is linked to the local absorption $a(\lambda)$ by :

$$Tr(\lambda) = \exp(-a(\lambda)dz) \qquad (16)$$

or

$$a(\lambda) = -\frac{\ln(Tr(\lambda))}{dz} \qquad (17)$$

The *shape* of the local emission spectrum $E_m(\lambda)$ of $O_2^*$ molecules is then obtained by multiplying $a(\lambda)$ by the

expression (14) of the ratio emission/absorption $\varepsilon/SS$, which depends only of the wave number $\nu$ and temperature T, according to a continuous function. The wave number $\nu$ in cm$^{-1}$ is equal to $10000/\lambda$ (in μm). Then the obtained local emission spectrum $E_m(\lambda)$ has to be normalized (by integration over wavelength) to the actual VER for which we wish to compute the local emission rate spectrum, yielding $E_{mn}(\lambda)$.

The advantage of this approach is that LBLRTM computes the local absorption spectrum with a number of

effects (line broadening, pressure shifts, gaussian profiles,…) that may become non-negligible at low altitudes. Applying equation (14) on the continuous spectrum, instead of applying it to the discrete set of wavelengths of each transition is justified by the fact that $\nu$ and expression (14) varies very little over the spectral extent of one individual line.

In order to compute the brightness spectrum of the $O_2^*$ emission along any LOS, one has to integrate $E_{mn}(\lambda)$

over length through the atmosphere. The "self-absorption" by $O_2$ molecules along the LOS must be accounted for, except in regions where it is negligible (high altitudes, say >80 km).

**3. The use of SCIAMACHY data for the study of the $O_2$ ($^1\Delta$) emission**

Several space instruments have been used in the past to the study of the $O_2$ ($^1\Delta$) emission, mainly to retrieve the $O_3$ concentration: the Solar Mesosphere Explorer satellite (SME, Thomas et al., 1984); one infra-red radiometer aboard the satellite OHZORA, (Yamamoto et al., 1988) ; one infra-red imager a part of Osiris instrument on board ODIN (Llewellyn et al., 2004) ; the SABER broad band photometer on board TIMED NASA mission (Gao et al., 2011) and SCIAMACHY spectrometer on board ESA ENVISAT mission. We have used the

SCIAMACHY data because of the spectral capability (resolution $\lambda/d\lambda\sim850$) and extensive produced data set during the ESA/ENVISAT mission.

**3.1 Description of SCIAMACHY investigation of $O_2$ ($^1\Delta$) emission**





SCIAMACHY is a multi-channel spectrometer dedicated to the study of Earth's atmosphere on board the European Space Agency Envisat satellite. The name is the acronym of SCanning Imaging Absorption SpectroMeter for Atmospheric CHartographY (Burrows et al., 1995, Bovensmann et al., 1999). It is an eight-channel grating spectrometer that measures scattered sunlight in limb and nadir geometries from 240 to 2,380 nm. In addition it was operated also in solar and lunar occultation. In this study, we have used both limb and nadir measurements covering the $O_2$ $(^1\Delta)$ band (1,230–1,320 nm) in the spectral channel 6 (1,050–1,700 nm).

In a recent study to retrieve the volume emission rates of $O_2$ $(^1\Delta)$ and $O_2$ $(^1\Sigma)$ in the mesosphere and lower thermosphere, Zarboo et al. (2018) have used a special mode of SCIAMACHY: the MLT limb scan mode, dedicated to the study of the mesosphere and lower thermosphere in the region 50-150 km. This mode was used only twice a month from July 2008 until April 2012. In contrast, we have used the normal limb mode viewing geometry, where SCIAMACHY tangentially observes the atmosphere from the surface up to about 100 km with a vertical step of 3.3 km. At each tangent point, the vertical resolution is 2.6 km, the horizontal along-track resolution is about 400 km, and the horizontal cross-track resolution is 240 km. To improve the signal-to-noise ratio, the four cross-track spectra at the same elevation step are co-added, reducing cross-track resolution to 960 km (the swath width).

To generate data for our study, we used the SCIAMACHY dataset level 1b version 8.02 that we converted into level 1c radiometrically calibrated radiances (in physical unit) by using the SCIAMACHY command line tool SciaL1c version 3.2. Before deriving the $O_2(^1\Delta)$ VER profiles, we had to perform a few corrections on the level 1c radiance spectra. First we subtracted the average of the 4 spectra measured above 105 km tangent height (generally around 150 km or 250 km) as a dark spectrum from the measured spectra at all of the other tangent heights. This high altitude spectrum contains some residual spectral (readout) patterns left from the calibration step. All spectra contain two bad pixels at wavelength 1262.267 nm and 1282.128 nm. In order to correct these two pixels we replaced their value by the average of their two surrounding pixels. When the tangent altitude of the LOS decreases, there is an increasing background signal due to the Rayleigh and/or aerosol scattering outside the $O_2$ band. We corrected the spectra from this signal by removing a straight line computed as a linear interpolation between the two "surrounding" average backgrounds in the [1235-1245] nm domain and in the [1295-1305] nm domain. The spectra after correction are ready to be used for the retrieval of the SCIAMACHY $O_2(^1\Delta)$ volume emission rate (VER) , as described in the next section 3.2.

### 3.2 Retrieval of $O_2(^1\Delta)$ volume emission rate (VER) from SCIAMACHY limb radiances

We have developed a numerical scheme to retrieve the vertical distribution of the $O_2(^1\Delta)$ volume emission rate (VER) from a series of limb radiances obtained during a SCIAMACHY limb scan, taking into account the absorption along the LOS by background $O_2$ of the $O_2(^1\Delta)$ emission between the location of the emission and the spacecraft (sometimes this $O_2$ absorption is called self-absorption, improperly in our opinion, because the $O_2(^1\Delta)$ molecule is different from the background $O_2$, as long it is in the excited state $(^1\Delta)$).

#### 3.2.1 The case with no absorption: the onion-peeling technique

When there is no absorption, the radiance B is related to the integral along the LOS of the VER (or emissivity) $\varepsilon(s)$, s being an abscissa along the LOS.


$$B = \frac{1}{4\pi} \int \varepsilon(s)ds \qquad (18)$$

B is the wavelength-integrated spectral radiance, expressed in photons/(cm$^2$ s sr), while ε(s) is expressed in photons/(cm$^3$ s).

A classical way to retrieve a vertical distribution of ε from a series of radiance measurements at the limb is the onion-peeling method (Fig. 8), which assumes that the emissivity (or VER) is locally spherically symmetric, depending only on the altitude. Furthermore, the problem is discretized by dividing the atmosphere in spherical layers, in which the VER is constant.

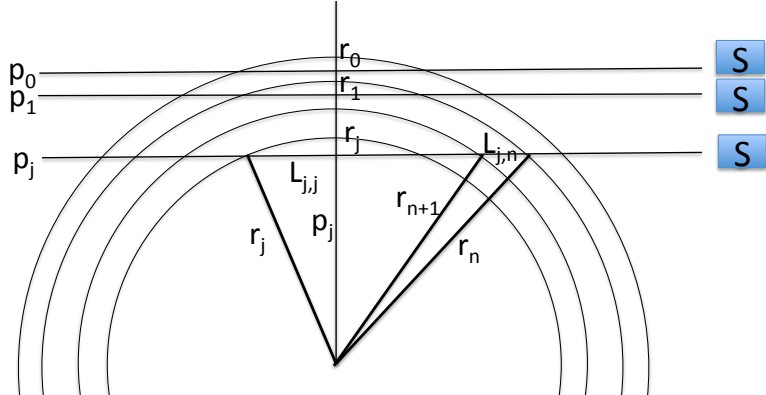

**Figure 8. Geometry of the lines of sight LOS at the limb of an emission organized in spherical layers. The various LOS are obtained either by drifting the observer S into an orbit with a fixed inertial direction (case of GOMOS/ENVISAT), or by angular scanning (case of SCIAMACHY/ENVISAT).**

Let $p_0,..p_j$ be the series of impact parameters of viewing LOS with p= tangent altitude $z + R_{earth}$, $p_0$ the largest.
We define a series of spheres with $r_0 > p_0$ and decreasing radiuses. Let $L_{j,n}$=length of one of the two equal segments (j,n) between spheres of radius $r_n$ and $r_{n+1}$ along the LOS with impact parameter $p_j$.

We have the expressions (19):

$$L_{0,0} = \sqrt{r_0^2 - p_0^2}$$

$$L_{1,1} = \sqrt{r_1^2 - p_1^2}$$

$$L_{1,0} = \sqrt{r_1^2 - p_0^2} - L_{1,1}$$

...

$$L_{j,n} = \sqrt{r_n^2 - p_j^2} - \sqrt{r_{n+1}^2 - p_j^2}$$

For the last segments when j=n, the formula is slightly different because the sphere of radius $r_{j+1}$ is irrelevant:



$$L_{j,j} = \sqrt{r_j^2 - p_j^2}$$

The radiances B(p) may then be expressed (20) as a function of $\varepsilon(z)$:

$$B(p_0) = \frac{1}{4\pi}\,\varepsilon(z_0) * 2 * L_{0,0}$$

$$B(p_1) = \frac{1}{4\pi}[\varepsilon(z_0) * 2 * L_{1,0} + \varepsilon(z_1) * 2 * L_{1,1}]$$

$$\dots$$

$$B(p_n) = \frac{1}{4\pi}[2\sum_k \varepsilon(z_k)L_{n,k}] \qquad (20)$$

This is a linear system of n equations with the n unknowns $\varepsilon(z_j)$. This system may be written under a matrix form
$4\pi\mathbf{B} = \mathbf{M}\,\boldsymbol{\varepsilon}$, **M** being a triangular matrix of order n, the number of atmospheric layers. Elements of the matrix are
the lengths of LOS segments within a layer between two spherical shells. Therefore the matrix may be simply
inverted to yield the vector $\boldsymbol{\varepsilon}$ from the vector of measurements B:

$$\boldsymbol{\varepsilon} = 4\pi\mathbf{M}^{-1}\mathbf{B} \qquad (21)$$

### 3.2.2 The case with absorption: a modified onion-peeling technique

When absorption by $O_2$ is considered, the equation (18) is modified into:

$$B = \frac{1}{4\pi}\int \varepsilon\,(s)\exp(-\tau(s))\,ds \qquad (22)$$

where $\tau(s)$ is the optical thickness between the emission point and the observer (here, assumed to be outside of
the atmosphere). The corresponding attenuation factor is:

$$Tr(s) = \exp(-\tau(s)) \qquad (23)$$

In reality, $\boldsymbol{\varepsilon}$, $\tau(s)$, Tr(s) and B all depend on wavelength $\lambda$. However, the quantities $\tau(s, \lambda)$, Tr(s, $\lambda$) may be
computed for a given atmospheric model defined by a vertical profile of temperature T(z) and pressure p(z), with
the dry air density n(z)= p(z)/k T(z), and $O_2$ density= 0.2095 n(z). We made use of the LBLRTM code and
HITRAN2016 data base to conduct such computations.

The detailed *spectral shape* of the local emission of the $O_2^*$ molecule $\varepsilon(z, \lambda)$ may be computed from the analysis
of Section 2. Therefore, we can compute a *wavelength-integrated* attenuation factor between the emission
location and the external observer, independent of the actual value of the local VER(z)= $\int\varepsilon(z,\lambda)\,d\lambda$ for every cell
of the onion-peeling scheme described above. We keep the matrix approach, but now each element of the matrix
is a length of a segment multiplied by the pre-calculated attenuation factor. While in the standard onion-peeling
scheme, there are two identical segments, symmetric w.r.t. the tangent point, giving equal contributions to the
observed intensity, the attenuation factor is different for the two segments (one in the foreground, the other in the





background).

For a given limb observation, each SCIAMACHY spectra is integrated in wavelength to yield the total limb radiance measurements B(z), and equation (23), where **M** matrix is modified to include the attenuation factor, is used to derive the vertical profile of the volume emission rate $\mathbf{\varepsilon}$(z). It may be shown that the attenuation factor

$FAS_{j,n}$ affecting both foreground and background segments on one LOS may be written as:

$$FAS_{j,n} = 0.5 \frac{\int \tau_{km}(z_n,\lambda) * \left[ \exp\left( -\tau_{j.n}^{f}(\lambda) \right) + \exp\left( -\tau_{j.n}^{b}(\lambda) \right) \right] d\lambda}{\int \tau_{km}(z,\lambda) d\lambda} \qquad (24)$$

where $\tau_{km}(z_n, \lambda)$ is the optical thickness per km of $O_2$ absorption at wavelength $\lambda$ and altitude $z_n$, $\tau_{j.n}^{f}(\lambda)$ and $\tau_{j.n}^{b}(\lambda)$ are respectively for the foreground segment and the background segment the optical thickness of $O_2$

absorption from the segment to the observer, along the LOS.

### 3.2.3 Comparison of methods used by others

Zarboo et al. (2018) have also used the SCIAMACHY limb scans in order to retrieve the vertical profile of the

$O_2*$ emissivity. However, their technique is quite different. They find a best fit to the whole series of observed spectra with a model of the vertical distribution of spectral emissivity, but they do not account for attenuation by $O_2$. Therefore, they must underestimate their emissivities, the more the lower latitude. Their method does not need to know the state of the atmosphere nor the theoretical shape of the emissivity; in principle, if their SNR would be large enough, they could interpret the *spectral shape* of their retrieved emissivities in terms of local

temperature, by comparing with model predictions built on our approach developed in Section 2. Because they have used a special SCIAMACHY mode of observation dedicated to the mesosphere and lower thermosphere (MLT), in which the scanning at the limb is in the range 50-150 km, the fact that they did not account for $O_2$ re-absorption along the LOS is probably not very important. They have limited their study to altitudes >50 km.

Sun et al. (2018) have studied also the SCIAMACHY spectra in the $O_2*$ band for the same purpose as us: for a

better retrieval of the $X_{CO2}$ mixing ratio. In order to retrieve the spectral emissivities at each altitude, they have developed two methods. The first one (they call it "linear inversion") is identical to the method of Zarboo et al. 2018, and does not account for $O_2$ absorption. In their second method (they call it "onion-peeling"), they account for $O_2$ absorption, except for the two tangent heights where absorption is negligible, from which they can derive the local temperature T by comparison with a model of the spectral local emissivity quite similar to what we

developed in Section 2. Then, they use the MSIS model pressure to get the $O_2$ density to compute the absorption in the second layer from top and propagate downward the computations. They can therefore compute the vertical temperature profile at each altitude.

### 3.2.4. Sensitivity of VER retrieval to the choice of atmospheric model and nadir radiance estimate

In our retrieval scheme of VER from observed limb radiances, an atmospheric model is needed to compute the absorption by background $O_2$. We tested the sensitivity of the VER retrieval to the choice of the atmospheric model, as shown on Fig. 9 for a SCIAMACHY limb observation obtained on January $1^{st}$, 2007 at (lat=-39.9°, SZA=37.2 °, from product 20070101_1256). The various VER profiles were obtained either with US





STANDARD (a mean model for all seasons and latitudes), or with the variants SUBARCTIC_SUMMER, SUBARCTIC_WINTER, and MIDLAT_SUMMER (this last one should correspond best to the conditions of this particular observation). The relative differences may reach ± 10% between 40 and 60 km, with smaller differences above 60 km (less absorption) and larger differences below 35 km (more absorption, but anyway the

VER is small). Therefore, for our following studies of many SCIAMACHY limb scans we have systematically used the most relevant standard profile, according to the latitude and season: a so-called CLIMATO_ADAPTED atmospheric profile ("adapted climatology").

In particular, for each studied limb scan we have first inverted the SCIAMACHY limb total radiances to retrieve a VER vertical distribution. Then we could compute what would be the nadir radiance in the $O_2$ band that should

have been observed with such a profile, to simulate the MicroCarb geometry of observation or any other GHG monitoring system. The vertical integration was done above 30 km up to 80 km to be consistent with REPROBUS model which stops at 80 km. Absorption by $O_2$ may be computed in the nadir viewing geometry, though attenuation in this geometry is small (2% for the Q branch, less outside of the Q branch).

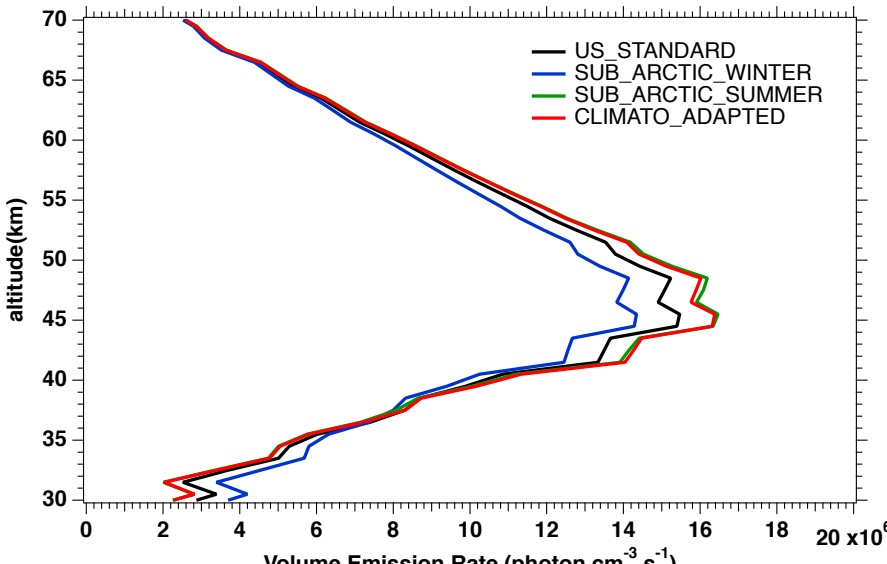

**Figure 9: VER airglow profiles obtained by taking into account $O_2$ absorption using a climatological atmospheric profile SUBARCTIC_SUMMER (green curve), SUBARCTIC_WINTER (blue curve), US STANDARD (black curve), and CLIMATO_ADAPTED (red curve). In the case of this profile measured in January at latitude=-39.9 ° the adapted climatology corresponds to the MIDLAT-SUMMER climatology.**

On Fig. 10 are plotted the nadir radiances (equivalent to intensities or brightnesses) derived from a series of SCIAMACHY limb scans along one particular orbit, as a function of Solar Zenith Angle (SZA), when different atmospheric models are used. For each model, there are two branches, corresponding to North and South along the dayside polar orbit of ENVISAT (the North branch is in winter, while the South branch is in summer for this orbit). We see that the choice of the atmospheric model in the computation of the $O_2$ absorption has a small

(~3%) but noticeable impact (on the brightness seen at nadir). We have also plotted the prediction of the REPROBUS model, as described in Section 4 and subsection 4.2.1. It should be noted that the choice of the



"adapted climatology" makes it possible to reduce the separation between the two branches and thus to be closer to the separation between the two branches obtained with the REPROBUS model.

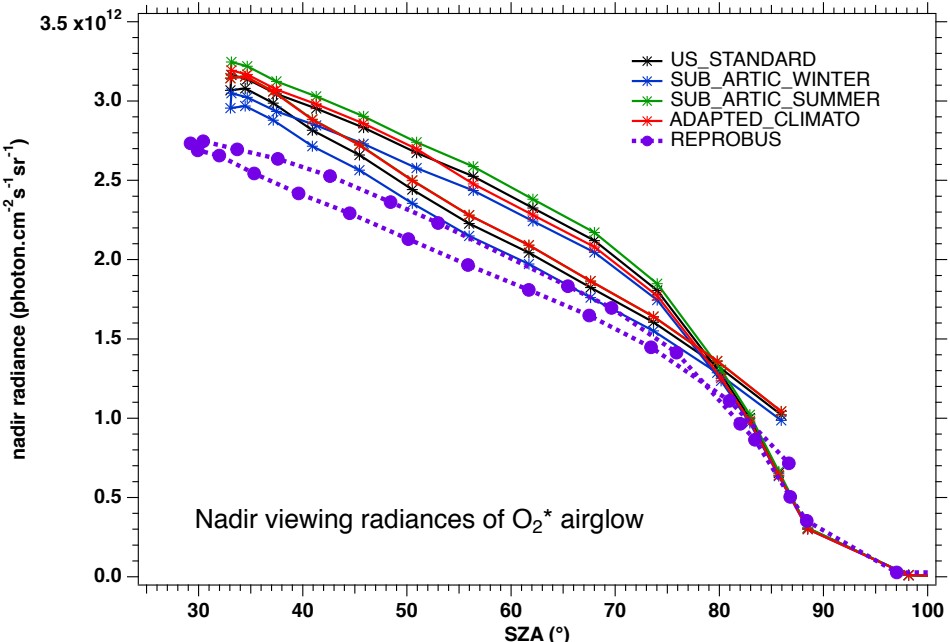

**Figure 10. Computed $O_2^*$ radiances in nadir viewing geometry, derived form SCIAMACHY limb radiances, as a function of SZA for orbit 20070101_1256 when the $O_2$ absorption is computed with various choices of atmospheric models: Climatology US_STANDARD (black), SUBARTIC_WINTER (blue), SUBARCTIC_SUMMER (green) and CLIMATO_ADAPTEE (red). There is a slight dependence of the nadir intensity on the choice of atmospheric model. The purple dashed curve (with filled circles) corresponds to the REPROBUS v02 model.**

### 3.3 Computation of synthetic spectra and comparison with SCHIAMACHY observed spectra

Once we have the vertical profile of VER corresponding to a given SCIAMACHY limb scan, we can compute the spectrum of the local emissivity (in absolute units of photons/ (cm$^3$ s$^1$ sr nm)) with the theoretical approach developed in Section 2. Then we may integrate the spectra with Abel's integral along horizontal LOS tangent at the limb, for a direct comparison with the actually observed SCIAMACHY spectra. In this particular exercise, we did not account for the $O_2$ absorption for simplicity, and for this reason we restricted our comparison to altitudes >60 km. The spectral resolution of SCIAMACHY was used to smooth the high resolution spectra (line by line) obtained from the approach described in Section 2.

On Fig. 11 are represented the locations of the tangent points of SCIAMACHY limb scans for a particular orbit of ENVISAT. On Fig.12a are represented the observed spectra, binned by altitudes (60-70 km, 70-80 km, and 80-90 km), along with our model spectra computed for the same scans and binned in the same way, for a particular limb scan (points in green on Fig. 11). The agreement is basically very good, both in shape and intensity. We note that the model is slightly brighter than the data, and the relative difference is larger for the bin 60-70 km than for the other bins. We assign this behaviour to the fact that we have neglected the $O_2$ absorption





along the LOS in the model, more important at 60-70 km than higher.

Figure 12b is the same as Fig.12a, with the simplified model in which the spectral shape of the $O_2^*$ emission is identical to the $O_2$ absorption. In this case, the R branch is systematically overestimated by this simplified model.

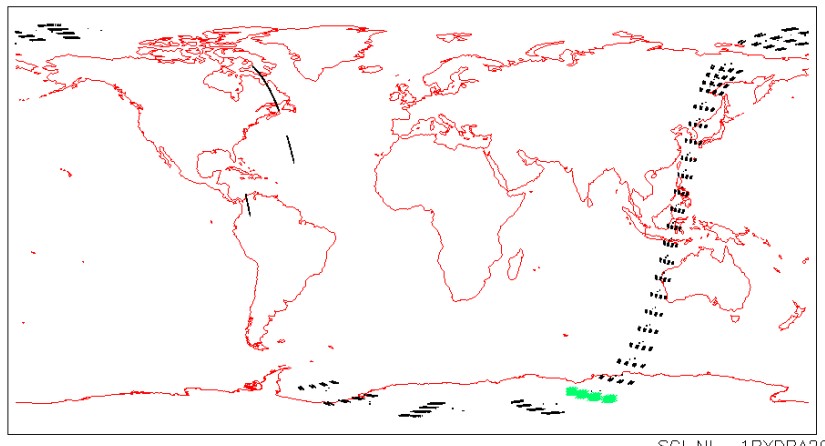

5      **Figure 11. Geographic positions of SCIAMACHY LOS tangent points at the limb (by groups of four) for ENVISAT orbit 20070101_1256. The green points are the locations of the limb spectra which are compared with our theoretical derivation.**

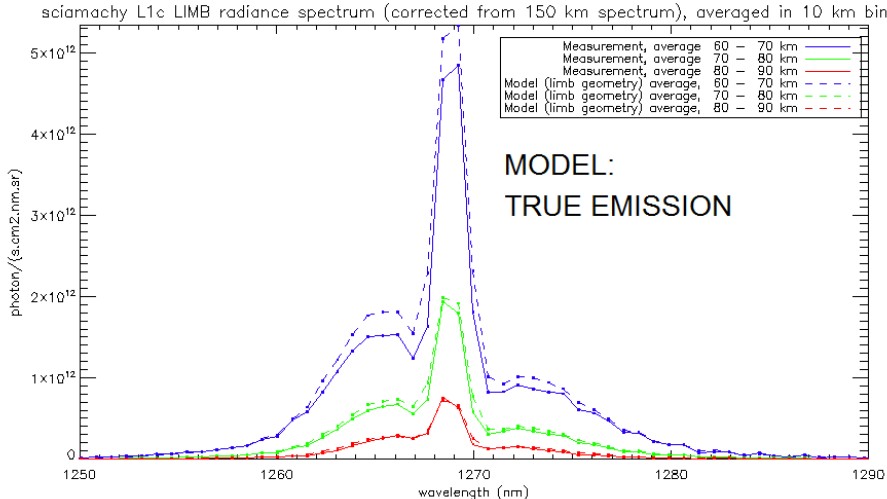

**Figure 12a. SCIAMACHY limb spectra (solid lines, absolute units are photons/(s cm$^2$ nm sr)), binned by altitudes (60-**
10    **70 km, 70-80 km, and 80-90 km), along with our model spectra computed for the same scans and binned in the same way.**





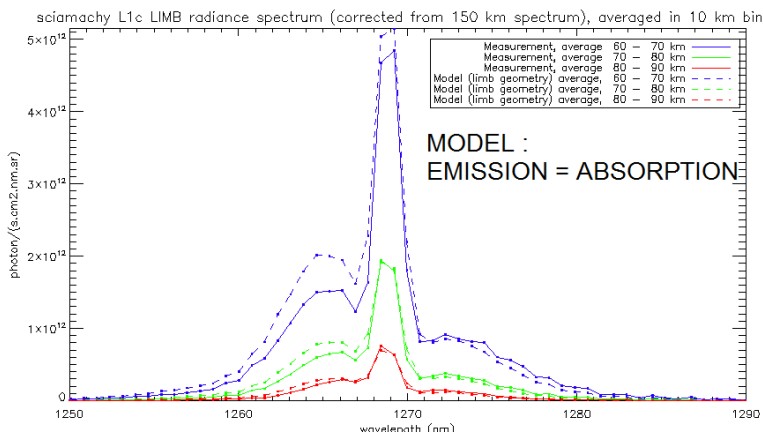

**Figure 12b. Same as Fig. 12a, but with the simplified model in which the shape of the emission of $O_2^*$ is identical to the absorption by $O_2$. This simplified model shows an excess of emission in the R branch (left) and a deficit in the P branch.**

The ratio of spectra measured /model, $S_{obs}$/ $S_{mod}$ were averaged together for all scans of that particular orbit within the same three altitude bins. They are represented on Fig. 13, both for the simplified model (absorption = emission, left), and for our "true" model of emission (right). It is clear that the simplified model does not represent well the observed spectra, while the model with the true emission agrees quite well with the data. This

10 comparison validates completely the approach that we developed in Section 2. It can be noted that the overall level of the ratio is slightly below 1 (right panel). Again we assign this behaviour to the fact that we have neglected the $O_2$ absorption along the LOS in the model, and it can be seen that the ratios are nearer 1 for larger altitudes. Below 1255 nm and above 1285 nm, the intensity of the spectra is very small and thus we attribute the noisy shape of the ratio spectra to low SNR.

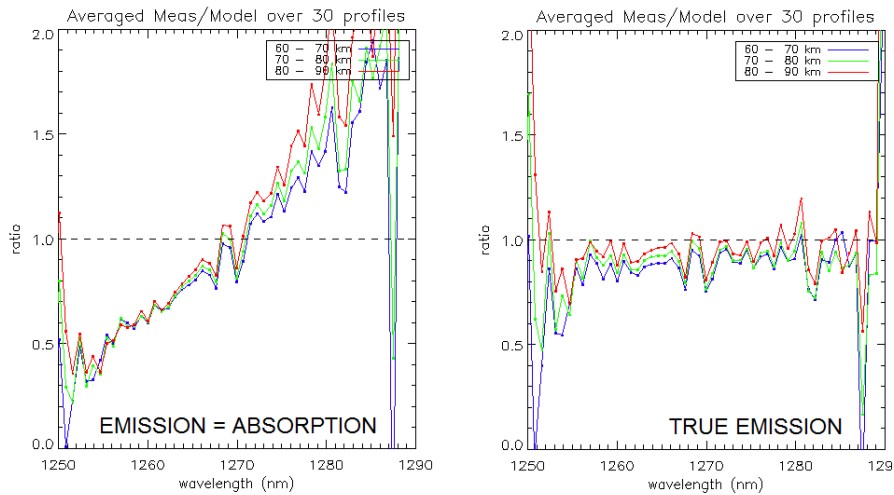

**Figure 13: Ratios of measurements/model of limb spectra, averaged over a whole ENVISAT orbit, and binned by altitudes (60-70 km, 70-80 km, and 80-90 km). Left: simplified model in which the shape of the emission of $O_2^*$ is**


identical to the absorption by $O_2$. Right: same ratios with our new model described in Section 2. The ratios are nearer 1 for larger altitudes because absorption by $O_2$ is neglected in this particular exercise.

**3.4 Climatology of $O_2$* VER derived from SCIAMACHY limb radiances**

To build up a climatology of the $O_2$* emission at 1.27 μm, we have applied our inversion scheme to get VER vertical distributions (sub-section 3.2) to all SCIAMACHY limb data collected during the first 3 days of each month of year 2007. Note that in the normal mode, the limb scans extend down to 0 km (our inversion is made >30 km), while in the special SCIAMACHY MLT mode, only altitudes >50 km are observed. Our data base
contains the analysis of 448 orbits, containing 12,400 limb scans which go down sufficiently for our purpose (some limb scans do not reach low enough altitudes).

The vertical inversion of SCIAMACHY limb radiances to get a VER vertical profile is done below 90 km, down to 0 km; but only results>30 km are significant, because at the limb and low altitudes, there is Rayleigh and aerosols solar radiation scattering which dominates over the $O_2$* radiance and pollutes the SCIAMACHY
measurements. Once a VER profile is obtained, it can be integrated vertically, taking into account the absorption by $O_2$. Therefore, a "SCIAMACHY" nadir radiance is obtained, which corresponds to the $O_2$* radiance that would be observed by SCIAMACHY if it were observing nadir at the position of tangent points where the limb radiances were obtained. In fact, the nominal operation mode of SCIAMACHY does indeed alternate limb-viewing and nadir-viewing observations to discriminate tropospheric ozone from stratospheric ozone (Ebojie et
al., 2014).

When this VER is integrated vertically to get a nadir radiance, the integration stops at the upper limit of 80 km in order to have a better comparison with the REPROBUS model which stops also at 80 km. The air atmospheric model selected to compute the re-absorption by $O_2$ is our so-called "adapted climatology" (see above). On Fig. 14 are displayed about one third of all VER profiles collected for the first three days of January 2007 (other
months are quite similar). The colour code corresponds to the SZA of the limb scan. We kept also scans near the terminator, where the VER is significant only above 80 km. Clearly the SZA is the factor dominating the shape, the peak altitude and the intensity of the airglow VER profiles between 30 and 80 km. This is due to UV photo-dissociation of ozone (the main process of $O_2$* production) penetrating more deeply when the SZA is small (because of ozone UV screening). The lower the SZA, the brighter is the airglow emissivity. Above 80 km, other
processes come into play and a second airglow peak is observed which seems less correlated with the SZA than is the main peak at 45-50 km. The altitude of the main airglow emissivity peak varies between 43-45 km for values of SZA below 50° and increases for higher values of SZA up to about 60 km.

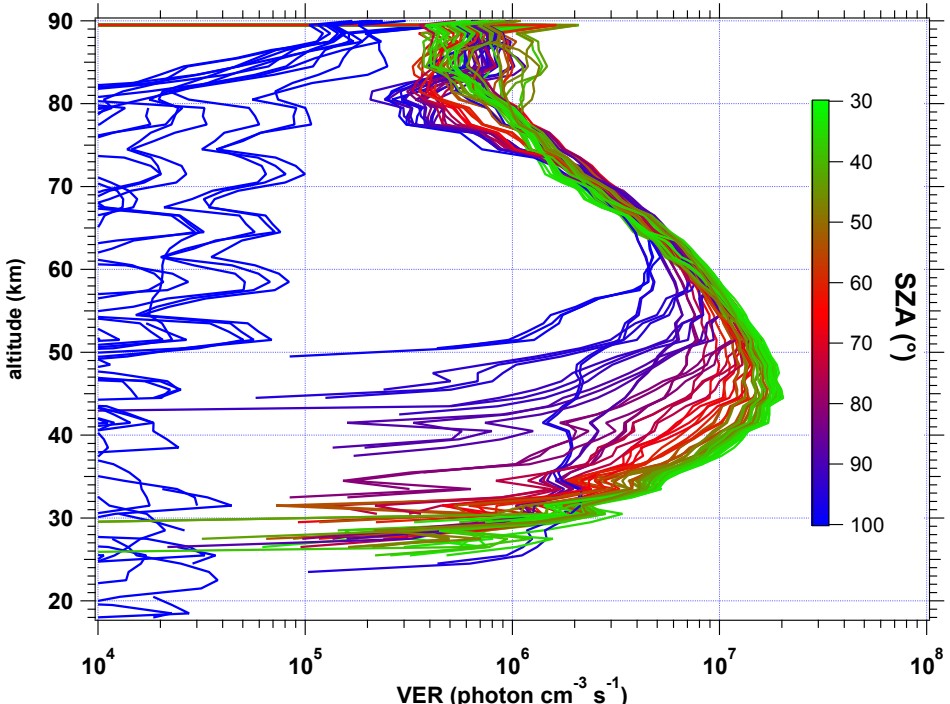

**Figure 14. Volume emission rate (VER) profiles of airglow at 1.27 µm retrieved from SCIAMACHY limb data for the first of January 2007 (80 profiles). The colour scale represents the SZA (Solar Zenith Angle). For the lowest SZA values probed by SCIAMACHY (33°), the peak VER is $2\times10^7$ photons $cm^{-3}$ $s^{-1}$ around 45 km. At large SZA values, the emission is present only at high altitudes (>80 km). Above 90°, there is almost no signal for inversion.**

On Fig.15 are mapped (longitude-latitude) all the "SCIAMACHY" nadir radiances obtained by inversion of limb radiances and integration of VER (first 3 days of each month of year 2007) for 4 typical months, January, April, July and October. The airglow brightness is almost independent of longitude. At high latitudes (North and South), the brightnesses are lower: this is the effect of larger SZA. The region of maximum brightness is displaced with season, following the latitude of the sub-solar point, again an effect of the SZA dependence of the $O_2^*$ radiance. This is illustrated on Fig. 16 where all the nadir radiances are plotted as a function of SZA, with a colour code on latitude. The lower SZA, the brighter is the nadir emission. Still, there is a separation of the curves in two branches that are relevant to northern and southern hemispheres. The separation between the two branches depends on the season. As we will see later on, the overall pattern of the $O_2^*$ radiance is directly linked to the climatology of upper stratosphere/ lower mesosphere ozone.



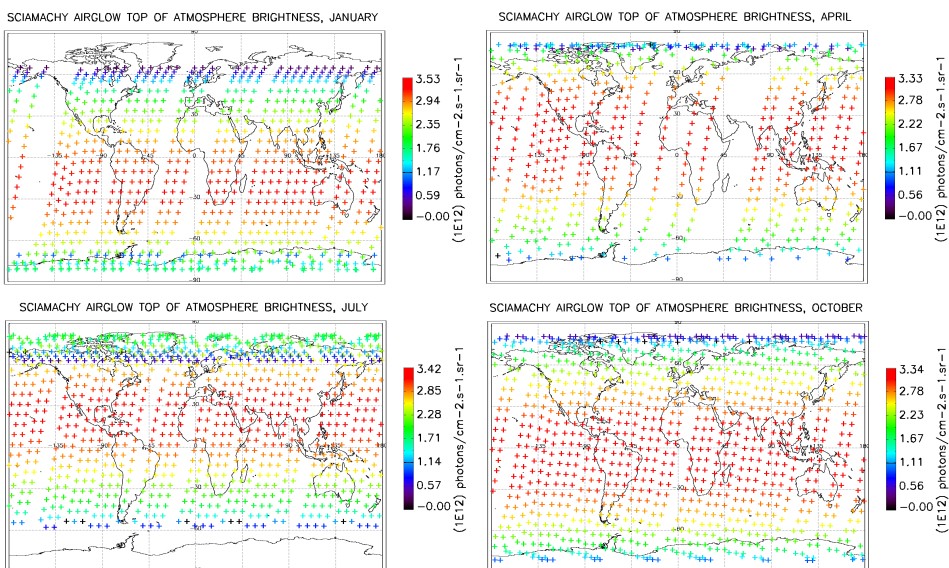

**Figure 15.** Airglow brightness maps as seen from space in nadir view, retrieved from SCIAMACHY limb viewing data for the months of January, April, July and October 2007 (first 3 days of each month only). The color scale represents brightness. Zones without data (holes), particularly numerous in April, are corrupted products that have been eliminated. SZA points > 90° have been eliminated.

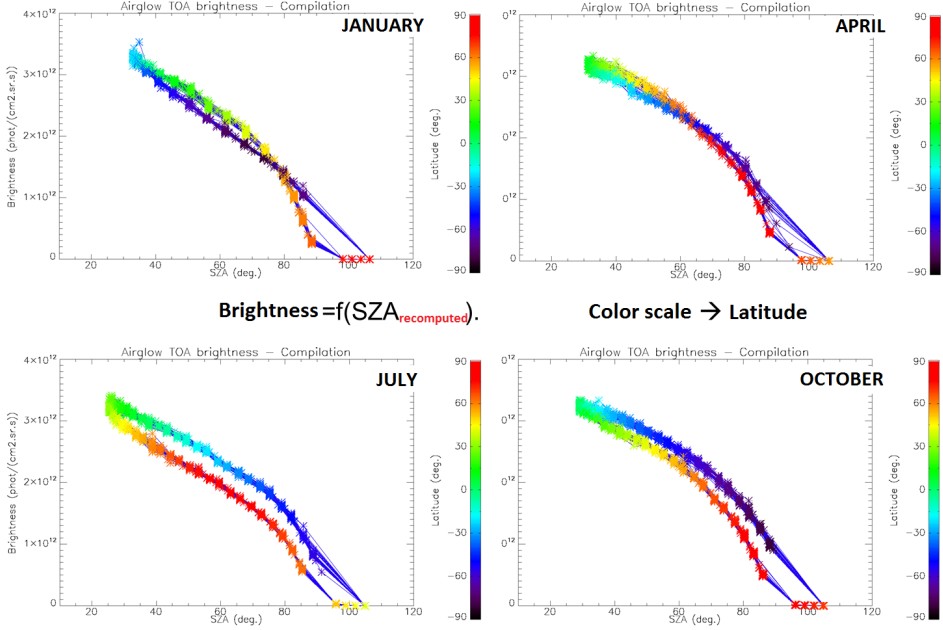

Brightness $= f(SZA_{recomputed})$.        Color scale → Latitude

**Figure 16.** $O_2^*$ airglow intensities that would be seen at nadir as a function of SZA for the months of January, April, July and October 2007 (first 3 days of each month only), retrieved from the processing of SCIAMACHY limb data. The colour scale represents the latitude. There is a geometrical correlation between the latitude and SZA imposed by the polar orbit of ENVISAT. The comparison of these intensities with those obtained by the REPROBUS airglow model is presented in subsection 4.2.1.





**4. Comparison between an airglow model based on REPROBUS and SCIAMACHY observations**

In this section we compare the predictions of a dedicated 3D model of the airglow emission of $O_2(a^1\Delta_g)$ at 1.27 μm to the airglow observations of SCIAMACHY. The comparison makes use exclusively of the SCIAMACHY

limb observations, but is made in two different ways. One way is to compare the SCIAMACHY VER (Volume Emission Rate) vertical profile retrieved from limb measurements through vertical inversion as described above in sub-section 3.2.2. The second way is to compare the nadir integrated emission $I_{ag}$ (brightness) of the airglow. Both model and data nadir emissions are obtained by vertical integration of the VER, respectively in the airglow model and in the SCIAMACHY-derived VER vertical profile. This nadir emission is directly relevant to the

GHG observations since, from an orbiter and nadir viewing, this signal is superimposed on the solar back-scattered emission from which the columns of GHG gases and $O_2$ must be retrieved. This is why it is not practical to use the nadir observations of SCIAMACHY to study the $O_2*$ airglow, since the nadir signal is dominated by surface back-scattered solar radiation (except in special conditions, over the seas, as we shall see in Sub-section 6.2.2).

Since the photolysis of ozone is the major source of the $O_2(a^1\Delta_g)$ airglow, it was also felt necessary to compare the ozone density predicted by our airglow model and GOMOS (Global Ozone Monitoring by Occultation of Stars) ozone measurements also on ENVISAT, simultaneous with SCIAMACHY observations (but not with the same geometry).

**4.1.  3D simulation of the airglow emission of $O_2(a^1\Delta_g)$ at 1.27 μm**

The airglow model is composed of two separated elements. The first element is the REPROBUS CTM (Chemistry Transport Model) computing the 3D distribution of ozone and other chemical species as a function of time, driven by analysed meteorological fields. The second element is an airglow model operated offline, which extracts from REPROBUS (for one location and one precise time and date) the information necessary for

the computation of the relevant VER profile.

**4.1.1.  REPROBUS 3D simulations**

REPROBUS is a global CTM (Chemical Transport Model) developed for the stratosphere (Lefèvre et al., 1994).

It includes a complete description of stratospheric chemistry using 58 species and about 100 chemical reactions. The winds and temperatures used by REPROBUS are forced by the ECMWF operational analyses, over a domain that extends from the ground to 0.01 hPa (about 80 km) and a horizontal resolution of 2°×2°. For the present study, we carried out a REPROBUS simulation covering the whole year 2007 with the results saved every hour. The choice of 2007 was motivated by the fact that we had already extracted SCIAMACHY data for

this year. From this new simulation, all the GOMOS or SCIAMACHY data obtained in 2007 can be compared to the CTM with a spatial difference less than or equal to 1° and a time difference less than or equal to 30 mn. It should be noted that for GOMOS the comparison with the model is limited to ozone profiles, since GOMOS does not have a channel at 1.27 μm and therefore does not observe airglow at this wavelength. SCIAMACHY observations of the 1.27 μm airglow were compared to the combination of REPROBUS and the off-line airglow

model.





Based on the results of REPROBUS available every hour of 2007 and the off-line airglow model, we have developed a procedure for the automatic extraction of vertical ozone profiles and $O_2(a^1\Delta)$ emission profiles as well as the integrated $O_2 (a^1\Delta)$ emission in coincidence with the GOMOS and SCIAMACHY measurements performed the same year. This dataset represents 4,026 profiles modelled in coincidence with GOMOS and

12,800 in coincidence with SCIAMACHY. The statistical analysis of the comparison between the model and observations is presented in sub-section 4.2 for SCIAMACHY observations and in sub-section 4.3 for GOMOS observations.

### 4.1.2. Simulation of airglow emission of $O_2(a^1\Delta_g)$ at 1.27 µm

Here we do not care about the details of the spectral shape of the emission, but rather we compute the local emissivity (VER) and the vertically integrated emission, in order to compare with SCIAMACHY observations. The airglow at 1.27 µm is calculated off-line from the 3D outputs of the REPROBUS model. It takes into account all the mechanisms of production and loss of $O_2(a^1\Delta)$, as detailed in Appendix A and shown in Fig. 17.

In practice, the $O_2(a^1\Delta)$ emission model uses as input the ECMWF temperature and pressure profiles as well as the $O_3$ and $O(^3P)$ profiles calculated by REPROBUS for the selected date and location. From the pressure and temperature are also calculated the total density and density profiles of $N_2$, $O_2$, and $CO_2$. The airglow model then provides the vertical profiles of the mixing ratios of $O(^1D)$, $O_2(b^1\Sigma)$, $O_2(a^1\Delta)$, the vertical profile of the volume emission rate (VER) at 1.27 µm expressed in $photon.cm^{-3}.s^{-1}$, and the vertically integrated intensity expressed in

$photon.cm^{-2}.s^{-1}.sr^{-1}$ (brightness or intensity, directly comparable to the radiance signal of the back-scattered solar radiation, atmosphere + aerosols+ surface).

Two versions of the airglow model were used. One early version of the model (v01) was later modified to a version v02 which yielded better agreement with SCIAMACHY observations. They differ only by the value of the quenching rate of the $O_2(^1\Delta)$. The early version v01 contained a quenching constant k:

$k_{\Delta,O2} = 3.6e^{-18}$ x exp(-220/T)     in $cm^3.molecule^{-1}.s^{-1}$, T = temperature (K)

recommended in the JPL compilation (Burkholder et al., 2015). The version v02 has a value of k slightly different, recommended by IUPAC (Atkinson et al., 2005):

$k_{\Delta,O2} = 3.0e^{-18}$ x exp(-200/T)     in $cm^3.molecule^{-1}.s^{-1}$, T = temperature (K).

At stratospheric temperatures, the value of $k_{\Delta,O2}$ is decreased with v02 by about 10%, enhancing the emission

rate of $O_2(^1\Delta)$. This gives a better fit (but not perfect) between SCIAMACHY observations and the airglow model. According to Wiensz (2005), this IUPAC recommended value gives a better agreement between OSIRIS/ODIN direct and indirect measurements of ozone. Unless otherwise specified, we are presenting in this paper the v02 results.



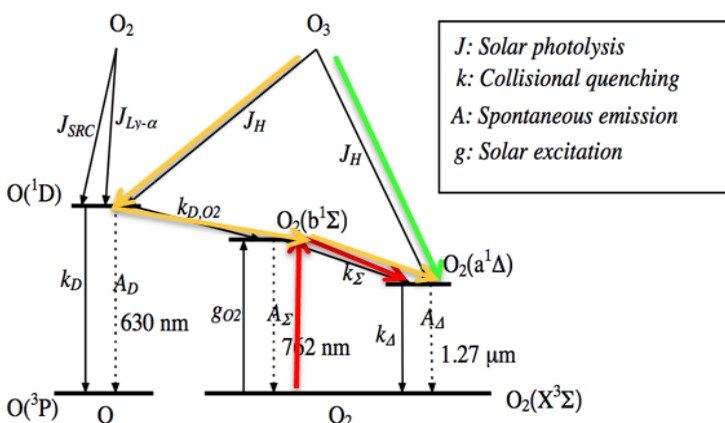

**Figure 17. Energy diagram of O and O$_2$ molecule showing both O$_2$ bands at 762 nm (A) and 1.27 µm. Coloured arrows are the pathways modelled to reach O$_2$* (a$^1\Delta$) with mechanisms of production and loss of O$_2$(a$^1\Delta$) taken into account in the airglow model. Only the O$_2$ photo-dissociation (J$_{SRC}$ (Schumann-Runge continuum) and J$_{Ly-\alpha}$) is not taken into account in our model, since this represents only about 1% of the integrated emission (reproduced and adapted from Wiensz, 2005).**

### 4.1.3. Some examples of model results

As an example, Fig. 18a,b,c show the REPROBUS results for 21 June 2007 at the pressure level 0.9 hPa (about 50 km) which corresponds to the altitude of maximum emissivity of O$_2$(a$^1\Delta$). The figures displays the O$_3$ volume mixing ratio (Fig. 18a), the corresponding volume emission rate of O$_2$(a$^1\Delta$) calculated by the airglow model (Fig. 18b), as well as the vertically integrated O$_2$(a$^1\Delta$) emission (Fig. 18c) (the possible reabsorption between the emission point and the top of the atmosphere by O$_2$ is here neglected).

On 21 June 2007 the volume emission rate of O$_2$(a$^1\Delta$) at 1.27 µm shows a maximum at high southern latitude that is obviously caused by a maximum of O$_3$ at the same location. However, the O$_2$(a$^1\Delta$) emission is very strongly modulated by the solar zenith angle. This effect is further exacerbated when the emission is vertically integrated: the intensity of the O$_2$(a$^1\Delta$) emission is then systematically maximum at local noon and its variations are entirely controlled by the solar zenith angle. At a given solar zenith angle, the intensity shows very little spatial variability and the effects of the heterogeneity of the ozone field are almost completely erased.



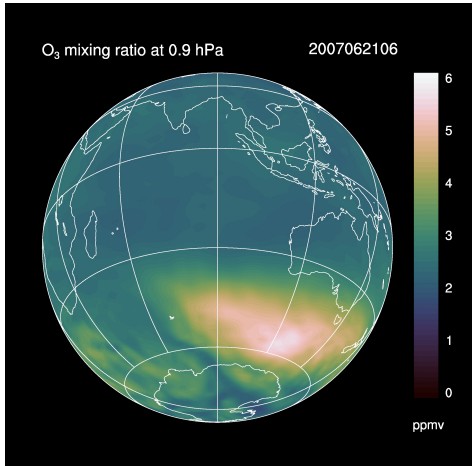

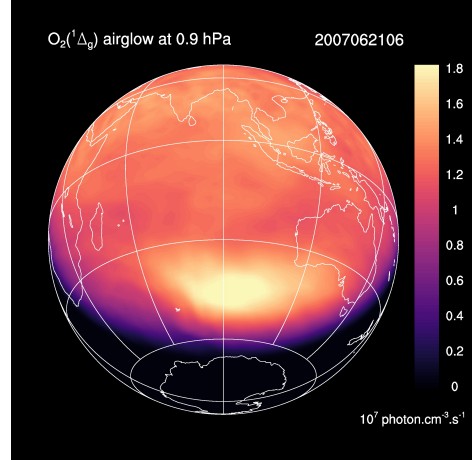

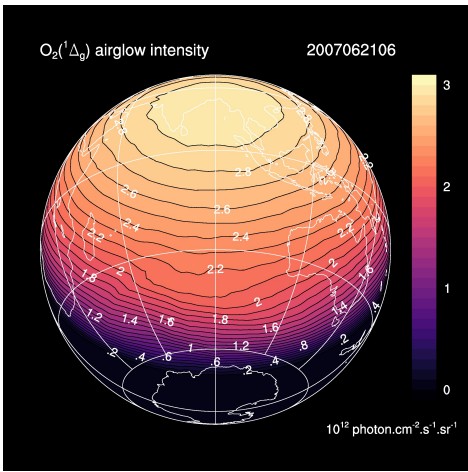



**Figure 18a,b,c. Simulations of the REPROBUS model for June 21, 2007 06:00 UT. This is the geographical distribution of three relevant quantities with their color code. Top: Ozone mixing ratio (ppmv) at 0.9 hPa (about 50 km). Middle: Volume emission rate of $O_2(a^1\Delta)$ at 0.9 hPa, in units of $10^7$ photon.cm$^{-3}$.s$^{-1}$. Bottom: Integrated vertical intensity of $O_2(a^1\Delta)$, in units of $10^{12}$ photon.cm$^{-2}$.s$^{-1}$.sr$^{-1}$.**

**4.2. Comparison of SCIAMACHY data with REPROBUS derived airglow model**

For each observation of our SCIAMACHY 2007 data set in limb viewing we have a co-located VER profile of $O_2(a^1\Delta)$ calculated by the REPROBUS-based airglow model. We were therefore able to make comparisons between the airglow of SCIAMACHY and that of REPROBUS in two different ways: i) The brightness of the

airglow as it would be seen by a TOA (Top Of Atmosphere) observer in nadir viewing. ii) the VER (Volume Emission Rate) airglow vertical profiles.

**4.2.1     Comparison of $O_2(a^1\Delta)$ airglow brightness as seen by a TOA observer in nadir view**

The airglow model brightness is obtained by a vertical integration of the VER produced by the airglow model. Re-absorption by $O_2$ in nadir geometry is small and has been neglected. However, this nadir model intensity cannot be directly compared with a nadir observation of SCIAMACHY, as it is most of the time completely dominated by terrestrial albedo. Therefore, to evaluate the nadir intensity corresponding to the SCIAMACHY data in limb viewing we proceeded as follows:

- for each vertical scan at the limb with SCIAMACHY, the total brightness of the airglow was first estimated by integrating spectrally the SCIAMACHY spectra at each altitude, and then the VER profile was determined with an onion-peeling method (taking into account horizontal re-absorption), as described in sub-section 3.2.

-then the VER was vertically integrated to yield  the intensity (or brightness) that an observer placed at the tangent point of the SCIAMACHY scan would see looking to nadir.

Figure 19 compares the nadir intensities as a function of SZA co-located for SCIAMACHY and REPROBUS for the first three days of January, April, July and October 2007. This represents the data collected over ~ 50 orbits of ENVISAT for each considered month, which is almost polar and descending in latitude on the day side (equator crossing around 10:30 Local Time,  descending node). In both data and model the SZA is the dominating factor on the intensity. The latitude (which is colour coded in the plot) plays also a small role

(through the ozone field), more important in July. The repeatability of the SCIAMACHY derived nadir intensities is obvious, with very little dispersion. The main difference between data and model is that at SZA < ~70° the REPROBUS/airglow model systematically underestimates by 10-20 % the airglow intensity compared to that seen in the SCIAMACHY data.

***Note regarding the SZA of the SCIAMACHY data:*** We noted that the SZA values provided in the

SCIAMACHY ESA products in limb viewing are different from the SZA values calculated by the REPROBUS model for the same location and time. The difference can sometimes reach 9°. This is even the case when the SCIAMACHY observation time is exactly the same as in the REPROBUS model (i.e. at "all round" times, for example 10:00, 11:00, 12:00,...). Investigations were carried out involving email exchanges with Gunter Lichtenberg of the DLR and we came to the conclusion that the SZA stored in SCIAMACHY products in limb

viewing (calculated using the CFI ENVISAT libraries) is not the SZA of the tangent point of the line of sight (LOS) but that of one of the two points corresponding to the intersection between the line of sight and TOA





(defined at 100 km altitude), which is incorrect. Therefore, we systematically recalculated the SZA at the tangent point of the SCIAMACHY data using an external tool (IDL routine). All results presented in this report are obtained using this recalculated SZA.

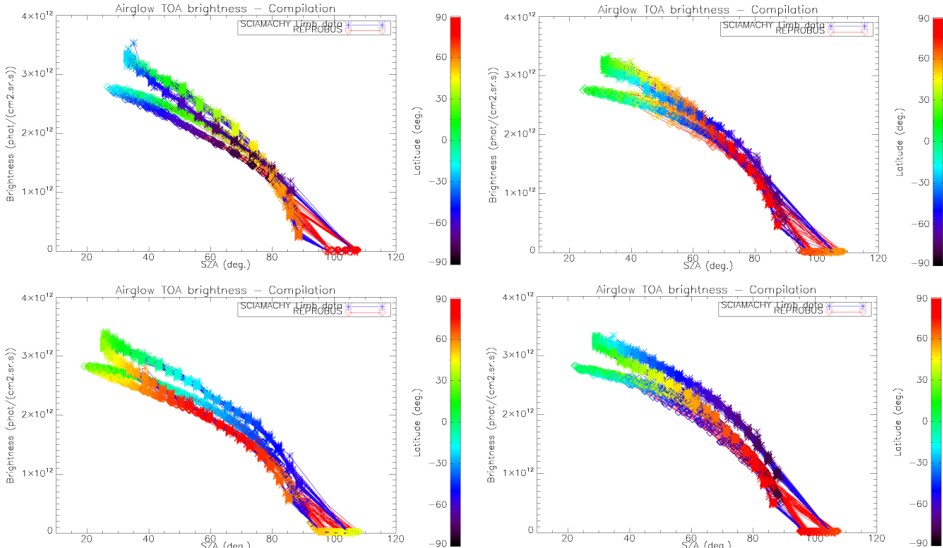

**Figure 19. Airglow intensity seen at nadir as a function of SZA for the months of January, April, July and October 2007 (first 3 days of each month only). Data (star symbol) come from the SCIAMACHY limb data (retrieval of VER with onion-peel inversion and subsequent vertical integration); (diamond symbol) from the REPROBUS/airglow model. The colour scale for symbols represents latitude. The SZA for the SCIAMACHY curve was recalculated with**
10 **an external tool.**

Note that in Fig. 19 there is a remaining difference between the minimum SZA of the SCIAMACHY data and the minimum SZA of the model of up to about 8°. This difference may be explained due to the time difference between the model and the data since, unlike the data, the model was calculated on a fixed time grid with a
15 round hour (e.g. 10:00, 11:00, etc.). There is therefore a time difference between model and data of up to 30 minutes, which can be both ways in difference of SZA, SZA(model)-SZA(data). The true SZA is the data one; and the model SZA may be the same as the data ± 8°, but only the negative differences SZA(model)-SZA(data) are obviously visible on the plot, when SZA(data) is at its minimum value, and SZA(model) is below SZA(data). Points with positive differences are just mixed with all other points.

**4.2.2 Comparison of VER vertical profiles**

We have seen systematic differences between the nadir intensities (vertically integrated VER) of SCIAMACHY and REPROBUS. It is therefore interesting to pinpoint at which altitude the differences are essentially located,
by directly comparing the vertical VER profiles produced by the REPROBUS model and those that could be derived from the SCIAMACHY limbs by onion-peel inversion. The comparison of some typical VER profiles of



SCIAMACHY and REPROBUS is illustrated in Fig. 20 and 21. It shows that in the lower part of the profiles, say up to 40 km altitude, there is a good match of VER values between SCIAMACHY and REPROBUS. At higher altitudes >40 km, REPROBUS/airglow model predicts less $O_2$ ($a^1\Delta$) airglow than observed by SCIAMACHY.

5     One obvious possible reason for this discrepancy would be the radiometric calibration of SCIAMACHY in this airglow band. For the time being, we reject this hypothesis for two reasons. The first is that the radiometry of SCIAMACHY must have been verified with nadir viewing observations when the surface albedo is dominating, and comparing to MODIS results for instance. The observation of Deep Convective Clouds (DCC) found in the tropics, which begin to become a standard in Earth Observations radiometric calibration, might be an additional

10     source of comparison. The second reason is that the onion-peel inversion scheme that we have designed to derive VER vertical profiles is a linear one. Therefore, changing the calibration of SCIAMACHY by a scaling factor would also change the VER profile by the same factor, while we see that the VER discrepancies are changing with altitude. Other sources of discrepancies might come from the ozone distribution as examined in sub-section 4.3 below, or in some details of the airglow model.

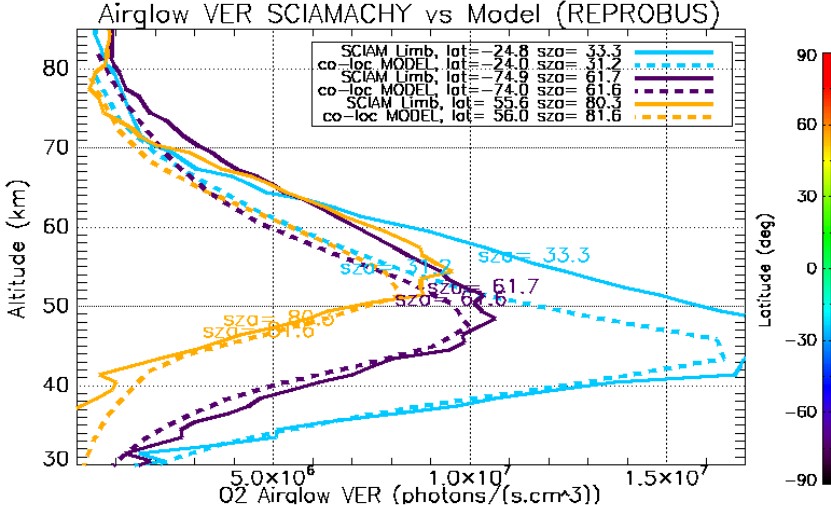

**Figure 20. Comparison of SCIAMACHY and REPROBUS airglow VER profiles for January 3, 2007. Three profiles were drawn at SZAs of about 30°, 60° and 80°, indicated in the legend. The SCIAMACHY profiles are plotted as solid lines and the geolocated REPROBUS profiles are in dotted line. The colour scale represents latitude.**



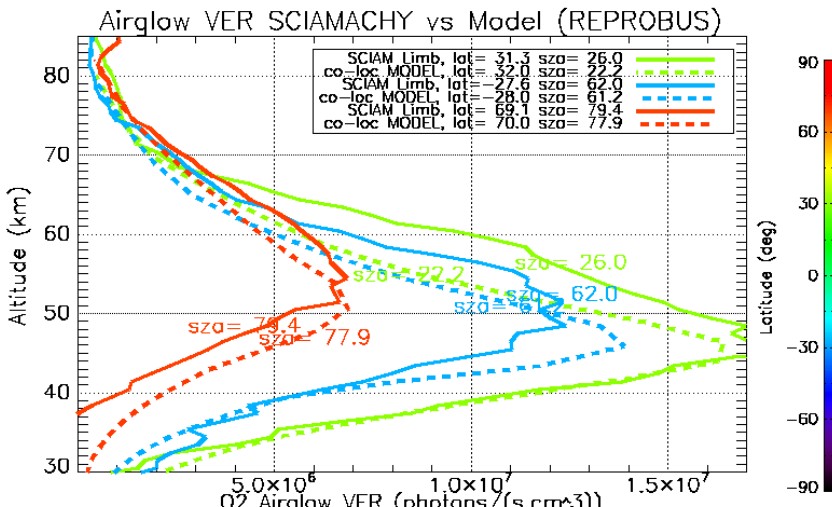

**Figure 21. Same as Fig.20 for three profiles acquired on July 3, 2007.**

The same type of comparison is presented on Fig. 22 (top) for a full set of SCIAMACHY limb observations

5    obtained during the first three days of January 2007, but still selecting the SZA of tangent points in slices around 30°, 60° and 80°, with a different colour for each SZA slice. The number of profiles were 66, 28 and 29 respectively for slices around 30°, 60°, 80°. Solid lines are the VER retrieved from SCIAMACHY, while dashed lines are calculated by the REPROBUS/airglow model. Only the data collected in the Southern hemisphere are presented here. The bottom panel of Fig. 22 represents the relative difference (SCIAMACHY - Model)/Model.

10    Focusing our attention to the altitude range 40-70 km where most of the emission occurs, it seems that the relative difference behaviours with altitude are identical for SZA= 80° and 60° (green and blue curves) with a peak of discrepancy at 67 km, while for SZA around 30° the peak of discrepancy is at a lower altitude ~ 58 km. In the next sub-section we compare the ozone profile calculated by REPROBUS with ozone measurements taken by GOMOS on ENVISAT during the same period of time (year 2007).



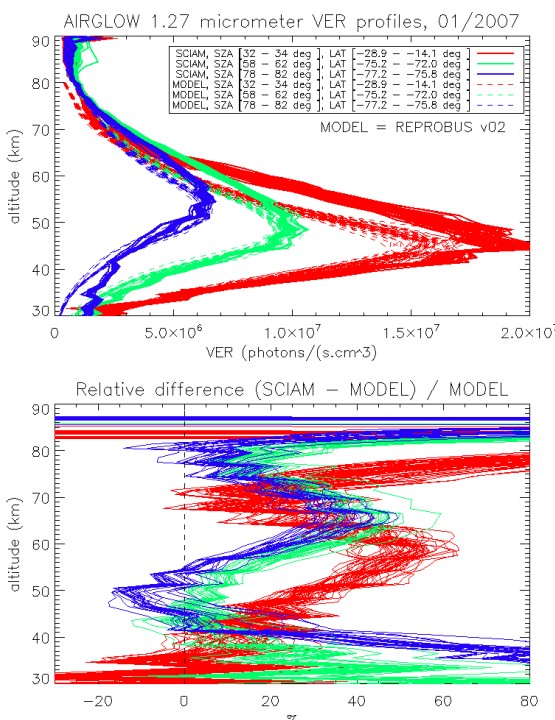

**Figure 22. Top: Comparisons of all SCIAMACHY and REPROBUS airglow VER profiles for the first 3 days of January 2007 in three SZA domains: SZA =[32 - 34 °] (red curves), SZA =[58 - 62 °] (green curves) and SZA =[78 - 82 °] (blue curves). Only profiles from the southern hemisphere were selected. Bottom: for each profile, relative**

**difference (SCIAMACHY -REPROBUS)/REPROBUS.**

### 4.3 Comparison of measured (GOMOS) and modelled (REPROBUS) ozone vertical profiles

The photo-dissociation of ozone is the main mechanism for producing $O_2(a^1\Delta)$ airglow in the atmosphere. It is

therefore important that the ozone profile calculated by the REPROBUS model is accurate. In order to verify this condition, comparisons were carried out with the ozone profiles measured by the GOMOS instrument on board ENVISAT. GOMOS (Bertaux et al., 2010) measured ozone profiles by the stellar occultation method under night and day illumination conditions. For daytime occultations, there is a contaminating signal both from the illuminated limb and from the nadir emission scattered by the GOMOS baffles. Though the processing pipelines

have been designed to correct for these contaminations, the resulting uncertainties in ozone density retrieval may be larger, depending on the geometrical conditions of the occultation and the altitude, lower altitudes having more stray light. Therefore, we are separating night and day conditions for the comparison.

### 4.3.1 Night time ozone profiles

One example of REPROBUS and GOMOS individual ozone concentration profiles under night conditions for ENVISAT orbit n° 25402 (January 8, 2007) is presented in Fig. 23. The top figure displays the ozone



concentration measured by GOMOS and predicted by REPROBUS at the same time and location. As seen on a log scale (in order to accommodate the five orders of magnitude of variation with altitude), the agreement is remarkable between GOMOS and REPROBUS in the altitude range 15-65 km. Even wiggles in the vertical profiles in the range 15-65 km are present both in the model and in the data. The bottom Fig. 23 represents the

5    relative difference (GOMOS-REPROBUS)/REPROBUS on a linear scale.

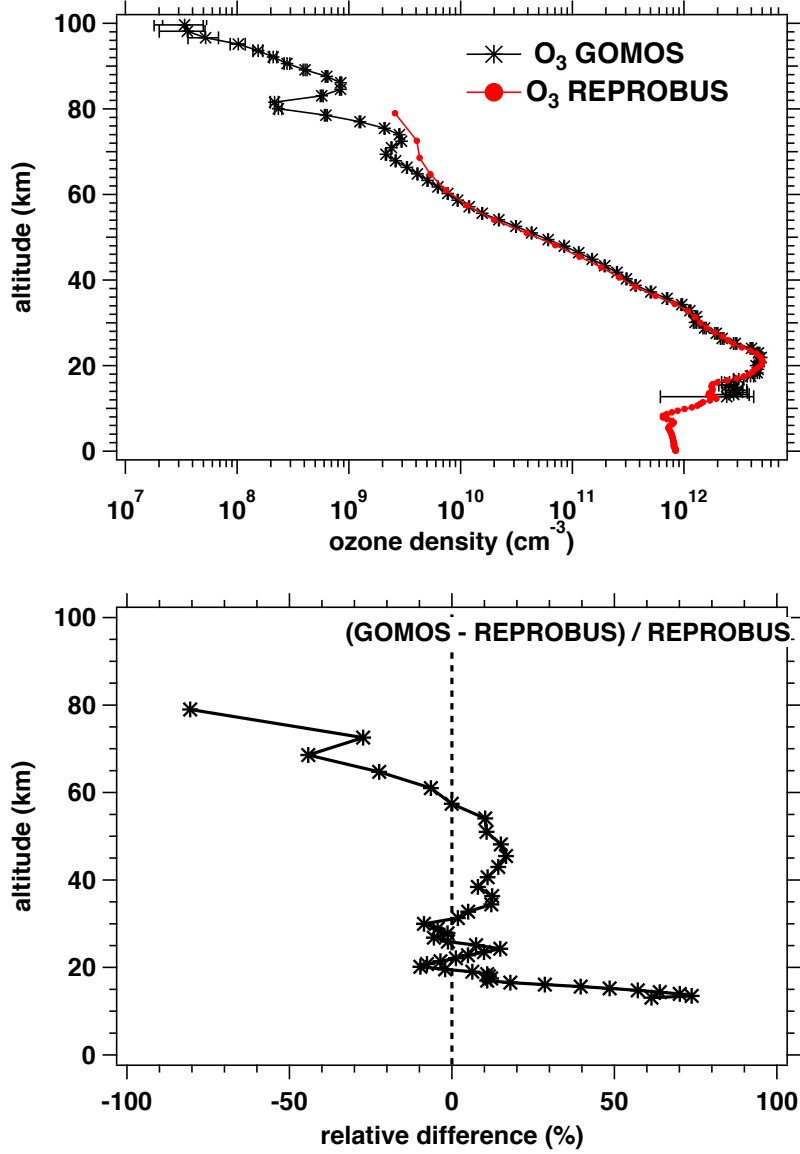

**Figure 23 Top: One typical example of comparison of a GOMOS vertical ozone profile (black curve) vs. REPROBUS prediction (red curve). The GOMOS profile was observed during the night on January 8, 2007. Bottom: relative difference (GOMOS - REPROBUS)/ REPROBUS.**


There is a significant difference above 60 km, where REPROBUS overestimates the amount of ozone relative to GOMOS. At night, GOMOS ozone profiles show a strong ozone depletion around 80 km. This is a true feature, which can be seen directly on the light of the star that increases again when the LOS passes at this altitude during the occultation of the star. This ozone "hole", explained by loss reactions with OH radicals at night, is not

reproduced by REPROBUS. The reason for this discrepancy is the assumption in the model that ozone concentration is much larger than atomic oxygen at night and thus can be set as equal to the odd oxygen family (i.e. $O_3 \simeq O_3 + O = O_x$). This approximation is justified in the stratosphere and the lower mesosphere but is wrong in the upper mesosphere, where oxygen atoms have a lifetime of the order of a day and a concentration similar to ozone during the night (e.g., Brasseur and Solomon, 2005). This shortcoming of REPROBUS will be

corrected in the next version of the model. It must be noted however, that this $O_3$ overestimation in the upper mesosphere by REPROBUS only occurs in night-time conditions.

Below 20 km the REPROBUS and GOMOS (night) curves are also diverging. Occultation measurements are less accurate below 15-20 km, because of the attenuation of the star signal, so this difference must be considered with caution. In any case, this bias is not relevant to our study since the airglow of $O_2(a^1\Delta)$ is negligible below

30 km.

We then compared all GOMOS ozone profiles in night occultation for 2007 with REPROBUS for 6 different stars, and plotted the relative difference (GOMOS-REPROBUS)/REPROBUS for the four brightest stars in Fig. 24.

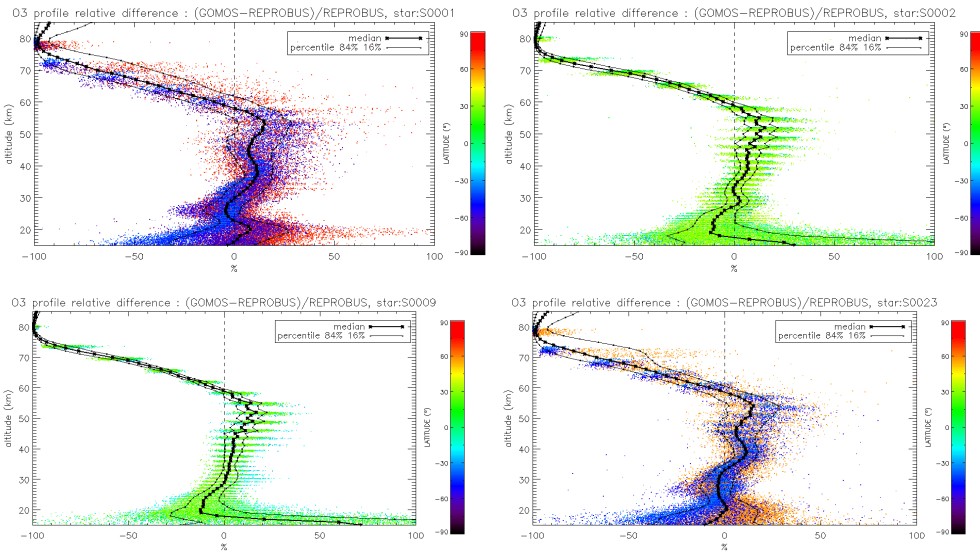

**Figure 24. Relative difference between the GOMOS and REPROBUS ozone night-time profiles for all 2007**
**occultations of the four brightest stars represented here from left to right and top to bottom: S0001, S0002, S0009, S0023. Each dot represents one GOMOS measurement of ozone. The colour of the dot is related to the latitude of each profile (see colour scale). The thick black line represents the median and the thin black lines represent the percentiles 16% and 84% of the distribution of the relative difference.**




The general trends observed on Fig. 24 are fairly similar for the six stars and in line with what is observed on the single profile of Fig. 23. The overestimation of ozone by the REPROBUS model above 60 km is confirmed. Below 60 km, there is an underestimate of ozone in REPROBUS relative to GOMOS with a maximum of about -15% at 55 km which gradually decreases to 0% at 20 km. At the location of the maximum airglow (45-50 km)

the bias is about -8 to -10%. This lack of ozone in REPROBUS is a major reason to explain why the airglow emission estimated by model is lower than observed by SCIAMACHY (Fig. 19). However, some exercises done by multiplying arbitrarily the REPROBUS $O_3$ profiles by a factor 1.2 (not shown here) show a small remaining underestimation of the airglow calculated by the model. This discrepancy certainly warrants future detailed studies that are well beyond the scope of the present paper.

**4.3.2 Day time ozone profiles**

During the day, the comparison of REPROBUS to GOMOS observations selected for their best quality (Fig. 25) shows a different behaviour for the two examples selected. With star S008 example (Fig. 25 right), there is more

ozone measured by GOMOS than predicted by the model below 60 km, as was found for the night side GOMOS ozone. On the star S005 example (Fig. 25 left), there is also a deficit (~ 20 %) of ozone in REPROBUS versus GOMOS around 60 km, but the vertical behaviour of the relative difference is quite different. However, high quality day side data are scarce with GOMOS, and definitive conclusions cannot be drawn at this stage.

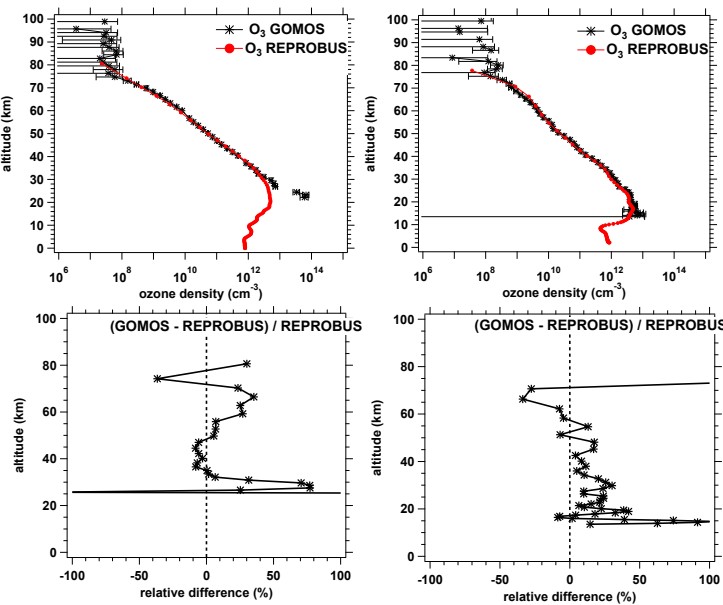

**Figure 25 Top: Two examples of comparison of a GOMOS vertical day side ozone profile (black curve) vs. REPROBUS prediction (red curve). The GOMOS profile was observed during the day on August 5, 2007. Bottom: relative difference (REPROBUS-GOMOS)/GOMOS.**

We may summarize the comparisons GOMOS/REPROBUS with the following points:

- In principle, only day side ozone is relevant for the prediction of $O_2$* airglow at 1.27 μm.



- GOMOS ozone concentration vertical profiles show quite similar values below 60 km between day and night, and quite lower values of $O_3$ at night above 60 km, a feature well understood from mesospheric chemistry.

- there is a known shortcoming of the chemistry of REPROBUS model affecting strongly night side predictions above 60 km, quite apparent with GOMOS ozone night side comparisons (too much ozone in REPROBUS).

- This night side model bias is quite negligible below 60 km, and therefore, because the $O_3$ diurnal variation is small below 60 km, the comparison GOMOS/REPROBUS on the night side showing a deficit (10-20 %) of the model versus GOMOS ozone below 60 km may be applied also to the day side. This is one reason for having model predicted $O_2$* intensities lower than observed with SCIAMACHY.

**5. The MicroCarb mission dedicated to $CO_2$ investigations**

In the domain of Earth Observations and GHG monitoring, CNES (Centre National d'Etudes Spatiales) has developed the MicroCarb mission, a space observatory dedicated to $CO_2$ monitoring. As a result of the studies conducted since 2016 by CNES concerning the use of the 1.27 μm $O_2$ band reported in the present paper, it was 15 decided to incorporate in the instrument the 1.27 μm $O_2$ band as band B4.

The MicroCarb mission builds on a high spectral resolution infrared grating spectrometer onboard a micro-satellite. The satellite platform is an enhanced version of the Myriade family. The total mass of the satellite including payload is 170 kg for a power of 100 W. MicroCarb will be launched in 2021 on an 11h30 ascending node or 13h30 descending node helio-synchronous orbit.

The MicroCarb data consists in the measurement in four spectral bands of the solar irradiance reflected by the surface and partially absorbed by atmospheric gases. Two bands are dedicated to the measurement of $CO_2$ around 1.60 μm (weak $CO_2$ B2) and 2.04 μm (strong $CO_2$ B3). Two spectral bands are dedicated to $O_2$ around 0.76 μm (strong $O_2$ B1) and 1.27 μm (weak $O_2$ B4). Figure 26 illustrates typical radiances in the four MicroCarb bands and Table 1 gives the main properties of the MicroCarb bands. The accomodation on a micro-satellite is 25 enabled by a very compact design of the instrument, having a unique telescope, one slit per band, a unique grating and a unique Sofradir Next Generation Panchromatic 1024 x 1024 pixels detector for the four bands [Pasternak et al. 2016].

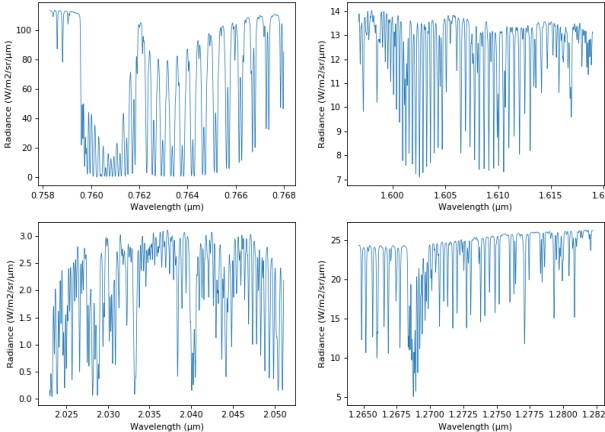

**Figure 26: Example of MicroCarb simulated spectra of the strong $O_2$ band B1 (left upper panel), the weak $CO_2$ band**



B2 (upper right panel), the strong CO$_2$ band B3 (lower left panel) and the weak O$_2$ band B4 (lower right panel).

**Table 1**

|  | B1 | B2 | B3 | B4 |
|---|---|---|---|---|
| Main species | O$_2$ | CO$_2$ | CO$_2$ | O$_2$ |
| First wavelength (nm) | 758.28 | 1596.77 | 2023.01 | 1264.63 |
| Last wavelength (nm) | 768.82 | 1618.95 | 2051.12 | 1282.19 |
| Resolution (pm) | 29.66 | 62.26 | 79.00 | 49.32 |
| Sampling (pm) | 10.34 | 21.69 | 27.53 | 17.24 |
| Reference radiance (W/(m$^2$ sr μm)) | 75.2 | 10.4 | 2.4 | 18.3 |
| SNR per channel at reference radiance | 480 | 579 | 249 | 503 |

**Spectral parameters of the four MicroCarb spectral bands**

MicroCarb provides spectra for individual footprints of 4.5 km across-track (ACT) per 8.9 km along-track (ALT). Three contiguous ACT footprints are acquired at once during the integration time of ~1.3 s. An

embedded imager, using the same telescope as the spectrometer, provides a 27 km ACT per 17.8 km ALT image centred on the three footprints for each integration time. Each image is made of 121 m x 153 m individual pixels. MicroCarb will look at nadir over lands or use a scanning mirror to get a swath up to +/- 200km. MicroCarb will look at sunglint over seas and oceans. Specific observations will be dedicated to calibration (target, sun, internal lamp, internal shutter, cold space, Moon) or probatory experiments (local mapping).

The MicroCarb ground segment will produce five levels of products: level 0 (L0) corresponding to raw telemetry, L1 to spectra calibrated for radiometry, spectrometry and geometry, L2 to dry air column-averaged CO$_2$ volume mixing ratios, L3 to space and time averages of the L2, and L4 to surface carbon fluxes.

The computation of L2 products from L1 data is a very active research field (see e.g Boesch et al. (2011), Crisp et al. (2017), Hasekamp et al. (2015), Heymann et al. (2015), Yoshida et al. (2011). The MicroCarb is

developing its own inversion tool named 4ARTIC (4AOP Radiative Transfer Inversion Tool). This tool is based on the optimal estimation described in Rodgers (2000). 4ARTIC retrieves CO$_2$ and H$_2$O on 19 vertical layers, mean and slope of albedo for each bands, surface pressure, aerosols properties, 0.76 μm fluorescence, potential instrumental parameters, and the 1.27 μm airglow emission as described hereafter.

The prior information will be provided by the ECMWF analysis for pressure, temperature and humidity, CAMS

for CO$_2$ and aerosols, PlanetObserver for the digital elevation model and Sentinel 2 images for albedo. The jacobians (partial derivatives of the spectrum with respect to geophysical variables) will be computed by the 4AOP radiative transfer code (Scott and Chedin 1981). The scattering by molecules and aerosols will be computed by a discrete ordinated scheme using LIDORT (Spurr, 2012).

A major difficulty for passive spectrometry space missions dedicated to trace gases is to handle the perturbation

of the light path by the aerosol scattering. Aerosols may increase or decrease the optical length, depending on





conditions. The available prior information about aerosols (type, density, vertical distribution, optical properties) is poor, as well as the aerosol information content in the spectrum. A specific retrieval scheme was therefore developed for 4ARTIC to handle aerosols as an equivalent distribution with a limited number of free parameters. The vertical distribution of aerosols is described as a Gaussian:

$$h(z) = A' \exp\left(-\frac{4\ln 2 (z-z_{aero})^2}{w_{aer}(z_{aero})^2}\right) \qquad (25)$$

where A' is a normalization coefficient, and the width of the Gaussian $w_{aer}$ is linked to its maximum height $Z_{aero}$ by:

$$w_{aer}(z_{aero}) = w_0 \exp\left(-\frac{4\ln 2 (z_{aero}-w_0)^2}{(2w_0)^2}\right) \qquad (26)$$

where $w_0$ equals to 4 km. This scheme is inspired from Butz et al. [2009]. The spectral dependence of Aerosol Optical Depth (AOD) is described by the Angström coefficient kaero:

$$AOD(\sigma) = AOD(\sigma_0)\left(\frac{\sigma}{\sigma_0}\right)^{kaero} \qquad (27)$$

where $\sigma$ is the wavenumber and $\sigma_0$ a wavenumber reference. 4ARTIC then retrieves three aerosol parameters at the same time as $CO_2$: the AOD at $\sigma_0$=0.76 µm, the altitude of the maximum of the gaussian ($Z_{aero}$) and the Angström coefficient (kaero). The Single Scattering Albedo (SSA) of one aerosol particle is currently fixed to 1

and the phase function is described by the Henyey Greenstein function with g currently fixed to 0.8.

One of the main purposes of the $O_2$ bands is to provide information about aerosols. Contrary to MicroCarb, most of the current $CO_2$ missions (e.g. GOSAT (Yokota et al., 2009) or OCO-2 (Crisp et al. 2017)) acquire only one $O_2$ band, at 0.76 µm. As this band is spectrally far from the $CO_2$ bands, any spectral dependence of the aerosols will disturb the evaluation of aerosol impact in the $CO_2$ bands. As an example the OCO-2 products are known to

be sensitive to aerosols, making a bias correction post-processing mandatory (O'Dell 2018).

The instrumental concept of MicroCarb gave the possibility to carry four spectral bands. The MicroCarb Mission Group chose the same 0.76, 1.60 and 2.04 µm bands as OCO-2 or GOSAT, and chose the additional 1.27 µm $O_2$ band in order to get aerosol information spectrally closer to the $CO_2$ bands, and therefore to better constrain the kaero Angström coefficient.

**6. Using a synthetic spectrum of $O_2$\*airglow emission to disentangle it from $O_2$ absorption in nadir viewing**

**6.1 Overview**

With the theoretical shape of the $O_2$* dayglow emission spectrum (Section 2) and a selected VER vertical profile (i.e., Fig. 33), it is possible to construct a synthetic spectrum of the dayglow in absolute radiance units at the native spectral resolution of LBLRTM (several $10^5$), which may be degraded to any spectral resolution to simulate various instruments. For a given surface albedo and SZA, the radiance of scattered solar radiation by



the surface modified by $O_2$ absorption may also be computed, using LBLRTM. Adding both spectra is a simulation of what will be seen by a nadir-viewing instrument (in the case of no aerosols) in the region around 1.27 μm, as shown on Fig. 27 for two cases with different albedos and SZA, where the two spectra are separated.

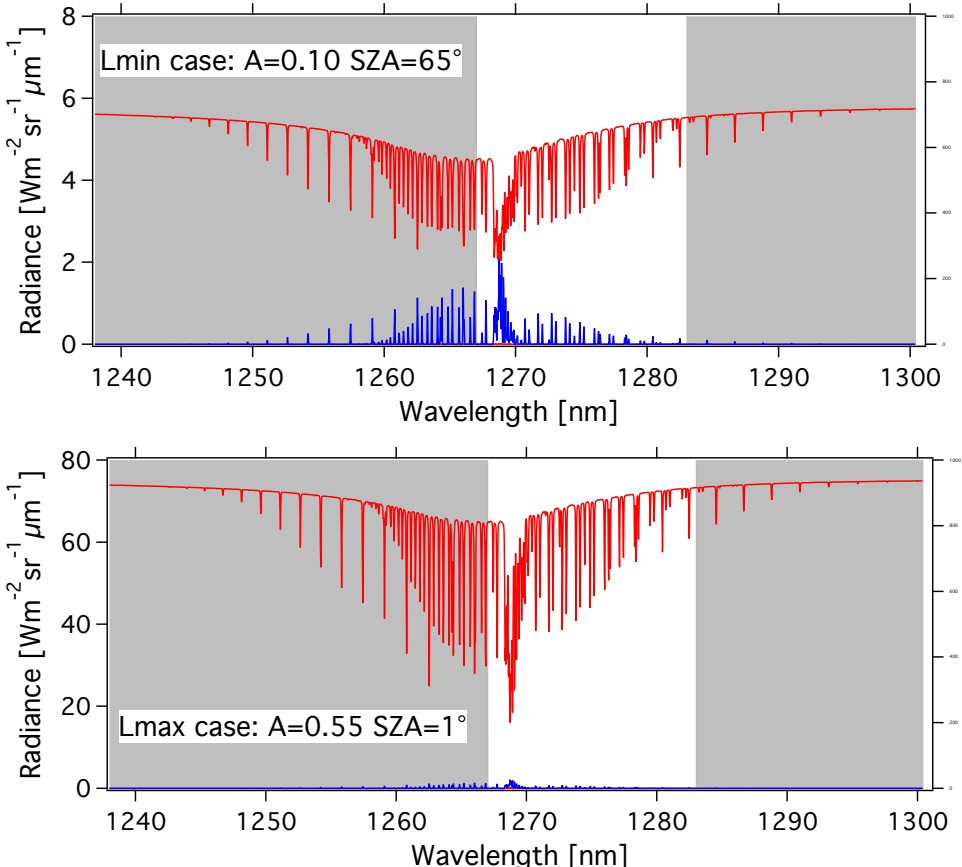

**Figure 27.** Airglow (blue) and scattered sunlight (red) absolute intensities and radiances (in W m$^{-2}$ sr$^{-1}$ μm$^{-1}$) for two cases: Top: low albedo A=0.1 and high SZA= 65° and bottom: high albedo A=0.55 and almost sub solar SZA= 1°. The depression on the continuum of reflected sunlight is due to the $O_2$ CIA (Collision Induced Absorption). Water vapour absorption lines are present on the right part of the spectra. The white area corresponds to the MicroCarb wavelength coverage.

While the $O_2$* emission spectrum (blue) is quite similar to the $O_2$ absorption spectrum imprinted on the albedo (red on Fig. 27, top), there are five factors which make them different (as noted before), allowing their disentangling by spectral profile fitting:

- the transmission $Tr(\tau)=\exp(-\tau)$ is not linear when $\tau > 1$ while the emission stays linear.

- airglow lines are narrower than absorption lines (no pressure broadening at high altitude).

- the CIA (Collision Induced Absorption) affects only the $O_2$ absorption spectrum.

- as shown in Section 2 the ratio of $O_2$* emission to $O_2$ absorption is a continuous function of the wavenumber $\nu$ (in cm$^{-1}$).



- the temperatures near surface and in the mesosphere are different, dictating different populations of rotational levels.

Basically a retrieval software tool allowing to determine from an observed spectrum simultaneously the albedo, the $O_2$ vertical column (or surface pressure $P_{surf}$, when the water vapour is ignored), and the $O_2$* airglow

intensity consists of two parts: a forward model to simulate what would be observed (depending on the parameters to be retrieved), and a scheme to minimize the chi$^2$ of the fit of the observed spectrum by the simulated spectrum (Levenberg-Marquardt).

For the present study we have developed the LATMOS breadboard, a software (in IGOR language) dedicated first to proof of concept for the use of the $O_2$ band at 1.27 µm in presence of $O_2$* airglow contamination, and

then applied to SCIAMACHY nadir observations, showing that when the albedo is weak, the $O_2$*airglow intensity may be identified and its intensity actually measured, as a scaling factor of a synthetic airglow spectrum (Sub-section 6.2). We have also used the 4ARCTIC software to evaluate the performances of the particular Microcarb instrument configuration (wavelength coverage and spectral resolution) and specified SNR as a function of spectral radiance of the nadir-viewing scenery (Sub-section 6.3).

### 6.2. Airglow inversion in nadir SCIAMACHY spectra

The possibility to extract 1.27 µm airglow information from nadir SCIAMACHY spectra is limited by the relatively low spectral resolution of this instrument (about 860). The intensity of the sunlight reflected by the

surface and the atmosphere is in general much larger than the airglow in this spectral band. This is true above continents but above ocean, where the near infrared albedo is very low, it should be possible to extract the airglow when the sky is clear. We tested this possibility using the spectral inversion LATMOS breadboard, originally developed to test the possibility to determine the airglow intensity in the 1.27 µm $O_2$ band in the MicroCarb $CO_2$ mission.

### 6.2.1 LATMOS inversion breadboard

As said above, the spectrum observed by SCIAMACHY in the 1.27 µm band at nadir is the sum of the nadir solar flux reflected by the ground and the atmosphere, partly re-absorbed by atmospheric $O_2$, and the airglow

$O_2$* emission spectrum. For clear sky conditions if we neglect the reflection by the atmosphere, the reflected solar spectrum may be expressed as:

$$I_{Nadir}(\lambda, sza) = A.F(\lambda).\frac{T_{atm}^{\left(1+\frac{1}{\cos(sza)}\right)}\cos(sza)}{\pi} \qquad (28)$$

where:

- $A$: albedo (assuming a Lambert law (isotropic) reflectance)
     - $\lambda$: wavelength
     - $F(\lambda)$: Solar spectrum outside atmosphere
     - $SZA$: solar zenith angle
     - $T_{atm}$: one way vertical atmospheric transmission



The airglow spectrum is assumed to be proportional to the logarithm of the atmospheric $O_2$ transmission at high altitude multiplied by the emission to absorption ratio $\varepsilon(\lambda)/SS(\lambda)$ as explained in sub-section 2.3. The relative intensity of the lines depends mainly on temperature. To take into account this dependency, we represent the airglow spectrum $Ag((\lambda)$ as a linear combination of a warm spectrum $Ag_{warm}(\lambda)$ and a cold spectrum $Ag_{cold}(\lambda)$.

These warm and cold spectra are computed using US standard atmosphere transmission tables from LBLRTM for nadir viewing around 50 km where the temperature is warmer (270 K) and 70 km where the temperature is colder (217 K).:

$$Ag_{warm}(\lambda) \approx C_{norm}.[Ln(T_{51km}(\lambda))-Ln(T_{49\,km}(\lambda)].[\varepsilon(\lambda)/SS(\lambda)] \qquad (29)$$

$$Ag_{cold}(\lambda) \approx C_{norm}.[Ln(T_{71km}(\lambda))-Ln(T_{69\,km}(\lambda)].[\varepsilon(\lambda)/SS(\lambda)] \qquad (30)$$

where $T_z(\lambda)$ is the transmission at wavelength $\lambda$ from altitude $z$ to the top of the atmosphere and $C_{norm}$ a normalisation constant determined in order that the integral of the spectrum is equal to the integral of a reference spectrum., and using equations (17) and (14).

A Levenberg-Marquardt (L-M) method is used to determine the parameters giving the best fit to SCHIAMACHY spectra. The total column of $O_2$, assimilated to surface pressure, the airglow, the $H_2O$ column and the albedo are inverted using the L-M converging scheme. As atmospheric transmission depends non-linearly on the $O_2$ column, its Jacobian is calculated at ground level from the difference in transmission between the ground and 1 km altitude.

The measured spectrum will therefore be expressed as:

$$F(\lambda) = K_1.\left[(1-K_2).I_{Nadir_{0km}}(\lambda) + K_2.I_{Nadir_{1km}}(\lambda)\right].T_{H2O}^{K5} \quad + K_3.Ag_{warm}(\lambda) + K_4.Ag_{cold}(\lambda) \quad (31)$$

where $K_1$ is the intensity of the reflected spectrum (albedo); $K_2$ is the sensitivity of the reflected spectrum to surface pressure; $K_3$ is the warm component of the airglow spectrum; $K_4$ is the cold component of the airglow spectru and $K_5$ is the ratio $H_2O$ column / reference column. The coefficients $K_1, K_2, K_3, K_4, K_5$ can be imposed or left free in the L-M inversion. The measurement uncertainty in each spectel is assumed to be proportional to the square root of the signal.

**6.2.2 Application to SCHIAMACHY nadir viewing observations: retrieval of $O_2$* airglow intensity at 1.27 μm**

The inversion scheme was applied to 3 days of SCHIAMACHY nadir data above ocean (1-3 April 2007). For the
L-M inversion, SCHIAMACHY nadir characteristics are taken from OSCAR https://www.wmo-sat.info/oscar/satellites

- Band 971-1773 nm
- Spectral resolution 1.48 nm (resolution 858 @ 1.27 μm)
- SNR 1500 @ 25 W m$^{-2}$ sr$^{-1}$ μm$^{-1}$



The SNR is probably optimistic but it does not matter for the present study. The uncertainty is not calculated from the SNR but evaluated from the dispersion of the results.

Figure 28 shows two examples of similar SZA spectra (35 ° -36 °), one with a high radiance over a thick cloud cover and one with a low radiance on a clear day. In the case of high-reflected radiance (Fig. 28, top), the 1.27 μm band is dominated by $O_2$ absorption, with the airglow filling only slightly the bottom of the lines. In the case of low reflected flux (Fig. 28, bottom), the band is dominated by the airglow.

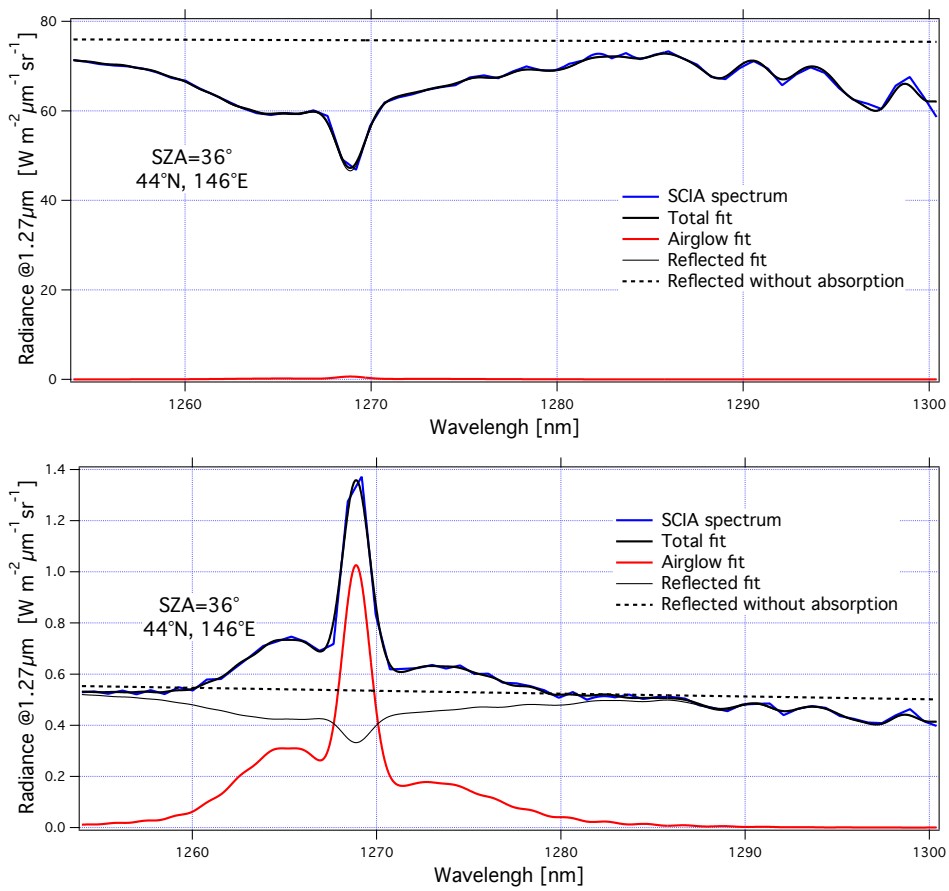

**Figure 28: Two examples of SCIAMACHY spectra at nadir; (top) above thick clouds and (bottom) above clear ocean. SCIAMACHY spectrum is in blue, fitted spectrum in thick black, determined airglow spectrum in red, fitted reflected spectrum in light black, reflected spectrum without absorption in dotted black. The word "radiance" in the label axis means the spectral radiance or intensity.**

In Fig. 29, it can be seen that when the reflected solar radiance is small, the values of the inverted airglow intensity are little dispersed. On the contrary, a very high dispersion is observed with a high-reflected radiance. It is concluded that airglow inversion is only possible at low reflected solar radiance, corresponding to situations above the ocean on a clear day or at a high SZA. For the rest of this study of SCIAMACHY nadir observations, we will limit the analysis to spectra with a reflected radiance lower than 5 W.m$^{-2}$.sr$^{-1}$ μm$^{-1}$.



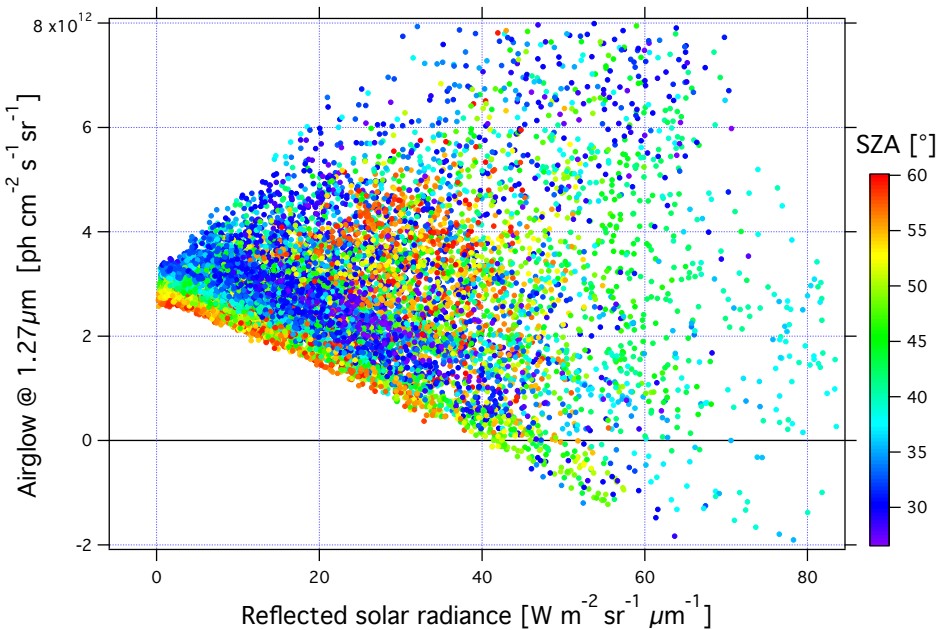

**Figure 29: Intensity of the inverted airglow as a function of the reflected solar spectral radiance (in units of W m$^{-2}$ sr$^{-1}$ μm$^{-}$). The points are coloured according to the SZA from blue to red from 27 ° to 60 °.**

In order to validate the airglow values inverted using nadir observations, we compare them to the values inverted using limb observations (Fig. 30). The latter are obtained by vertical inversion of the airglow profile observed at limb and integrated over the vertical column as described in Sub-section 3.4. The good agreement observed between the two methods gives us confidence in the results obtained with nadir observations. On the other hand, when the data are compared to our model there is an underestimation of the nadir intensity of the airglow

simulated by REPROBUS v02 of about 10-15%. This underestimation had already been found for limb comparisons. The inverted airglow follows the same SZA dependence as that simulated by REPROBUS. It can be noticed that for SZA> 90 °, the airglow values at nadir are divided into two families of different intensity. Figure 31 shows these values with distinction between morning and evening data. The values higher in the evening than in the morning are due to the long lifetime of $O_2(a^1\Delta g)$, greater than 1 hour in the absence of

quenching in the high mesosphere. REPROBUS cannot reproduce this morning-evening difference, the concentration of $O_2(a^1\Delta)$ being calculated by assuming the photochemical equilibrium.

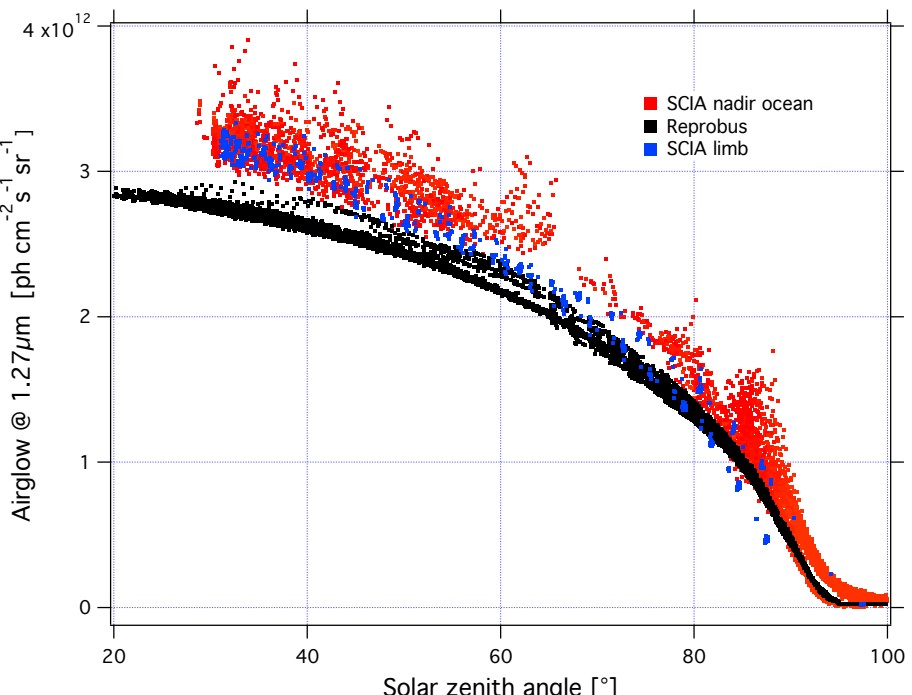

**Figure 30: Airglow intensity at nadir according to the SZA for the month of April 2007: in red SCIAMACHY measurements at nadir with a reflected solar spectral radiance <5 W.m$^{-2}$.sr$^{-1}$ μm$^{-1}$, in blue nadir intensities retrieved from SCIAMACHY measurements at the limb, in black simulations with REPROBUS v02. The airglow intensity at its maximum reaches 3.2x10$^{12}$ photons cm$^{-2}$ s$^{-1}$ sr$^{-1}$ which corresponds to 5 mW m$^{-2}$ sr$^{-1}$ (spread over a few nanometers of wavelength), to be compared to a solar backscattered luminance (radiance) of ~ 5 to 70 Watt m$^{-2}$ sr$^{-1}$ μm$^{-1}$ (see Fig. 27).**



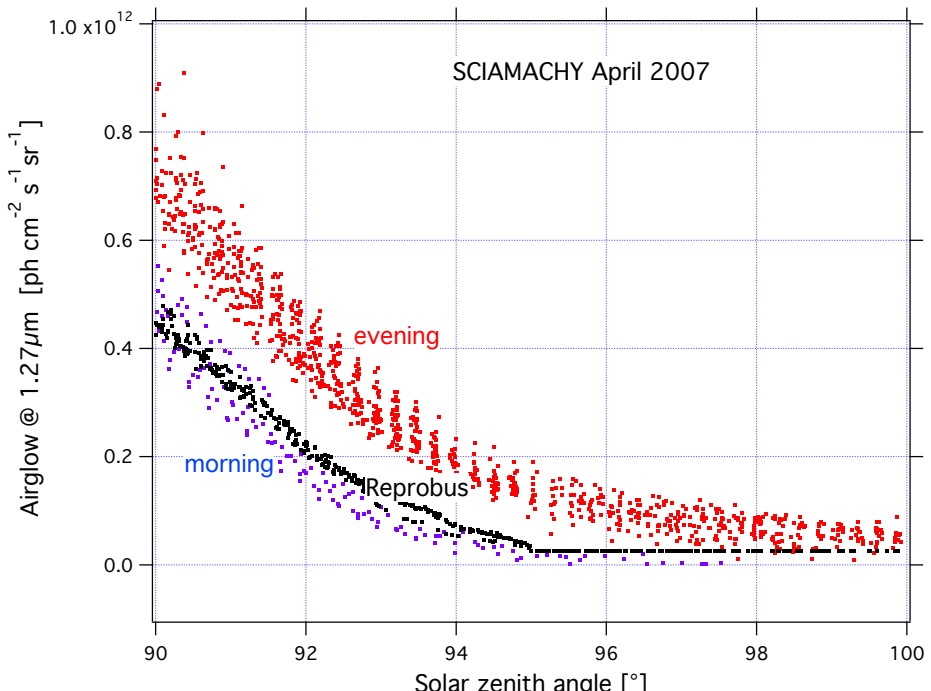

**Figure 31: Airglow nadir intensity versus SZA for SZA> 90°; SCIAMACHY measurements in red during evening and in blue during morning; REPROBUS v02 simulations in black. The observed difference between morning and evening values is due to the long lifetime of $O_2(a^1\Delta g)$.**

In order to evaluate the underestimation of the airglow by REPROBUS, a linear regression is performed between the SCIAMACHY measurements with nadir and the nearest REPROBUS values in time and position (Fig. 32). The slope of the regression is 1.13. SCIAMACHY therefore sees on average 13% more airglow than estimated with REPROBUS. The regression line does not go through the origin but to $2\times10^{11}$ photon $cm^{-2}s^{-1}sr^{-1}$. This can

10     be attributed to not taking into account in REPROBUS the airglow above 80 km that is present both day and night.



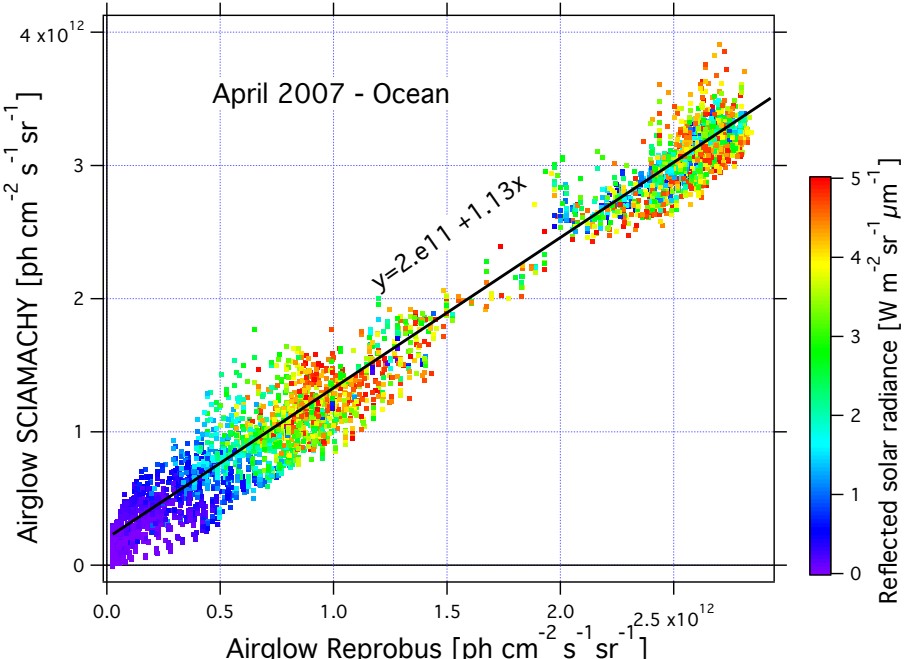

**Figure 32: Nadir Airglow SCHIAMACHY observed intensities versus REPROBUS v02 simulation (same units on both axes, photons cm$^{-2}$ s$^{-1}$ sr$_{-1}$). The dots are coloured according to the reflected solar radiance from 0 in blue to 5 W m$^{-2}$.sr$^{-1}$ in red. Linear regression of the correlation is in black. Airglow intensity values are expressed in ph.cm$^{-2}$.s$^{-1}$.sr$^{-1}$.**

We have demonstrated that, despite the moderate spectral resolution of SCIAMACHY (~860 against 25,000 for the future MicroCarb mission), it is possible to extract the $O_2$ airglow at 1.27 μm in nadir spectra provided that the spectra are selected above the sea with a low reflected solar flux (clear sky conditions). The inverted airglow is on average 13% higher than that simulated by REPROBUS v02, in agreement with REPROBUS - SCIAMACHY comparisons at limb (section 4.2.1). The inverted airglow follows the same SZA dependency as that simulated by REPROBUS except at twilight where the morning-evening difference is not reproduced by REPROBUS which assumes photochemical equilibrium. Sun et al. (2018) had concluded that it is not possible to extract nadir airglow from SCIAMACHY measurements. We have shown on the contrary that this is possible if we select the low flux spectra reflected over the ocean in clear weather.

With MicroCarb data and its higher spectra resolution, one will be able around coastal zones to compare the $O_2$* airglow intensity measured above the sea and above the ground, just nearby. They should be very similar, because the characteristics of spatial lengths of intensity variations are large (according to REPROBUS model). This comparison would provide an important "sanity check" of the retrieval of Psurf (or $O_2$ column).

### 6.3 Surface pressure retrieval on simulated nadir spectra contaminated by $O_2$* airglow

We performed the inversion of nadir simulated spectra with the 4ARTIC v4.2 software in order to have an estimation of the performance of the $O_2$ 1.27 μm band on the retrieval of the surface pressure "Psurf" when the





spectra are contaminated by the $O_2$* airglow signature.

In order to build the synthetic spectra simulating the data to be fitted in our inversions we followed the following three steps:

1. Computation of a very high resolution "reflected" spectrum in the B4 Microcarb band (1.27 μm), without noise, by calling the radiative transfer model 4AOP. This spectrum is then degraded at the resolution of the Microcarb instrument (resolution power = 25000) and resampled on the Microcarb wavelength grid.

2. Adding an airglow spectrum as seen from TOA with nadir view. This spectrum is assumed to be proportional to the logarithm of the atmospheric $O_2$ transmission at high altitude multiplied by the emission to absorption ratio $\varepsilon(\lambda)/SS(\lambda)$ as explained in section 2.3.7. In the frame of this study, we developed a software tool for building such a spectrum.

3. Finally generate 1000 noisy spectra by adding a randomly generated Gaussian noise with amplitude based on the Microcarb SNR.

We tested two inversion methods:

- Method #1: Simultaneous inversion of airglow and Psurf.
- Method #2: Inversion of only Psurf using a spectel mask, eliminating from the fit the most contaminated spectels (as recommended by Sioris 2003 for the $O_2$ A band).

For both methods the performance of estimation of Psurf is based on a Monte-Carlo approach with the inversion of 1000 noisy spectra. The two statistical performance estimators are the Psurf random error which is calculated as the standard deviation of the 1000 retrieved Psurf values and the Psurf bias which is calculated as the difference between the Psurf true value (1013 hPa) and the average of the 1000 retrieved Psurf values.

The inversion scheme used by 4ARTIC is based on the Optimal Estimation Method (OEM) described by Rodgers (2000) which uses a Bayesian approach (use of a-priori information to constrain the inversion). The elements of the state vector are: Psurf, mean Albedo, slope on albedo, dry air mixing ratios $X_{CO2}$, $X_{H2O}$ and (only for method #1) the airglow scaling factor(s).

Both methods are described below with their associated results of the Psurf performance estimators. We remind that the MicroCarb requirements on the Psurf retrieval for a Lmoy luminance scenario are 0.1 hPa in term of bias and 1 hPa in term of random error. Lmoy corresponds to an observation with SZA=36° and albedo at 1.27μm = 0.2. For both methods, these values have been used for computing the reflected spectrum with 4AOP (see step 1 above). Only a clean atmosphere scenario, i.e without aerosol, was tested for both methods.

Method #1:

In the first method, we try to invert the $O_2$* airglow at the same time as Psurf (and the other state vector elements). We tested three different approaches concerning the inversion of the airglow:

1. The shape of the airglow spectrum is considered to be perfectly known. In the state vector, we invert an 'airglow scaling factor' which associated jacobian is the airglow spectrum that we put in the simulated data. The starting value for the scaling factor is equal to 0 and the true value which is expected to be retrieved is 1.



2.  The shape of the airglow spectrum is not considered to be perfectly known. We still use a single 'airglow scaling factor' but its associated jacobian spectrum has a slightly different shape than the spectrum that we put in the simulated data. We took for the jacobian spectrum the airglow spectrum obtained with the REPROBUS VER profile in colocation with the SCIAMACHY profile used to build the airglow spectrum put in the simulated data. This is illustrated in Fig. 33 and 34. Thus the error done on the shape of the airglow spectrum is representative of the error that the REPROBUS model does on the computation of an airglow VER profile.

3.  The shape of the airglow spectrum is considered not perfectly known. However we try to "approach" it as much as possible by inverting a linear combination of a cold airglow and a warm airglow (different mesospheric temperatures), both having slightly different shapes. A cold airglow spectrum is built with our tool by using a SCIAMACHY VER profile at SZA=85° (which peak is around 60 km) and a warm spectrum by using a SCIAMACHY VER profile at SZA=36° (peak near 45 km). These two spectra are then normalized at the intensity of the airglow spectrum that is put inside the simulated spectrum to invert. The cold and warm "normalized" spectra are used in the inverse model as jacobians of two elements of the state vector, respectively a cold and a warm airglow scaling factor.

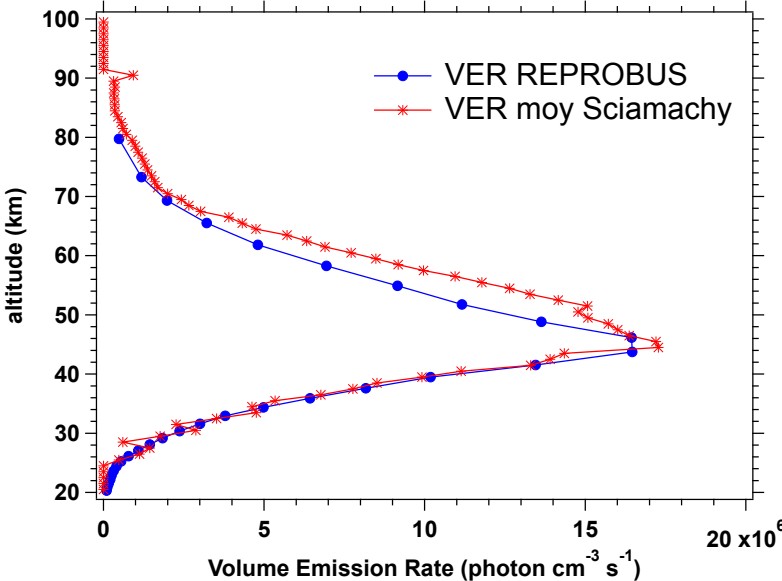

**Figure 33: (In red)** VER SCIAMACHY profile used to build the synthetic airglow spectrum put in the simulated spectrum to be inverted. **(In blue)** associated (co-located) REPROBUS VER profile, showing the underestimation of REPROBUS airglow modelling over 50 km. The airglow synthetic spectra associated to these two VER profiles are illustrated on Fig. 34.



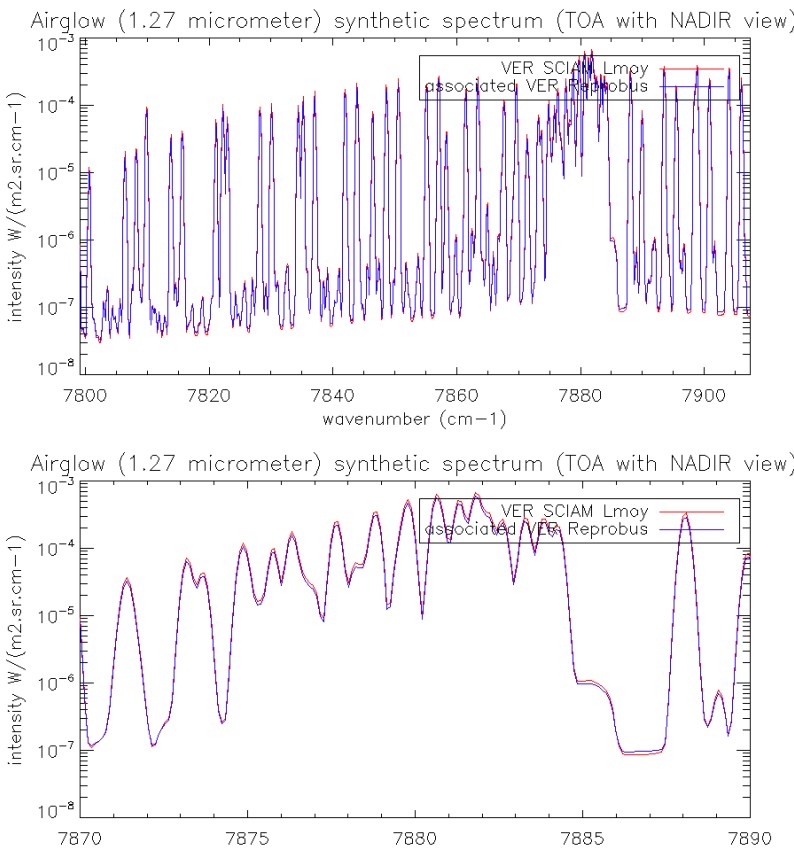

**Figure 34: (Upper plot) synthetic airglow spectra corresponding to the (red) SCIAMACHY VER profile of Fig. 33, (blue) associated REPROBUS VER profiles of Fig. 33. For the 3 tests, the red spectrum is put in the simulated data to invert. For the test #1 it is also used as the jacobian of the inverse model. For the test #2, the blue spectrum is used as jacobian of the inverse model (blue). The jacobian spectra of test #3 are not represented here. (Bottom plot) zoom on the region of the Q branch.**

The results of the Psurf inversion by method #1 for the three approaches, using only the 1.27 μm band, are presented in Table 2.

**Table 2**

| Test | description | Psurf | |
|------|-------------|-------|---|
| | | **Bias (hPa)** | **Random Error (hPa)** |
| #1 | shape perfectly known, 1 airglow parameter inverted | -0.002 | 0.88 |
| #2 | shape not perfectly known. Error on the shape, 1 airglow parameter inverted. | **0.26** | 0.88 |
| #3 | shape not perfectly known. cold + warm airglow parameters. | **-0.11** | 0.88 |

**Results of the Psurf inversion by method #1 (airglow also inverted). Values printed in bold numbers are non compliant with Microcarb requirement. Non-bold means compliant.**



The results presented in Table 1 show that the Microcarb B4 band allows retrieving Psurf with a random error of 0.88 hPa which is compliant with the Microcarb requirement (1 hPa). Adding a second element of airglow does not seem to increase the Psurf random error. When considering that the shape of the airglow spectrum is perfectly known (test #1), the bias on Psurf is completely negligible. However in true conditions, this will not be the case since the shape of the airglow spectrum is dependant of not perfectly determined variables like the temperature profile or the airglow VER profile. When forcing the shape of the spectrum with an error representative of the error done by REPROBUS model on the determination of the airglow VER profile (Test #2), we obtain a significant bias of 0.26 hPa. However by letting the inversion process adjusting the shape of the airglow spectrum, i.e using a linear combination of a cold and warm airglow (test #3), the Psurf bias is reduced to -0.11 hPa which is very close to the Microcarb requirement (0.1 hPa).

Method #2:

In the second method the airglow is not inverted. Instead, we used a mask of spectel in order to discard the spectels which are the most contaminated by the airglow emission. Indeed, since the emission lines are thinner than the absorption lines and centred on the same wavelength, we can apply a narrow mask on the central part of each absorption line which will remove most of the airglow emission lines while keeping the wings of the absorption lines as well as the CIA continuum, both bringing useful information to retrieve Psurf. As we consider that the airglow spectrum can be a-priori estimated (by a model) and corrected with an accuracy of about 90%, the airglow spectrum which is put in the simulated spectrum to be inverted is only 10 % of the intensity of the airglow spectrum used in method #1.

We tested three pixel masks which remove all spectels which intensity is higher than respectively 10%, 1% and 0.1% of the maximum intensity of the airglow spectrum (located at 7882 cm$^{-1}$). This is illustrated in Fig. 35.

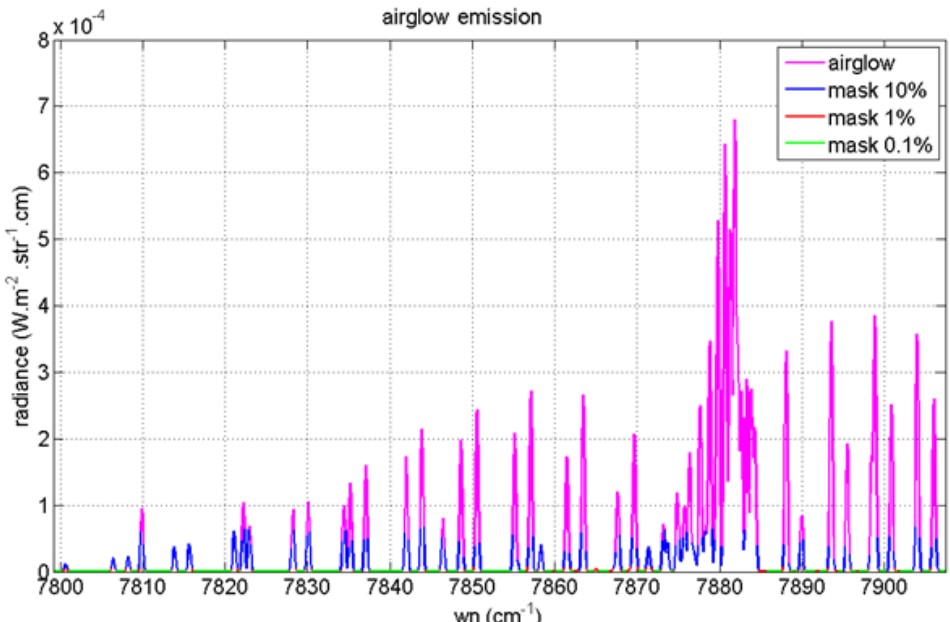

**Figure 35: Airglow spectra at 1.27 µm, at Microcarb resolution, used in our tests of inversion of Psurf with method #2 for several value of the spectel mask: no mask (pink), 10% mask (blue), 1% mask (red), 0.1% mask (green). The three masks remove respectively 14%, 31% and 44% of the spectels in the B4 band.**

The results of the Psurf inversion with the 3 masks, using only the 1.27 µm band are presented in Table 3.

**Table 3**

| Mask | Bias (hPa) | Random Error (hPa) |
|---|---|---|
| #1 (14% spectels discarded) | **-2.78** | 0.97 |
| #2 (31% spectels discarded) | **-0.13** | **1.38** |
| #3 (44% spectels discarded) | -0.04 | **1.52** |

**Results of the inversion of Psurf in Microcarb band B4 (1.27 µm) with method #2 (spectel mask). Values printed in bold numbers are non compliant with Microcarb requirement. Non-bold means compliant.**

The spectels contaminated by airglow are a source of bias on the Psurf retrieval. The more contaminated spectels we discard with a mask, the more this bias is expected to be reduced. However, at the same time, the more spectel we discard, the more we increase the random error since we lose some useful information on Psurf in the absorption bands. Thus we are looking for a mask which provides a good compromise between bias and random error. Table 3 shows however that the spectel mask method does not allow meeting the Microcarb requirement simultaneously in term of bias and random error using the Microcarb B4 band only. Values are however not that far from Microcarb requirements and mask number #3 for example allows being in spec in term of bias (-0.04 hPa) while keeping a reasonable random error of 1.52 hPa. This method can thus be foreseen as a good alternative to the method #1 and is worth being tested on Microcarb data when available.



Therefore, we have demonstrated with the above simulations that with the spectral resolution of Microcarb (25000), and only using the $O_2$ IR band B4 at 1.27 μm, we may retrieve Psurf (or similarly the $O_2$ vertical column of dry air) with an accuracy almost compliant with the Microcarb requirements. Other tests were also made with 4ARCTIC, by using simultaneously both $O_2$ bands B1 (0.76 μm) and B4 (1.27 μm) of Microcarb. Of

course, it improves very much the retrieval accuracy, but since we are investigating the use of the B4 band because band B1 is suspected to present some problems, we do not show the results here. It will be particularly interesting to investigate the improvement of using the B4 band, in addition to the B1, in the presence of aerosols. This is way beyond the scope of the present study. However, the capability to disentangle the spectrum of the $O_2^*$ emission from the $O_2$ IR absorption with their fine structure should not depend very much on the

presence of aerosols, in view of their slowly varying spectral signature with wavelength.

**7. Contamination of absorption measurements by airglow emissions**

The contamination of the absorption $O_2$ band at 1.27 μm contaminated by the corresponding $O_2^*$ airglow

emission is an extreme case, in view of the strong intensity of the airglow (up to 40 megaRayleigh). However, this particular band is not a unique case. In all nadir viewing observations from space analysing scattered sunlight, the LOS goes across all the airglow layers (Fig. 1) and their emissions are superimposed on the scattered sunlight spectrum. We list below a few commonly used absorption bands and their potential contamination by airglow.

Zarboo et al. (2018) have retrieved form SCIAMACHY limb observations the VER of the emission by the $O_2$ molecule excited in the $^1\Sigma$ state when it decays to the fundamental in the $O_2$ band A at 0.76 μm (their figure 5). One could estimate the nadir-viewing radiance from the total column. This is a contribution to radiance measurements in the A band which (to our knowledge) has been ignored up to now, that should be subtracted systematically from any nadir A band measurements, with its detailed spectral shape, before analysis. Sioris

(2003) computed a synthetic spectrum of the A band $O_2$ emission, including resonance scattering of solar photons. Then he performed a column $O_2$ retrieval, with and without the airglow contribution. In spite of the fact that the airglow may account for more than 10% of the total radiance in the core of strong lines, the bias (negative) when neglecting to subtract the A band airglow was found to be only 0.0061 % on the $O_2$ column retrieval. An independent study performed in the frame of the present work found -0.0073 %, confirming the

Sioris estimate. However, both studies have assumed a total absence of this emission below 50 km, while the VER profile derived from SCIAMACHY observations suggest a fast increase of the emission with decreasing altitude around 50 km (Figure 5 of Zarboo et al. 2018). Non-accounting for this emission below 50 km will be more detrimental (for the accuracy of Psurf retrieval) in low albedo regions and low SZA values (resonance scattering penetrates more deeply at low SZA).

**The $CO_2$ molecule** illuminated by the sun is subject to resonance fluorescence, with non-ETL emission in the bands where the absorption is measured (weak or strong $CO_2$ bands). This is to our knowledge not yet accounted for in GHG retrievals. In addition, there is a (0,1) transition at 1.58 μm of the $O_2$ ($^1\Delta$) around 1.58 μm that interfers with a pair of $CO_2$ weak bands. Though this emission is about 50 times smaller than the 1.27 μm emission, it is preferable to use the other pair of $CO_2$ band at 1.60 μm, as do OCO-2 and MicroCarb instruments.

**The $CH_4$ molecule:** Lopez-Puertas et al. (2005) reported limb observations of non-LTE emission of $CH_4$ in the



thermal IR at 7.6 μm from MIPAS/ENVISAT FTS instrument. It began to be noticeable at 45 km of altitude, and represented up to 60% of the emission at 70 km. Therefore, this non-LTE emission triggered by solar radiation forms an airglow layer that is intercepted in a nadir viewing geometry. Such a known-to-exist contribution should be subtracted systematically from nadir-viewing measurements (i.e., IASI measurements). This

contribution has been ignored by de Wachter et al. (2017), analyzing nadir IASI methane measurements. A rough estimate from figures of Lopez-Puertas et al. (2005) indicates a non-LTE horizontal emission of 100 nw/cm$^2$/sr, over 4 cm$^{-1}$ at 50 km altitude. A vertical viewing would reduce this value by a Chapman factor of several tens, probably below the noise of IASI measurements ( ~10 nw/(cm$^2$ sr cm$^{-1}$). However, it should be noted that there is also certainly some non-LTE emission below 50 km (difficult to see for MIPAS), and that this

spectrum has a very similar shape as the LTE emission. Therefore, the non-LTE emission should be modelled, scaled to MIPAS determinations at high altitudes, and subtracted blindly form nadir viewing (IASI-type measurements). It is possible that other bands of $CH_4$ used for GHG retrievals may be similarly affected.

The fluorescence of **$H_2O$ molecule** excited by solar radiation has been observed at 2.67 μm in the coma of several comets (i.e., Bockelée-Morvan et al. 2015) and should be present in the atmosphere of the Earth, as well

as in the other lines which are present in GHG bands (i.e. at 2.0 μm). These $H_2O$ airglow emissions superimposed to surface radiation have not been subtracted from nadir observations or their intensity even estimated, to the best of our knowledge.

The fluorescence of **CO molecule** excited by solar radiation has been observed at 4.53 μm in the upper atmosphere of Venus with high spectral resolution (R~43,000) by Marcq et al. (2015). The same must happen in

the atmosphere of the Earth, introducing a bias of about 1% on the CO column retrieval (rough estimate), may be not so important because well below the currently achieved accuracies accuracy of some tens of % on CO columns.

## 8. Conclusions

In this paper we have reported the results of a three years (2016-2018) scientific research effort to revisit the use of the $O_2$ absorption band at 1.27 μm in the problem of GHG retrieval from space observations of the detailed spectrum of the solar radiation scattered by aerosols, atmosphere and surface. It is widely recognized that this 1.27 μm band being nearer in wavelength to the $CO_2$ bands at 1.6 μm and 2.0 μm, it would be less uncertain to

"transport" aerosols optical properties (wavelength dependent) from this $O_2$ band to the $CO_2$ bands, than from the $O_2$ A band at 0.76 μm. However, the $O_2$ absorption band at 1.27 μm is contaminated by a strong dayglow emission, presenting a similar spectral structure. This airglow is mainly due to photolysis of ozone in the mesosphere, letting an $O_2$ molecule in an excited state $O_2^*$ which spontaneously de-excite.

### 8.1 Nadir $O_2^*$ airglow Intensity

As a first approach, we needed to have an idea of the absolute amount of $O_2^*$ airglow contamination, in order to compare to the expected radiances in nadir viewing which are well documented. We could not use the SCIAMACHY nadir observations around 1.27 μm for this purpose, since they are contaminated (!) by solar

scattered radiation from the surface. Therefore, we used the limb-viewing observations of SCIAMACHY in this

band, which are not contaminated when looking above ~30 km of altitude. We have designed a method to retrieve the vertical profile of the $O_2$* VER from a limb scan of SCIAMACHY, taking into account the re-absorption of $O_2$ along each LOS. A "fictitious" nadir-viewing intensity from SCIAMACHY could therefore be derived by vertical integration of the VER. When extracting the data for various periods of the year, it was found

that the major factor governing this $O_2$* airglow intensity was the Solar Zenith Angle, with little variability otherwise.

In parallel, we have conducted a major effort to model the intensity of the $O_2$* airglow emission, as an off-line extension of the CTM REPROBUS model providing the ozone field within the ECMWF meteorological field. We found the same overall behaviour with SZA and some weak seasonal dependence. A systematic comparison

of 2007 SCIAMACHY 12,833 limb-scans and corresponding fictitious nadir-viewing intensities with the prediction of our model (and also VER vertical profiles) indicates an overall good agreement, with though a deficit of about 15% in the modelled intensity w.r.t. the SCIAMACHY intensity. For the time being, we assign this deficit to be due to an ozone deficit in the REPROBUS model, as suggested by comparisons with day-side GOMOS ozone vertical profiles obtained by the technique star occultation (not sensitive to an absolute

calibration).

In summary, we have found that the $O_2$* airglow is well organized, and quite predictable, with a dispersion of probably only a few per cent around a climatological average. Also, it should be almost as good as the ozone field in CTM models which are run with the actual meteo fields like ECMWF. Therefore, one could imagine that a GHG nadir viewing observation in the $O_2$ band could be corrected by subtraction of a model of the $O_2$* airglow

to get the pure nadir solar scattered intensity (on which is imprinted the $O_2$ absorption that we are analysing to get the $O_2$ column). However, the degree of accuracy that is needed for the determination of Psurf for useful measurements of GHG gases is very large, about ~ 0.1 hPa for the bias and 1 hPa for random error. According to our simulations, and if the airglow is ignored in the inversion but subtracted from a model, this airglow intensity model would have to be accurate to ~1.5% (for a mean radiance with albedo= 0.2) to achieve the 1 hPa

random error. Therefore, in most cases it is insufficient to rely entirely on a model to predict the actual airglow intensity to be subtracted from an observation. We need to disentangle in the observed spectrum itself the contribution of the airglow and the contribution of the solar scattered radiation. For this, we will rely on the fact that the spectrum *shape* of the $O_2$* airglow is different from the $O_2$ absorption spectrum.

**8.2 Spectral shape of the $O_2$* airglow**

It is clear that if the dayglow spectrum of $O_2$* were strictly identical to the $O_2$ absorption spectrum, it would be impossible to disentangle one from the other, and one would have to subtract blindly a model of the airglow emission from the observed spectrum, with an associated uncertainty.

While all the transitions of the $O_2$ 1.27 μm do exist in both the absorption spectrum and in the $O_2$* airglow emission spectrum, the resulting spectra in a nadir viewing geometry are different for two reasons and under three aspects.

First, the emission happens at high altitude and low air densities, while the absorption happens in the dense, lower atmosphere. Therefore, each absorption line is broadened by collisions, as shown on Fig. 2. Also, the

transmission at the centre of strong absorption $O_2$ lines is very low and the reflected solar flux is far to be



linearly dependent on line intensity (Tr($\tau$)=exp(-$\tau$)) contrary to the airglow that is directly proportional to it. Finally, there is the CIA effect producing a broad band absorption (Fig. 27), which is totally negligible in the upper atmosphere where emission takes place.

Second, for the same line, the rotational states J', J'' of the upper and lower states are inverted between absorption and emission, and since the rotational levels are populated differently, the P and R branches ($\Delta$J=$\pm$ 1) of the electronic transition have a different intensity distribution populated differently.

We have computed the theoretical shape of the dayglow spectrum from the Einstein coefficients $A_{21}$ of spontaneous emission from data contained in the HITRAN data base. It depends on the temperature of the atmosphere at the place of emission (rotational relative populations). We have compared our theoretical spectra degraded to the resolution of SCIAMACHY (~860) with limb observations of SCIAMACHY and found an overall excellent agreement, validating our theoretical approach of the airglow spectrum. This allowed performing some simulation exercises with good confidence about their ability to represent reality, showing that with the resolution of 25,000 of the MicroCarb instrument, it is indeed possible to disentangle the airglow emission from the $O_2$ absorption in the $O_2$ band at 1.27 $\mu$m. We note that the broad CIA band would not require such a high spectral resolution to be useful for the disentangling.

### 8.3. Simulation of Psurf Retrievals

Simulation exercises performed with the 4ARCTIC software have demonstrated that with the spectral resolution of Microcarb (25000), and only using the $O_2$ IR band B4 at 1.27 $\mu$m, we may retrieve Psurf (or similarly the $O_2$ vertical column of dry air) with an accuracy almost compliant with the strong MicroCarb requirements. The use of the B1 band (0.76 $\mu$m), in addition to the B4 band, will most likely yield a better understanding of the behaviour of this band which at present is not fully understood. Once better understood, it will certainly improve the accuracy of the Psurf MicroCarb retrieval by constraining further the aerosols optical properties. One may even hope that such a new knowledge could be used for an improved retrieval of other GHG missions.

One may wonder if the 25,000 spectral resolution is absolutely necessary to disentangle the $O_2$* emission from the $O_2$ absorption at 1.27 $\mu$m with sufficient accuracy for GHG retrievals. Sun et al. (2018) have explored spectral resolving power down to ~ 4,200, and found (their figure 4) an accuracy on the $O_2$ column of ~ 0.35% for an SNR of ~500 and a spectral resolution of 5,700, which would result on an error of 1.4 ppm for $X_{CO2}$, marginally insufficient. However their whole analysis was done without accounting for the CIA $O_2$ absorption, which broad size and smooth pattern is insensitive to spectral resolution (Fig. 27). Also noticeable on Fig. 27 is that with the same number of spectels as MicroCarb and a coarser spectral resolution (and sampling), the whole $O_2$ band would be measured for a better constraining of the CIA absorption band and $O_2$ column retrieval. The larger spectral sampling gives additional photons per spectel, which may be traded-off for an increased spatial resolution.

On the other hand, with our Igor–software breadboard tool, we have shown that with the SCIAMACHY spectral resolution (~ 860), it was possible to determine the intensity of the $O_2$* airglow over oceans (low albedo), but not to disentangle it and retrieve the $CO_2$ column, whatever the albedo, with a useful enough accuracy. We believe though, on the basis of our analysis and the results of Sun et al. (2018), that when CIA


would be taken into account, a spectral resolution of about 5,000 and a high SNR would yield a sufficiently good accuracy on the Psurf retrieval in the $O_2$ band at 1.27 μm, and could improve the treatment of aerosols and their wavelength dependent optical properties, being nearer the $CO_2$ bands, for a better $X_{CO2}$ retrieval. The $CO_2$ Mission space mission ($CO_2$-M) is sponsored by European Community, and developed and operated by ESA, as

a space segment for the monitoring of $CO_2$ anthropogenic emissions, with potentially a series of three operational spacecraft on sun-synchronous orbits. The present typical baseline optical design of the $CO_2$-M has a spectral resolution of about 6,300 at 0.76 μm, and 5,400 for the weak $CO_2$ channel at 1.60 μm. There is no channel for the $O_2$ band at 1.27 μm but it can be imagined by interpolation that a new channel for this band derived from the baseline design would have a resolution of about 5,750. Based on the discussion in the previous

paragraph and on the whole present study, we advocate for the inclusion in the design of $CO_2$-M instrument the inclusion of a channel at 1.27 μm.

       The Tanso-Gosat is a Fourier Transform Spectrometer dedicated to GHG monitoring on JAXA GOSAT space platform. The Tanso–Gosat 2 was launched in October 2018 and use bands at 0.76 (for $O_2$), 1.60 μm weak $CO_2$ band and 2.0 μm (strong $CO_2$ band). We advocate for the inclusion of an additional channel dedicated to the

measurement of the $O_2$ band at 1.27 μm in the design of future Tanso instruments, as well as for the chinese family of TANSAT following the TANSAT-1 already collecting data.

**Competing interests.** The authors declare that they have no conflict of interest.

**Author contribution**: JLB conceptualized revisiting the use of the 1.27 μm $O_2$ band for GHG retrieval and prepared the manuscript with contributions from all co-authors. JLB and PA developed the theory of airglow emission spectrum and the building of a synthetic spectrum. FMB is the principal investigator of the MicroCarb mission, in the frame of which this study was performed. This study was organized by DJ and technically managed by LB at ACRI and scientifically by JLB at LATMOS. JLB and AH developed the algorithms for the

analysis of SCIAMACHY data. FL developed the REPROBUS model and the $O_2$* airglow model and participated to comparisons with GOMOS and SCIAMACHY data. PL maintained the 4ARCTIC software. LB performed the SCIAMACHY data analysis for VER retrieval from limb measurements, and comparison with REPROBUS model and ozone GOMOS data. LB computed also the airglow synthetic spectra and made comparisons with SCIAMACHY limb spectra.

**Acknowledgements**
The present study was led by LATMOS and ACRI-ST, and funded by CNES (Centre National d'Etudes Spatiales) in 2016-2018 in the frame of preparation of the MicroCarb mission, a space mission entirely dedicated to the study of $CO_2$. JLB, AH, FL, and FMB acknowledge support of Centre National de la Recherche

Scientifique (CNRS). JLB and PA acknowledge partial support from Ministry of Education and Science of the Russian Federation grant n° 14. W 03.31.0017. We wish to acknowledge the useful involvement of other ACRI company members: Nicolas Chapron, Jean-Luc Vergely, Stéphane Ferron and Meriem Chakroun, and Claude Camy-Peyret for useful discussions. We wish to thank ESA for the ENVISAT programme, allowing the operations, data processing and archiving of SCIAMACHY and GOMOS data.



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



## Appendix A: Modelling of the terrestrial airglow emission of $O_2(a^1\Delta)$ at 1.27 μm

### A.1: Physics of $O_2(a^1\Delta)$ emission in the Earth's atmosphere

The emission in the near infrared of a photon at 1.27 μm occurs when molecular oxygen in its first excited

electronic state $O_2(a^1\Delta)$ spontaneously relaxes to its fundamental state $O_2(X^3\Sigma)$. It is one of the most intense lines measured in the atmosphere of telluric planets. This first part describes the mechanisms of production and relaxation of $O_2(a^1\Delta)$ in the Earth's atmosphere, shown in Fig. 17. They form the theoretical basis of the $O_2(a^1\Delta)$ emission model that we have developed for the present study.

### A.1.1   Production of $O_2(a^1\Delta)$

#### A.1.1.1 Direct Production

The main direct process leading to the formation of $O_2(a^1\Delta)$ in the Earth's middle atmosphere is the photo-dissociation of ozone at wavelengths shorter than 310 nm. This produces an $O_2$ molecule in the electronic state

$(a^1\Delta)$ and an oxygen atom in its excited state $O(^1D)$, with a quantum yield of about 0.9 :

$$J_H : \qquad O_3 + h\nu \ (\lambda < 310 \ nm) \to O_2(a^1\Delta) + O(^1D)$$

Another direct production process of $O_2(a^1\Delta)$ is the recombination of oxygen atoms in their fundamental state $O(^3P)$ :

$$O(^3P) + O(^3P) + M \to O_2(a^1\Delta) + M$$

This production of $O_2(a^1\Delta)$ only occurs in the thermosphere between 80 and 100 km of altitude. Its relative contribution to the vertically integrated emissivity of $O_2(a^1\Delta)$ being in the range of $10^{-4}$ (Wiensz, 2005), it will be neglected here.

#### A.1.1.2 Indirect Production by $O(^1D)$

A minor but not negligible contribution to the production of $O_2(a^1\Delta)$ is of an indirect character since it occurs

through the excited oxygen $O(^1D)$ and the subsequent energy cascade to $O_2(a^1\Delta)$. As shown in Fig. 17, this energy transfer is carried out via the intermediate state $O_2(b^1\Sigma)$.

Once $O(^1D)$ is formed by the photo-dissociation of ozone occurring at the rate $J_H$, its relaxation towards $O(^3P)$ occurs in a radiative or collisional way (quenching). The radiative process leads to the emission of a photon at 630 nm dictated by its probability $A_D$ ($s^{-1}$):

$$A_D : \qquad O(^1D) \to O(^3P) + h\nu \ \ (630 \ nm)$$

The oxygen "red line" at 630 nm is observed in the thermosphere where several important processes of $O(^1D)$ formation exist. This emission occurs with a time constant $1/A_D$ of about 2 minutes (Baluja and Zeippen, 1988). In the mesosphere and stratosphere this process is a minor one, because the deactivation of $O(^1D)$ is much more efficient through collisions with a molecule of $N_2$ or $O_2$:





$$k_{D,N2}: \quad O(^1D) + N_2 \rightarrow O(^3P) + N_2$$

$$k_{D,O2}: \quad O(^1D) + O_2 \rightarrow O(^3P) + O_2(b^1\Sigma)$$

The last process $k_{D,O2}$ indicated above produces an $O_2$ molecule in its second excited state $O_2(b^1\Sigma)$. This unstable molecule in turn transmits its energy either radiatively or collisionally. Radiative relaxation to the fundamental

5   state $O_2(X^3\Sigma)$ occurs with a time constant $1/A_\Sigma$ of 13 s (Mlynczak and Solomon, 1993) and is accompanied by the emission of a photon at 762 nm:

$$A_\Sigma: \quad O_2(b^1\Sigma) \rightarrow O_2(X^3\Sigma) + h\nu \text{ (762 nm)}$$

The collisional relaxation (quenching) of $O_2(b^1\Sigma)$ is carried out with O, $O_2$, $O_3$, $N_2$, and $CO_2$, and results in the lower energy state $O_2(a^1\Delta)$ :

$$k_{\Sigma,O}: \quad O_2(b^1\Sigma) + O \quad \rightarrow O_2(a^1\Delta) + O$$

$$k_{\Sigma,O2}: \quad O_2(b^1\Sigma) + O_2 \quad \rightarrow O_2(a^1\Delta) + O_2$$

$$k_{\Sigma,O3}: \quad O_2(b^1\Sigma) + O_3 \quad \rightarrow O + 2\,O_2 \qquad 70\%$$

$$\rightarrow O_2(a^1\Delta) + O_3 \quad 30\%$$

$$k_{\Sigma,N2}: \quad O_2(b^1\Sigma) + N_2 \quad \rightarrow O_2(a^1\Delta) + N_2$$

$$k_{\Sigma,CO2}: \quad O_2(b^1S) + CO_2 \quad \rightarrow O_2(a^1\Delta) + CO_2$$

Taking into account the contribution of $O(^1D)$ to the emission of $O_2(a^1\Delta)$ therefore requires a calculation of the concentration of $O_2(b^1\Sigma)$ which involves its radiative relaxation and its collisional relaxation rates with the 5 molecules above.

### A.1.1.3 Indirect production by solar excitation of $O_2(b^1\Sigma)$

20   For an accurate calculation of the $O_2(a^1\Delta)$ emission profile, it is also necessary to take into account the solar excitation (or optical resonance) of $O_2(b_1\Sigma)$ at 762 nm, whose indirect contribution to the $O_2(a^1\Delta)$ emission becomes significant above 60 km. Indeed, the absorption by $O_2$ of a solar photon in the $O_2$ A-band at 762 nm raises it directly to the energy level $O_2(b^1\Sigma)$:

$$g_{O2}: O_2 + h\nu \text{ (762 nm)} \rightarrow O_2(b^1\Sigma)$$

25   The photo-excited $O_2(b^1\Sigma)$ molecule is then affected by the same processes as those mentioned in the previous paragraph, which in case of quenching lead to the $O_2(a^1\Delta)$ state.

### A.1.2 Relaxation of $O_2(a^1\Delta)$

Once produced in its excited state $O_2(a^1\Delta)$, molecular oxygen transmits its energy either by radiative relaxation or by quenching with surrounding molecules. Radiative relaxation operates by emitting a photon in the near infrared at 1.27 μm, which lowers $O_2$ to its fundamental level $O_2(X^3\Sigma)$:

$$A_\Delta: \qquad O_2(a^1\Delta) \rightarrow O_2(X^3\Sigma) + h\nu \ (1.27 \ \mu m)$$

This transition is carried out with a fairly long time constant. The current consensus supported by the laboratory measurements of Lafferty et al. (1998) is $1/A_\Delta$ = 75 mn for the radiative lifetime of $O_2(a^1\Delta)$. The collision relaxation of $O^2(a^1\Delta)$ can be performed with $O_2$, $N_2$, O, $O_3$, or $CO_2$. According to current kinetics data (Burkholder et al., 2015; Atkinson et al., 2005), collisions with $O_2$ are very largely dominant in the Earth's middle atmosphere:

$$k_\Delta: \qquad O_2(a^1\Delta) + O_2 \rightarrow O_2(X^3\Sigma) + O_2$$

The frequency of collisions between $O_2(a^1\Delta)$ and $O_2$ obviously increases with density and therefore when descending in altitude. Thus the collision relaxation of $O_2(a^1\Delta)$ dominates the radiative relaxation of $O_2(a^1\Delta)$ at all altitudes below 75 km. It is 10 times larger at 60 km, and 50-100 larger at 50 km (Wiensz, 2005).

### A.2. Description of the $O_2(a^1\Delta)$ emission model

**A.2.1 Input data**

The input data used to calculate the volume emission rate of $O_2(a^1\Delta)$ (VER) are the vertical profiles of temperature, ozone $O_3$, and atomic oxygen $O(^3P)$. These profiles are provided by the REPROBUS three-dimensional chemistry-transport model (Lefèvre et al., 1994). REPROBUS is a global model with a horizontal resolution of 2°x2°. It includes a complete description of stratospheric chemistry using 58 species and about 100
chemical reactions. The winds and temperatures used by REPROBUS are forced by ECMWF operational analyses. Since 2013, REPROBUS has been integrated in near-real time on the IPSL servers in a vertical configuration identical to those of the ECMWF analyses: chemistry and transport are calculated on 137 pressure levels distributed from the ground to 0.01 hPa, or about 80 km altitude. The chemical species distributions calculated by REPROBUS are stored and available every day at 12 UT on the whole Earth.

In practice, the user of the $O_2(a^1\Delta)$ emission model developed here enters the latitude-longitude coordinates and the date he wishes to process. The model then extracts the ECMWF temperature profile as well as the $O_3$ and $O(^3P)$ profiles calculated by REPROBUS for the selected date and location. From the pressure and temperature are also calculated the total density and density profiles of $N_2$, $O_2$, and $CO_2$, adopting uniform mixing ratios of 0.78, 0.21, and $380 \times 10^{-6}$ for the latter species respectively.

**A.2.2 Processes included in the model**

**A.2.2.1 Photo-dissociation of ozone**

The calculation of the $J_H$ photo-dissociation rate of ozone is performed in the same way as in the 3D REPROBUS model. It uses an ultraviolet radiative transfer model inherited from the TUV model developed at



NCAR (Madronich and Flocke, 1998). It uses both spherical harmonics and discrete ordinates to represent the radiation field, resolved with a spectral interval of 0.5 nm in the Hartley band of ozone. The effective ozone absorption cross-section and the quantum efficiency of O($^1$D) production are derived from the latest JPL recommendation (Burkholder et al., 2015). To save computation time, $J_H$ is pre-calculated for the entire range of

zenithal angles (0-90°) that can be used in the 1D model. It is then stored in a "look-up table" (LUT) which is a function of the total air and ozone columns above the considered point, as well as the zenith angle. A simple three-dimensional interpolation in the look-up table allows to determine the vertical photo-dissociation profile $J_H$ for a given ozone profile and zenith angle.

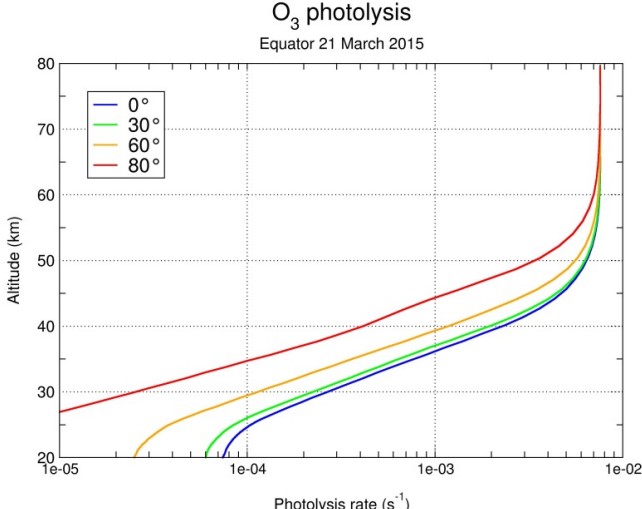

**Figure A1. Example of photo-dissociation rate $J_H$ of ozone (s$^{-1}$) calculated for equatorial conditions at the equinox and various solar zenith angles (0-80°).**

### A.2.2.2 Energy transfer by O($^1$D)

The indirect source of O$_2$(a$^1\Delta$) following the de-excitation of O($^1$D) requires first of all to calculate the concentration of O($^1$D), which is easily obtained by making the justified hypothesis of photochemical

equilibrium: the O$_2$(a$^1\Delta$) production rate $J_H$[O$_3$] is equal to the destruction rate which is proportional to [O($^1$D)], where the [ ] denotes the concentration in cm$^{-3}$:

$$J_H\,[O_3] = [O(\,^1D)]\big(A_D + k_{D,O2}\,[O_2] + k_{D,N2}\,[N_2]\big)$$

yielding:

$$[O(\,^1D)] = \frac{J_H\,[O_3]}{A_D + k_{D,O2}\,[O_2] + k_{D,N2}\,[N_2]}$$

with :

$A_D = 7.46 \times 10^{-3}$ s$^{-1}$       Spontaneous probability of transition (*Baluja and Zeippen*, 1988)

$k_{D,O2} = 3.3 \times 10^{-11}$ exp(55/T)       quenching by $O_2$       *Burkholder et al.*, 2015

$k_{D,N2} = 2.15 \times 10^{-11}$ exp(110/T)       quenching by $N_2$       *Burkholder et al.*, 2015

Once the concentration $[O(^1D)]$ has been determined, the density of $O_2(b^1\Sigma)$ can be calculated, which is obtained

by making the justified hypothesis of photochemical equilibrium by equalizing the terms of production and loss:

$$[O_2(b\ ^1\Sigma)] = \frac{k_{D,O2}\ [O(\ ^1D)][O_2] + g_{O2}\ [O_2]}{A_\Sigma + k_{\Sigma,O}\ [O] + k_{\Sigma,O2}\ [O_2] + k_{\Sigma,O3}\ [O_3] + k_{\Sigma,N2}\ [N_2] + k_{\Sigma,CO2}\ [CO_2]}$$

with

$A_\Sigma = 0.0758$ s$^{-1}$       *Mlynczak and Solomon*, 1993

$k_{\Sigma,O} = 8 \times 10^{-14}$       *Burkholder et al.*, 2015

$k_{\Sigma,O2} = 3.9 \times 10^{-17}$       *Burkholder et al.*, 2015

$k_{\Sigma,O3} = 3.5 \times 10^{-11}$ exp(-135/T)       *Burkholder et al.*, 2015

$k_{\Sigma,N2} = 1.8 \times 10^{-15}$ exp(45/T)       *Burkholder et al.*, 2015

$k_{\Sigma,CO2} = 4.2 \times 10^{-13}$       *Burkholder et al.*, 2015

In the above expression $g_{O2}$ is the solar excitation rate of $O_2(b^1\Sigma)$ at 762 nm, detailed in the following paragraph.

The term $S_\Sigma$ representing the source of $O_2(a^1\Delta)$ via $O(^1D)$ and $O_2(b^1\Sigma)$ can therefore be expressed (in

molecule.cm$^{-3}$.s$^{-1}$) with:

$$S_\Sigma = (k_{\Sigma,O}\ [O] + k_{\Sigma,O2}\ [O_2] + k_{\Sigma,O3}\ [O_3]\ \phi_{O3} + k_{\Sigma,N2}\ [N_2] + k_{\Sigma,CO2}\ [CO_2])\ [O_2(b\ ^1\Sigma)]$$

by noting $\Phi_{O3} = 0.3$ ( Sander et al., 2011) the quantum efficiency of $O_2(a^1\Delta)$ production in collisions between $O_2(b^1\Sigma)$ and $O_3$ molecules.

### A.2.2.3 Solar excitation of $O_2(b^1\Sigma)$

The calculation of the solar excitation rate $g_{O2}$ of $O_2(b^1\Sigma)$ in the A-band at 762 nm is parameterized from the

line-by-line calculations of Wiensz (2005). The expression of $g_{O2}$ used here is the form:

$$g_{O2}(N_{O2}) = g_\infty \exp\left(-b\ N_{O2}^c\right)$$

where $N_{O_2}$ is the oblique column of $O_2$ above the considered point and $g_\infty = 6.1 \times 10^{-9}$ s$^{-1}$ the solar excitation rate at the top of the atmosphere. The coefficients b and c (power law) are determined by a fit log/log of the data from Wiensz (2005). As an example, Fig. A2 illustrates the $g_{O_2}$ values calculated at the Equator for several zenith angles. We verified that these results were in good quantitative agreement with the complete calculation of Wiensz (2005).

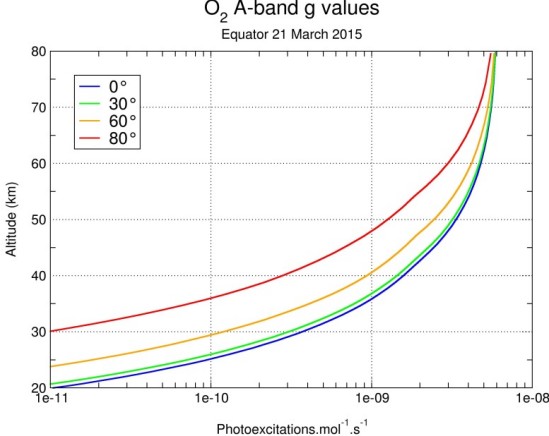

Figure A2 : Solar excitation rate $g_{O_2}$ of $O_2(b^1\Sigma)$ in the A-band at 762 nm (photo-excitation in molecule$^{-1}$.s$^{-1}$) calculated for equatorial conditions at the equinox and various solar zenith angles.

**A.2.2.4 Emission of $O_2(a^1\Delta)$**

The final step is to calculate the $O_2(a^1\Delta)$ emission rate. Its number density (molecule.cm$^{-3}$) is obtained using the source terms $J_H$ and $S_\Sigma$ calculated previously, assuming photochemical equilibrium:

$$[O_2(a\ ^1\Delta)] = \frac{J_H\ [O_3] +\ S_\Sigma}{A_\Delta +\ k_{\Delta,O2}\ [O_2]}$$

with

$A_\Delta = 2.237 \times 10^{-4}$ s$^{-1}$          *Lafferty et al., 1998*

$k_{\Delta,O2} = 3.0 \times 10^{-18}$ exp(-200/T)          *Atkinson et al., 2005*

The volume emission rate (or emissivity) $\epsilon_\Delta$ at 1.27 µm expressed in photons.cm$^{-3}$.s$^{-1}$ is simply obtained by multiplying the concentration of $O_2(a^1\Delta)$ by $A_\Delta$ :

$$\epsilon_\Delta = [O_2(a\ ^1\Delta)].A_\Delta$$





Since here we have assumed an instantaneous photochemical equilibrium for $O_2(a^1\Delta)$, our model cannot reproduce properly the situations when the loss term of $O_2(a^1\Delta)$ becomes very long with respect to its production term, as it is the case at very large zenith angles at dusk (Fig. 31).

**A.2.3 Processes neglected in the model**

In addition to the direct production of $O_2(a^1\Delta)$ by recombining atomic oxygen, three other physical mechanisms have been neglected in our model, based on the following reasons:

- The production of $O(^1D)$ from the photo-dissociation of $O_2$ in the Schumann-Runge continuum (SRC) and at Lyman-$\alpha$. These processes noted respectively $J_{SRC}$ and $J_{Ly-\alpha}$ in Fig. 17 become important only in the
thermosphere above 90 km. Their contribution to the emission of vertically integrated $O_2(a^1\Delta)$ from the ground is between 0.1 and 1% (Wiensz, 2005).

- The photo-excitation (or optical resonance) of $O_2(a^1\Delta)$ at 1.27 μm. This process occurs throughout the stratosphere and mesosphere. It contributes less than 1% of the integrated $O_2(a^1\Delta)$ emission (Wiensz, 2005).

**A.3. Airglow Model Results**

Our model provides the vertical profiles of the mixing ratios of $O(^1D)$, $O_2(b^1\Sigma)$, $O_2(a^1\Delta)$, the emissivity (same as VER) profile at 1.27 μm expressed in photon.cm$^{-3}$.s$^{-1}$, and the vertically integrated emissivity $I_{ag}$ expressed in photon.cm$^{-2}$.s$^{-1}$.sr$^{-1}$.

**A.3.1 Vertical profiles of the species involved in the $O_2(a^1\Delta)$ emission**

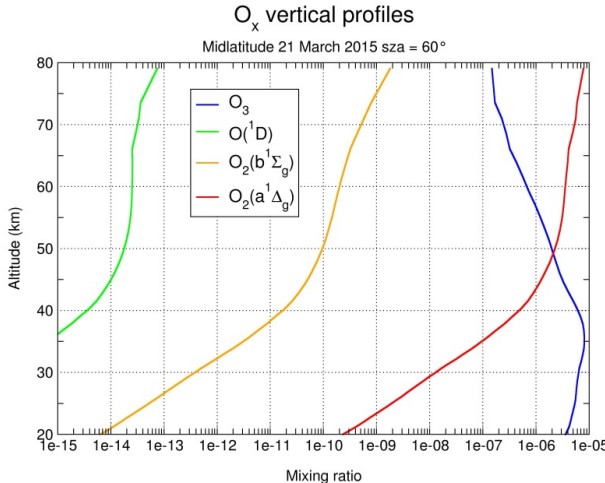

**Figure A3. Mixing ratios of $O_3$, $O(^1D)$, $O_2(b^1\Sigma)$, $O_2(a^1\Delta)$ calculated by the model at mid-latitudes at the equinox, at a solar zenith angle of 60°.**




### A.3.2 Contributions to the $O_2(a^1\Delta)$ emission.

On Fig. A4 are plotted the various contributions to the $O_2(a^1\Delta)$ emission: the $O_3$ photo-dissociation, the $O(^1D)$ energy transfer, and the $O_2(b^1\Sigma)$ photo-excitation at 762 nm. In the stratosphere, the photo-dissociation of $O_3$ represents about 80% of the total $O_2(a^1\Delta)$ emission and the energy transfer of $O(^1D)$ about 20%. The solar
excitation of $O_2(b^1\Sigma)$ becomes important in the mesosphere. It contributes 20% to $O_2(a^1\Delta)$ at 60 km and 50% at 80 km. These results are consistent with the literature (Mlynczak and Olander, 1995; Wiensz, 2005) but an accurate comparison would require the use of identical input profiles (especially ozone).

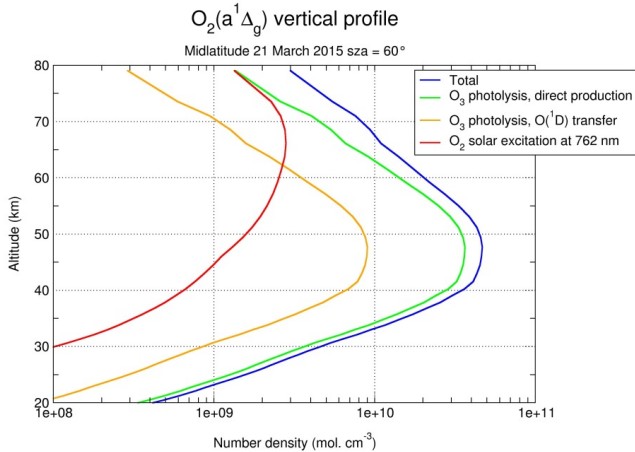

**Figure A4. Total number density of $O_2(a^1\Delta)$ and respective contributions of $O_3$ photo-dissociation, $O(^1D)$ energy**
**transfer, and $O_2(b^1)$ photo-excitation at 762 nm. These profiles are calculated by the model for a zenith angle of 60° at mid-latitudes and at the equinox.**

### A.3.3 $O_2(a^1\Delta)$ Emissivity Vertical Profiles

Fig. A5 present two examples of emissivity profiles at 1.27 μm (VER, Volume Emission Rate in photon.cm$^{-3}$.s$^{-1}$) calculated at fall equinox, two latitudes and various zenith angles. This is to illustrate the spatio-temporal
variability of the vertical profile of the $O_2(a^1\Delta)$ emission.



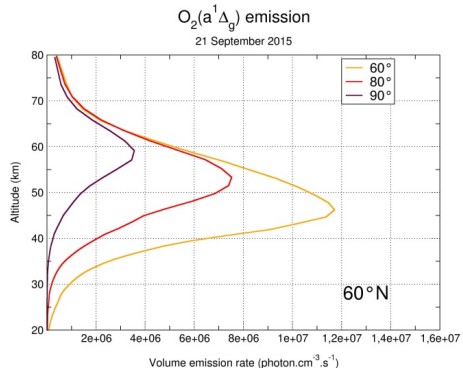

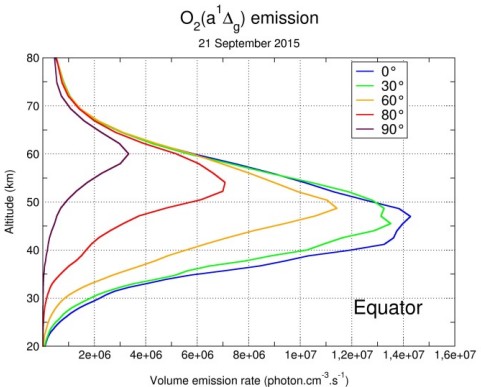





**Figure A5. Emissivity profiles of O$_2$(a$^1$Δ) calculated by the model at the fall equinox (21 September 2015) for two latitudes and various zenith angles (with different colours for indicated SZA).**

**A.3.4 Integrated vertical intensity I$_{ag}$**

The model also provides the output emissivity of O$_2$(a$^1$Δg) vertically integrated between the ground and 80 km altitude, or integrated vertical intensity I$_{ag}$ (index ag for airglow), which is observed in nadir viewing expressed in photon.cm$^{-2}$.s$^{-1}$.sr$^{-1}$. On Fig. A6 (left) are represented the values of I$_{ag}$ for equinoxes and solstices (Mars, June, September, December) at various latitudes (labelled 60S, 40 S 40N, and 60 N) as a function of solar zenith angle (SZA). The main factor dictating the emerging vertical intensity I$_{ag}$ is the value of SZA. Figure A6 (right) represents the variation of the I$_{ag}$ intensity calculated at

12:00 UT during a complete year, from a 3D simulation of REPROBUS forced by the 2015 ECMWF analyses. For a given date, the latitude distribution of the intensity reflects the slow seasonal/latitude evolution of the ozone field.

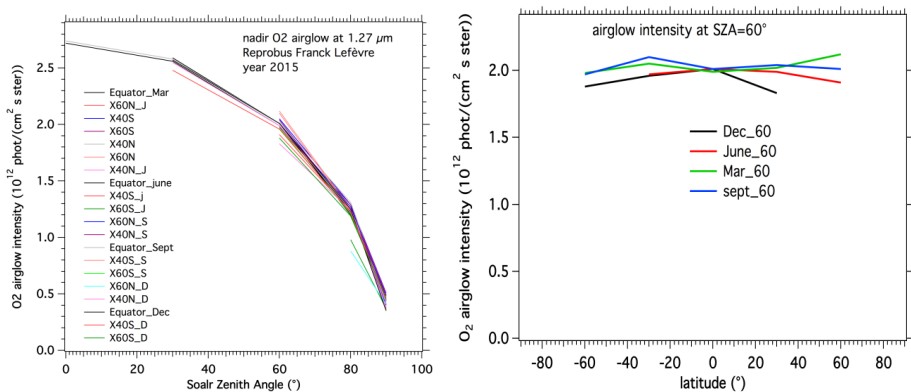

**Figure A6. Left: O$_2$(a$^1$Δg) airglow intensities I$_{ag}$ in nadir viewing (10$^{12}$ photon.cm$^{-2}$.s$^{-1}$.sr$^{-1}$) as a function of SZA and for various latitudes and seasons. Right: I$_{ag}$ intensities calculated at noon (12 UT) for four days of the year representative of various seasons at**

**solar zenith angle of 60° as a function of latitude, for various seasons.**

Finally, on Fig. A7 are plotted the variations of I$_{ag}$ airglow intensity for a point of longitude 0 and 12H local time as a function of the day of the year along year 2015. Small fluctuations represent the day-to-day variability of the ozone field.



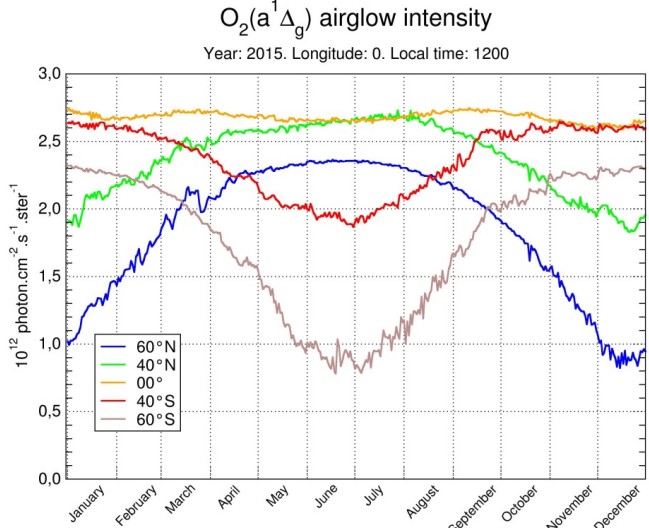

**Figure A7. Evolution of $I_{ag}$ airglow intensity for geographical points of longitude 0 and five latitudes (60°N, 40°N, 0°, 40°S, 60°S) and local time 12:00 LT as a function of the day of the year 2015. Small fluctuations represent the day-to-day variability of the ozone field.**

