# Peer review of "The use of $O_2$ 1.27 $\mu m$ absorption band revisited for Green House Gases monitoring from space and application to MicroCarb"

_Atmospheric Measurement Techniques, 2019_

## Referee Comment (RC1) · Anonymous Referee #1 · 30 Jun 2019

The feasibility of using the 1.27 $\mu$m, or the "singlet Delta" O$_2$ band for satellite nadir remote sensing of greenhouse gases has been investigated previously (Sun et al. 2018). However, this manuscript, to my knowledge, firstly presents this observation strategy within the framework of a planned satellite mission (MicroCarb). Since such a satellite mission may cost hundreds of millions of euros/dollars and provide crucial data with global coverage for many years, this strategy may potentially have a far-reaching impact, and this work may become an import reference for future missions. Given the aforementioned significance, this manuscript at this stage may still be substantially improved in terms of both presentation quality and technical rigor. The following lists my general comments concerning the manuscript structure and specific comments con-

cerning individual sentences, equations, and figures.

**1 General comments**

**1.1**

In general the manuscript has been clearly improved since the initial submission. That said, it still reads a bit fragmental with different sections apparently from different groups of authors. The $O_2$ molecules in $a^1\Delta_g$ state, for example, was denoted as $O_2^*$, $O_2(a^1\Delta_g)$, $O_2^*(a^1\Delta)$, $O_2(1\Delta)$, and $O_2(a^1\Delta)$ throughout the manuscript.

As to the paper structure, Section 5 describes the MicroCarb mission and its physics-based retrieval, 4ARTIC. Then it was deviated to a retrieval of SCIAMACHY nadir airglow using a different tool called LATMOS breadboard (sections 6.1–6.2) before 4ARTIC was applied to test different airglow mitigation approaches in section 6.3. The authors may consider moving sections 6.1–6.2 after section 3, because the SCIAMACHY nadir/limb airglow comparison naturally follows the onion-peeling VER retrieval, and combining section 5 with section 6.3. Some parts of section 7 seem to be out of the scope of this study (e.g., the discussion of methane and CO fluorescence), and the remaining parts may be combined with section 2.1.

**1.2**

Section 2.3 could be clarified and largely removed from the manuscript.

First, the main conclusion from this section, equations 14 and 15 that will be used to convert absorption spectra simulated by LBLRTM to airglow emission spectra, seems to have already been demonstrated by equations 3–5 in Sun et al. (2018). Combining

equations 4 and 5 in Sun et al. (2018):

$$r_g(\nu) = n(O_2(a^1\Delta_g)) \sum_{ij} A_{ij} \frac{g' \exp(-c_2(E_i + \nu_{ij} - \nu_{0,0})/T)}{Q(T; \text{airglow})} f(\nu, \nu_{ij}, T, p)$$

Following the terminology in this manuscript, define a similar "emission rate per $O_2^*$ molecule" for transition $ij$, denoted by $\varepsilon_{ij}$:

$$\varepsilon_{ij} = A_{ij} \frac{g' \exp(-c_2(E_i + \nu_{ij} - \nu_{0,0})/T)}{Q(T; \text{airglow})}$$

Line intensity $S_{ij}$ is given by equation 3 in Sun et al. (2018), copied here:

$$S_{ij} = I_a \frac{A_{ij}}{8\pi c \nu_{ij}^2} \frac{g' \exp(-c_2 E_i/T)(1 - \exp(-c_2 \nu_{ij}/T))}{Q(T)},$$

Take the ratio

$$\frac{\varepsilon_{ij}}{S_{ij}} = 8\pi c \frac{Q(T)}{Q(T; \text{airglow})} \nu_{ij}^2 \frac{\exp(-c_2 \nu_{ij}/T) \exp(-c_2(E_i - \nu_{0,0})/T)}{\exp(-c_2 E_i/T) \exp(-c_2 v_{ij}/T)(\exp(c_2 v_{ij}/T) - 1)}$$

and simplify

$$\frac{\varepsilon_{ij}}{S_{ij}} = 8\pi c \frac{Q(T)}{Q(T; \text{airglow})} \nu_{ij}^2 \frac{\exp(c_2 \nu_{0,0}/T)}{\exp(c_2 v_{ij}/T) - 1}$$

The equation above is identical to equations 14–15 in the manuscript. As such, most equations/figures in section 2.3.2–2.3.6 can be replaced by a simple reference to equations 3–5 in Sun et al. (2018).

Second, the symbols are not consistently defined, making section 2.3 challenging to follow. For example, the upper state energy is $E_2$ in equation 9, $E_{2i}$ in equation 10, and $E_i$ in line 9, page 11. The total partition sums for both absorption transition and airglow emission are denoted as $Q_{tot}(T)$, but separately as $Q_{tot}^{lo}(T)$ and $Q_{tot}^{up}(T)$ in equation

15. The summation index is $i$ in equation 10, but appears to be $J'$ in equation 11. The local emission rate spectrum is given by $E_m(\lambda)$ (line 17, page 15) and $E_{mn}(\lambda)$ (line 18, page 15) in section 2.3.7, but it is unclear what this quantity really is.

Finally, it will be very useful to discuss the improvements over previous studies. I did not find an equation that calculates $Q_{tot}^{up}(T)$ in Sun et al. 2018, which is given by equation 10 in the manuscript. However, I cannot reproduce the results $Q_{tot}(T = 296K) = 147.196$ (line 10, page 11) using equation 10 and HITRAN 2016 data because it is unclear what "index $i$ refers to each value of $E_2$" (line 36, page 10) exactly means.

**1.3**

One significant missing piece of this study is the fine-scale spatial variation of airglow, potentially due to gravity waves. The REPROBUS CTM ($2 \times 2°$) and SCIAMACHY limb data ($400 \times 960$ km) used in the study are too coarse to capture those fine-scale variations. The fine-scale airglow features may propagate to the retrieved $XCO_2$ as artefacts from the disturbed mesospheric temperature (spectral shape of both emission and re-absorption), excited $O_2$ molecule concentration (emission), and $O_2$ molecule concentration (re-absorption). The authors may have missed one opportunity as the SCIAMACHY nadir data over ocean have much higher spatial resolution and may reveal some gravity wave features. Larger scatter is seen from SCIA nadir ocean compared to SCIA limb (Figure 30). Is that just retrieval random errors or real spatial variation? How do they look on a map?

**2  Specific comments**

Line 31, page 1: the statement that the theoretical synthetic spectrum is from "a new approach" is contradictory to the following statement that this approach "is very similar

(likely identical) to the approach of Sun et al. (2018)" (lines 7–8, page 2). The current work should be more clearly distinguished from previous before claiming it is "new".

Line 32, page 2: may include OCO-3 and GOSAT-2.

Line 40, page 4: why is pressure broadening proportional to the "square" of air density?

Figure 2 caption: how can "transmittance" be "larger" than emission? Did the authors mean "wider"?

Line 35, page 6: equation 3 looks the same as the second equation in the Appendix A1.1.1. They should be consistent.

Line 32, page 7: Dave Crisp et al. should be just Crisp et al.

Line 30, page 9: quenching is significant below 50 km, and the airglow peak is also below 50 km. Are there any consequences if the quenching is ignored?

Line 15, page 14: $E_0 = 7892.02$ cm$^{-1}$ is inaccurate. Refer to Leshchishina et al. (2010).

Line 40, page 16: $\varepsilon$ appears to be spectrally integrated here, but spectrally resolved in equation 13, and back to be spectrally resolved in line 28, page 18 again.

Line 5, page 18, equation 21, and line 4, page 19: a bold face $\varepsilon$ is used, but definition is unclear.

Line 24, page 21: why does $O_2$ absorption have to be neglected? It is important and all parameters should be readily available to include $O_2$ absorption in the analysis.

Figure 11: this figure may be removed. The SCIAMACHY limb data are binned across track so it is unnecessary to show separate across-track positions. The globe distribution of SCIAMACHY orbits is well represented by Figure 15.

Figure 12b and the left panel in Figure 13: these should be removed, see the following argument.

Line 8, page 23: This is not a "simplified" model; the equation for absorption cross

section is not simpler than equation for airglow emission. This is an "inaccurate" or "inappropriate" model.

Figure 16: SZA$_{recomputed}$ appears in the figure but is only explained at the end of page 31.

Lines 5–6, page 27: VER has been defined and should be only defined at first appearance.

Section 4.1.1, page 27: the authors are suggested to better articulate the usefulness of the REPROBUS model. Its low resolution is not sufficient to capture the fine-scale variation; its model top (0.01 hPa) does not fully capture the airglow profile (what is the error from neglecting airglows above 0.01 hPa?); the model is not used in the following MicroCarb retrieval tests.

Lines 13–14, page 29: how much is the error due to neglecting the reabsorption, and what is its relative contribution to total error?

Figure 18: it's maybe useful to mark the subsolar point in the plot.

Line 23–24, page 31: may change to "...an observer placed above the tangent points..." as there are multiple tangent points that are close to each other.

Line 41, page 31: the SZA is defined at "one of the two points corresponding to the intersection between the line of sight and TOA". Which one?

Figure 19: removing the lines between markers may improve the figure clarity.

Line 7, page 33: it could probably be delivered in a more rigorous way than asserting: "must have been".

Figures 20–22: Figure 22 should be enough to present all results so Figures 20–21 can be removed.

Line 2, page 36: modified to "five orders of magnitude of variation of ozone with alti-

[Figure]

tude".

Figure 25: what is the difference between the left and right? What are the observation time/SZA?

Line 5, page 39: the claim that "the night side model bias is quite negligible below 60 km" contradicts the main conclusion of this section, i.e., the ozone deficit below 60 km of 10–20% in the REPROBUS model is the main reason of airglow underestimation. Moreover, is it possible to compare with other reanalysis/observation data, especially MERRA-2?

Line 8, page 41: change "'maximum height $Z_{aero}$" to something like "'peak AOD height $Z_{aero}$" to avoid confusion.

Line 27, page 41: the 2.04 $\mu$m band is not the same as OCO and GOSAT satellites, which use the 2.06 $\mu$m band. Why MicroCarb is using a slightly shifted window (lower left panel, Figure 26)?

Lines 8–9, page 42: the water vapour lines are not present in this figure.

Figure 27: what are the vertical axes on the right? In addition, it is hard to see, but there is a red line very close to the horizontal axes. What is that?

Lines 12–19, page 42: those are repetition of lines 36–40, page 4, but not exactly the same. The same argument does not need to be repeated twice in the article.

Figure 28: The "@$1.27\mu$m" may be removed in the vetical axis label.

Line 18, page 49: The REPROBUS model is coarse and cannot capture small-scale intensity variations, if they exist.

Line 4, page 50: what is the resolution of the "very high resolution" spectrum?

Lines 23–26, page 50: what are the a priori errors, especially for Psurf and airglow scaling factors? What is the impact of a priori on posterior error? Does the airglow

impact retrieved $XCO_2$?

Line 29, page 50: what is "Lmoy"?

Lines 10–16, page 51: the cold/warm spectra here differ from what has been defined in equations 29–30 (spectra simulated at 217/270 K). Does this make any difference?

Line 31, page 56: change "dayglow" to airglow.

Line 39, page 56: may remove the exclamation mark after "contaminated".

Lines 13–15, page 57: the comparison with day-side GOMOS ozone profiles appears to be inconclusive according to section 4.3.2 and may not support the statement here.

Lines 31–32, page 57: the first sentence may be inaccurate because a few factors stated previously in the manuscript. The absorption is still saturated while airglow emission grows linearly, the absorption is much more pressure-broadened, and only CIA is present in absorption. All those factors still hold even the airglow spectral shape were identical to absorption.

Line 5, page 58: the $O_2$ $a^1\Delta_g$ band is significantly more complicated than just P, Q, and R branches. The same argument applies to discussion of transitions in sections 2.3.3–2.3.5, which is suggested to be removed.

Lines 31–32, page 58: I've found CIA being considered in Figure 3 and Eq. S5 in Sun et al. (2018), so it is inaccurate to state that "their whole analysis was done without accounting for the CIA $O_2$ absorption".

---

## Referee Comment (RC2) · Anonymous Referee #4 · 12 Jul 2019

General Comments: I believe that the authors intend to show in the submitted manuscript that $O_2(1\Delta_g)$ airglow can be modeled with sufficient accuracy to use the 1.27 $\mu$m $O_2$ absorption band to retrieve $O_2$ columns for greenhouse gas studies. It seems that the authors have done a lot of very good work over the past 3 years or so, and a lot of it appears in this manuscript. The paper is very long at 75 pages, and I am not sure that all this detail needs to be in the paper, as some of it is in the published literature ($O_2$ spectroscopy and non-LTE calculations, for example) and references might suffice. I feel that the paper might be easier to follow and make a stronger case for the conclusion if content in the main body were limited to what is needed to support the conclusion, and use references or supplemental material otherwise. More specific

suggestions follow below.

I thought it might also help to include some discussion regarding how the $O_2$ column retrievals using the 1.27 $\mu$m band will be validated, given the very high accuracy ( 0.01%) that is required. Will they be compared against the $O_2$ A-band retrievals? But if $O_2$ A-band retrievals are good enough to be the standard, then what is the benefit of switching to the 1.27 $\mu$m band, given the added complication of the airglow correction? How will one know if the new retrievals are better?

Specific Comments:

I found the papers by Zarboo et al. (2018), Sun et al. (2018), and Simeckova et al. (2006), all cited by the authors, to be particularly helpful, and I think that there are places where the present manuscript presents conclusions or material that is similar (although clearly independently derived), so there are opportunities to make use of references to shorten the text. Again, my intent in making this comment is to find a way to limit the material in the paper to what is required to support the conclusion.

a) Introduction, sections 2.0, 2.1: I think this was about the right length, although I'm not sure that the discussion of observations of Venus and Mars add much to supporting the goal of the paper.

b) 2.2,2.3: This section is around 7.5 pages, and includes a lot of standard spectroscopy and line-by-line radiative transfer calculation information, including the use of Einstein A coefficients for non-LTE situations. The discussion could be shortened considerably by the use of references and limiting the text to what is unique. I was a little uncomfortable with the way that LBLRTM is being used in section 2.3.7, as it would be cleaner to just start from scratch with fresh code and do it right, but I appreciate that this may not have been practical given the resources available and it seems to have worked.

c) Section 3, the use of SCIAMACHY data: The authors have done a lot of work here,

but I would suggest including only those elements of this section that are directly relevant to section 6. This section is 11.5 pages, and it is not clear to me how the onion peeling retrieval of VER from limb scans is relevant to simultaneous nadir retrievals of $O_2$ column and airglow from MicroCarb.

d) Section 4, comparison between REPROBUS airglow model and SCIAMACHY observations: The conclusion seems to be that the model underpredicts ozone and so underpredicts airglow, and so is not suitable for estimating airglow instead of retrieving it. Not sure if this is worth 12.5 pages; perhaps this work could be summarized?

e) Sections 5 and 6: It seems to me that this is the heart of the paper, and other sections should be adjusted so that they contain just what is needed to support the material in these two sections.

f) Section 7: This seems to be a literature review, and not directly relevant, except perhaps the comments regarding $CO_2$ airglow and potential impact on MicroCarb retrievals.

g) Section 8, Conclusion: might need adjusting if the revisions above are considered.

Technical Corrections: Some of these corrections may be OBE if the major changes identified above are considered:

1)page 17, eq. 19: I think that the expression under the sq root in the third line should be $r_0^2 - p_1^2$, NOT $r_1^2 - p_0^2$

2) Figure 19: It was very difficult to distinguish the stars and diamonds in the plots.

3) page 40, line 21: 4ARTIC retrieves $CO_2$ and $H_2O$ on 19 vertical layers: what is the typical number of degrees of freedom for these retrievals? Is the profile information actually meaningful, or is it really just a column retrieval?

4) page 41, line 20: "...Henyey Greenstein function with g currently fixed to 0.8." A reference might be helpful here, for the function and for the choice of g.

5) page 44, line 4: drop the extra "(" in $Ag((\lambda)$; line 13: what reference spectrum is used?; line 14: drop the "." after "spectrum.,"; line 27: change "spectru" to " spectrum"; line 28: So random error only, no calibration error, channel crosstalk, etc?

6) page 55, line 14: delete "contaminated" (redundant after "contamination"); line 35: change "ETL" to "LTE"?;

7) page 59, line 11: delete "inclusion of a" (redundant with previous "inclusion in the")

8) page 66, line 15: I think that $O_2(b^1S)$ should be $O_2(b^1\Sigma)$?

---

## Referee Comment (RC3) · Anonymous Referee #2 · 23 Jul 2019

General comments:

This paper deals with an interesting and relevant aspect, i.e. the contamination of nadir O2 measurements in the 1.27 micron IR atmospheric band caused by the corresponding airglow emission. The paper will be relevant for the community and the approaches employed in the study appear to be robust. However, there are two major issues with the paper. It is unnecessarily long (which distracts from the main content) and it is full of typos, minor inconsistencies and little errors. Many sections make the impression of hastily written drafts that were not proof read. It took me two full days to go through this manuscript, which is not acceptable. We expect from young scientists and postdocs

that manuscripts are in tip-top shape and it should also be expected from senior scientists. It is certainly not the reviewer's task to correct all the mistakes. Please correct the manuscript carefully.

Specific comments:

P. 1, line 18: "i.e. OCO-2 .. " -> "e.g. OCO-2 .."; otherwise all instruments carrying out these measurements should be listed.

P. 1, line 38: "model underestimate" -> "model underestimates"

Same line: "This is fully confirmed .."

It is not entirely clear, what this refers so. Is it confirmed that the model underestimates by 15%? Or that the airglow intensity is mainly determined by the SZA. Please clarify.

P. 2, line 13: "Dioxyde" ->"dioxide"

P. 2, line 14: "The atmospheric fraction .."

I suggest being more precise here. Mention $CO_2$ explictly and that the fraction is a mass ratio – that's what it is, right?

P. 2, line 30: "The first satellites to be launched with the aim .."

Why don't you list SCIAMACHY/Envisat? SCIAMACHY is certainly not as specialized to $CO_2$ retrievals as the other instruments, but it was also built to measure $CO_2$ and it should be listed here. The $CO_2$ retrievals were also quite successful.

P. 3, line 13: "mission" -> "missions"

Same line: "Airglow has a spectrum that is very similar .."

This statement is not generally true for all airglow emissions occuring in the atmosphere and should be phrased more precisely. There are many other airglow emissions apart from O2, for which the statement is not valid.

P. 3, Figure 1: the figure shows O3 photolysis as the only source of the 1.27 micron emission. Ozone photolysis is only one of several excitation mechanisms. I suggest stating in the caption that it is the main mechanism on the dayside.

Caption Fig. 1, line 3: "are crossing" -> "is crossing"

Same line: "which emission" -> "whose emission"

P. 4, line 6: "confirms" -> "confirmed"

P. 4, line 9: "mission" -> "missions"

P. 4, line 12: "to determine from nadir viewing observations the CO2 vertical columns"

Word order wrong. Please replace by: "to determine CO2 vertical columns and mixing ratios from nadir viewing observations"

P. 4: "secular variation" -> "secular variations"

P. 4, line 21: "plugged to a" -> "combined with a"

P. 4, line 22: "We also note that the TCCON ground-based spectrometer array, observing the sun, uses this 1.27 $\mu$m band to derive the CO2/dry air mixing ratio"

This is the third time this is mentioned and can be removed.

P. 4, line 32: "intimately intricated."

Grammar wrong, please correct.

P. 4, line 38: "Second, the transmittance Tr=exp(-$\tau$) saturates at high optical thicknesses $\tau$>1,while the emission does not."

I don't understand this statement and think it's wrong. Emission is certainly also limited if the optical depth becomes quite large. What you probably mean is that extinction is not important for the emission, because the emission mainly occurs above 30 km. But this is not what the sentence states.

P. 5, line 1: "on Fig." -> "in Fig."

P. 5, line 3: "which positions" -> "whose positions"

P. 5, line 5: "rending" ??

Do you mean "rendering"? Word order is also wrong: "rendering this proposal unpractical"

P. 5, line 7: "contributes" -> "contribute"

Caption Fig. 2, line 1: "The transmittance within an individual O2 line (red) is much larger than"

"transmittance .. much larger" is not well phrased. The transmittance is zero in the center of the line. Please be more precise.

P. 5, line 22: "compared" -> "compare"

Same tense as in previous sentences

P. 6, line 1: "while in Section 6 are detailed the accuracy and bias results of"

Word order incorrect

P. 6, line 4: "In section 7 are examined briefly"

Word order wrong: "In section 7 some other cases . . . are examined"

P. 6, line 20: suggest to define Rayleigh the first time it is used.

P. 7, line 3: "strong solar light scattered component" -> "strong contribution of scattered solar radiation"

P. 7, line 5: "From high altitude the O2 absorption will be a little bit reduced."

This is incomprehensible? Do you mean that the absorption is weaker at higher altitudes?

P. 7, line 26: please cite Bovensmann and/or Burrows here. For all the other instruments you provide a citation, but not for SCIAMACHY.

P. 7, line 37: "On Fig. 3 (from Khomich et al., 2008) are represented the various electronic"

"On Fig." -> "In Fig" and the word order is incorrect: "Fig. 3 presents the various .."

P. 7, last line: "on Fig." -> "in Fig."

P. 8, line 2: "on Fig." -> "in Fig."

P. 8, equation (4): "C" -> "c"

Same equation: the units are incorrect, i.e. the equation is not valid as is. Please correct.

P. 8, line 18: "Fraunhofer" -> "Fraunhofer's"

P. 9, line 24: "solar effect"

One can tell what you mean, but it is not well phrased, too unspecific.

P. 9, line 31: "emitted photon" -> "emitted photons"

P. 9, equation (5): the middle part of the equation is incorrect. During the day, there will essentially be a steady state, i.e. $O_2^*$ is produced by $O_3$ photolysis (mainly) and removed by emission, i.e. $d[O_2^*]/dt = 0$.

P. 10, title section 2.3.3: remote period from section title

P. 10, line 15: "A –coefficients" -> "A-coefficients"

P. 10, line 16: "second members"

Unclear, what you mean.

P. 10, line 34: please provide k_B in SI units

P. 11, line 4: "on all" -> "over all" (2 occurrences)

P. 11, line 11: "on Fig." -> "in Fig."

P. 11, line 12: "term by" -> "term and"

P. 11, caption Fig. 4, line 3: "There are 5, 7, or 8 values (and transitions) for each black circle on the figure"

Not clear to me, why there are 5, 7 or 8 values for each black circle. Please explain.

Same line: "on the figure" -> "in the figure"

P. 11, line 21: "On Fig. 5 are represented the various energy"

"On" and word order incorrect: "Fig. 5 presents .."

P. 12, Title section 2.3.4: add space at beginning

P. 12, line 6: "sate" -> "state"

Next line: "sum on" -> "sum over"

P. 12, line 21: "We found that the total decay rate is A21tot=2.29 10-4 s-1."

Above you determined the total decay rate to be 2.22 x 10-4 s-1. What does "We found" refer to? This is not clear.

Next line: "in average" -> "on average"

P. 13, line 9: "on Fig." -> "in Fig."

Figure 6: spell out "eps" in Figure caption

P. 14, line 16: "on Fig." -> "in Fig." and word order needs to be adjusted

P. 15, line 30: "to the study" -> "for the study" or "for studying"

P. 15, line 32: "Osiris" -> "OSIRIS"

[Figure]

P. 15, line 34: "Gao et al."

Please cite the main SABER paper by Russell.

Same line: Please also cite a SCIAMACHY paper (Bovensmann and/or Burrows). It almost seems as if the authors avoid citing SCIAMACHY papers.

P. 16, line 12: "At each tangent point, the vertical resolution is 2.6 km"

That's the FWHM of the FOV, the vertical resolution is worse.

P. 16, lines 20 – 28: please show sample spectra and illustrate the correction procedure.

P. 17, line 5: Onion peeling is prone to noise, particulary lower down and is usually not the method of choice, but OK ..

P. 17, line 15: "radiuses" -> "radii"

P. 18, equation (21): this usually does not work well, but leads to unrealistic oscillations.

Section 3.2.2: is the model atmosphere divided into several angular segments in order to describe the attenuation within a given atmospheric layer properly? This doesn't seem to be the case and this should be stated explicitly, i.e. the technique applied is only an approximative treatment of the self-absorption.

P. 19, line 2: "each .. spectra" -> "each .. spectrum"

P. 19, line 17: "the more the lower latitude."

Why should it depend on latitude? Do you mean altitude? This part of the sentence is also incomplete.

P. 19, line 38: "at (lat" -> "(at lat"

P. 20, line 12: "Absorption by O2 may be computed in the nadir viewing geometry, though attenuation in this geometry is small (2% for the Q branch, less outside of the

[Figure]

Q branch)."

This is only valid for z > 30 km, right? I suggest mentioning this explicitly.

P. 20, line 20: "On Fig." -> "in Fig." and word order needs to be adjusted

P. 21, Caption Fig. 10, line 1: "form" -> "from"

Same Caption, line 4: "ADAPTEE" -> "ADAPTED"

P. 21, line 19: "On Fig." -> "in Fig." and word order needs to be adjusted

Same on line 29.

P. 22, Figure 11: Figure is truncated at bottom

Caption Fig. 11: "for ENVISAT orbit 20070101_1256."

This is not the orbit number. Please provide orbit # and date.

P. 23, line 6: "The ratio of spectra measured /model, Sobs/ Smod"

This phrase is sloppily, please be more precise.

P. 23, line 10: "validates completely"

Well, there is roughly a systematic 10% difference in the right panel of Figure 13, i.e. I would not speak of "complete" validation.

Caption Fig. 12: "Ratios of measurements/model of limb spectra"

Sloppy phrase, please be more precise.

P. 24, line 11: "(some limb scans do not reach low enough altitudes)."

What does this mean? Does it refer to the MLT measurements? Please clarify.

P. 24, line 14: "aerosols" –> "aerosols"

P. 24, line 14: "and pollutes the SCIAMACHY measurements"

This sounds like this is an instrumental problem, which is certainly not the case. Most of the SCIAMACHY limb data procucts are based on scattered radiation. Scattering does not generally "pollute" the limb measurements. I also suggest using a different word.

P. 24, line 23: "On Fig." -> "in Fig." and word order needs to be adjusted

P. 25, line 7: "On Fig." -> "in Fig." and word order needs to be adjusted

P. 26, Fig. 15: "Brightness = f(SZA.." and "Color scale -> latitude"

This should be mentioned in Figure caption, not overplotted onto the Figure. This seems like an unfinished figure from a presentation, unsuitable for a paper.

P. 28, line 21: "radiation, atmosphere + aerosols+ surface)."

Please form a sentence.

P. 28, line 28: equality sign in equation is subscript.

P. 29, line 10: "Fig." -> "Figs."

P. 31, line 27: "which is almost polar and descending"

As it is "which" appears to refer to "month", which doesn't make sense. Please adjust.

P. 31, line 29: "on the intensity" -> "for the intensity"

P. 32, line 1: "which is incorrect"

It is specified like that in the SCIAMACHY documentation, I think, i.e. it's not correct to state, that this is incorrect. It's not the natural choice, but it is as documented, I believe. Maybe I'm wrong.

P. 32, Figure 19: Order of panels not specified. Which is which?

Same Figure: the legends in the individual panels overlap. Overall, the quality of the figure not suitable for publication. Please improve. Also: It's essentially impossible to

separate the SCIA and the model symbols. Needs to be replotted.

P. 32, line 15: which local time was used for the model data?

P. 33, Figure20: increase spacing between legend lines.

P. 34, Figure 21: increase spacing between legend lines.

Caption Fig. 21: "Fig.20" -> "Fig. 20"

P. 37, line 2: "At night, GOMOS ozone profiles show a strong ozone depletion around 80 km"

I suggest using "minimum" rather than "depletion", because the phrase suggests that there is less ozone during night than during the day, which is not the case.

P. 38, line 1: "on Fig." -> "in Fig."

Same line: "observed on" -> "observed in"

P. 38, Caption Fig. 25: you need two GOMOS measurements for occultations with two different stars, right? Were both on the same day?

P. 39, line 1: "GOMOS ozone concentration vertical profiles show quite similar values below 60 km between day and night, and quite lower values of O3 at night above 60 km, a feature well understood from mesospheric chemistry."

?? Statement unclear. Do you mean the comparison of GOMOS with the model or the comparison of GOMOS night vs. day measurements? Nighttime O3 in the mesosphere is significantly larger than daytime O3 (photolysis during the day). The statement is not correct as is.

P. 39, line 18: "11h30 ascending node or 13h30 descending"

This is impossible. One of the two times is incorrect (at least).

Next line: "consists in" -> "consists of"

P. 40, line 12: "lands" -> "land"

P. 40, line 13: "over seas" -> "over lakes" ? (sea = ocean)

P. 40, line 18: "e.g" -> "e.g."

P. 40, line 22: "slope of albedo"

?? slope with respect to what? This is unclear

Same line: "for each bands" -> "for each band"

Same line: "aerosols properties" -> "aerosol properties"

P. 40, line 24: define "CAMS"

Next line: "Sentinel 2"

Which instrument on Sentinel 2?

Same line: please add reference for "PlanetObserver"

P. 41, last line: "scattered solar radiation by the surface"

There is also scattering by the atmosphere

P. 42, line 18: "to O2 absorption is a continuous function of the wavenumber"

What exactly does that mean? The ratio will certainly be a function of wavenumber.

P. 43, line 10: "then applied IT to"

P. 43, line 19: "spectral resolution" -> "resolving power"

P. 44, line 6: I suggest speaking of higher / lower temperature, not warmer / colder. Temperature cannot be warm or cold, strictly speaking.

P. 44, line 14: "spectrum.,"

2 lines below: "SCHIAMACHY" -> "SCIAMACHY"

P. 44, line 27: "spectru"

Next line: "spectel"

I assume this is not a typo, please explain.

P. 44, second line bottom-up: "resolution" -> "resolving power"

There are several more cases, where "resolution" is used rather than the correct "resolving power". Please search for them all and correct them.

P. 48, caption Fig. 31, line 3: "g" in the term symbol should be subscript.

P. 49, Caption Fig. 32: "SCHIAMACHY" -> "SCIAMACHY"

P. 49, line 7: "spectral resolution" (see comment above)

P. 49, line 16: "spectra resolution" (see comment above)

P. 50, line 6: "resolution power" (see comment above)

P. 50, line 25: "slope on albedo"

Unclear, specify.

P. 50, line 29: "Lmoy"

Please define. If this is french, please change to english.

P. 50, line 48: "which associated" -> "whose associated"

P. 51, line 11: "which peak" -> "whose peak"

P. 51, line 13: "These two spectra are then normalized at the intensity of the airglow spectrum that is put inside the simulated spectrum to invert."

I read this sentence several times, but didn't get it. Please rephrase.

Figure 33: "VER moy"

Replace by english term.

Figure 34: legends overlap with figures, please correct.

P. 53, line 22: "which intensity" -> "whose intensity"

Same line: move "respectively" after the numbers.

P. 54, caption Fig. 35, line 2: "value" -> "values"

P. 55, line 1: "spectral resolution" (see comment above)

P. 55, line 35: "non-ETL" -> "non-LTE"

2 lines below: "transition at 1.58 $\mu$m of the O2 (1Delta) around 1.58 $\mu$m"

2 lines below: "CO2 band" -> "CO2 bands"

P. 56, line 7: "nw" ??

Next line: "nw" ??

P. 56, line 19: "spectral resolution" (see comment above)

P. 57, line 10: "of 2007 SCIAMACHY 12,833 limb-scans"

Word order wrong.

Next line: "with though" -> "although with"

P. 57, line 16: "the O2* airglow is well organized"

What is this supposed to mean? Airglow is well organized?

P. 57, line 39: "on Fig." -> "in Fig."

P. 57, last line: "and the reflected solar flux is far to be"

This makes no sense.

P. 58, lines 10, 13, 20, 27, 30, 38: "resolution" (see comment above)

P. 58, line 32: "which broad" -> "whose broad"

Same line: "on Fig." -> "in Fig."

P. 58, line 34: "for a better constraining" -> "allow to constrain the .. absorption better"

P. 59, lines 1, 7, 9: "resolution" (see comment above)

P. 59, line 4: "Mission space mission"

Is the repetition intended?

P. 67, line 27: "From the pressure and temperature are also calculated the total density an"

Word order incorrect.

P. 71, line 2: "the loss term .. becomes very long" ?

Do you mean "large" rather than "long"? "Long" doesn't really make sense.

P. 72, line 13: "present" -> "presents"

P. 74, line 7: "On Fig. A6 (left) are represented the"

Word order incorrect.

Same comment on line 16 on the same page.

P. 74, line 8: comma missing in "40 S 40N"

---

## Author Comment (AC1) · 21 Sep 2019

Response to comments of Anonymous Referee #2: responses are in Arial character, with new text in blue, more visible in the attached .pdf file.

General comments: This paper deals with an interesting and relevant aspect, i.e. the contamination of nadir O2 measurements in the 1.27 micron IR atmospheric band caused by the corresponding airglow emission. The paper will be relevant for the community and the approaches employed in the study appear to be robust. However, there

are two major issues with the paper. It is unnecessarily long (which distracts from the main content) and it is full of typos, minor inconsistencies and little errors. Many sections make the impression of hastily written drafts that were not proof read. It took me two full days to go through this manuscript, which is not acceptable. We expect from young scientists and postdocs that manuscripts are in tip-top shape and it should also be expected from senior scientists. It is certainly not the reviewer's task to correct all the mistakes. Please correct the manuscript carefully.

General Response to general comments. We are very much grateful for the great amount of time spent by Referee #2 in reading carefully our manuscript. Not only he (she) identified a large number of english-language typos or mistakes; he (she) also identified sentences that seemed to be obvious for the authors, but whose meaning is not obvious for an outside reader. These sentences were corrected with a more appropriate rephrasing. We apologize for not being english-language born scientists. We hope there is still some room in science for those scientists.

Length of the paper: We recognize that this paper is long, but we still believe that its overall length is deserved. At an early stage one reviewer suggested to split the paper in several papers but we have been quite reluctant to continue along this line (split or shorten substantially) for the following reasons. All parts of the paper are relevant to the same subject: is it possible to use the O2 1.27 $\mu$m absorption band for CO2 mixing ratio retrieval, in spite of the strong airglow contamination? The team that was assembled for this scientific research had to cover several scientific aspects: our understanding of this airglow, building a model for the intensity, and a model for the spectral shape, validation with comparisons with SCIAMACHY/ENVISAT data, separation of airglow from absorption. One reader is not obliged to read carefully all sections, he can pick up what he is most interested in. We estimate that if we would split our paper into two papers, the overall total length of the two papers would be longer than the present version, because of unavoidable repetitions (each paper must be self-consistent, including references). It would require also twice more reviewers and Editor work. AMT

stands for Atmospheric Measurements Techniques and therefore our paper is perfectly in scope with the profile of the publication. Our paper is long because it is deliberately rather detailed, because we wish to ease the possibility that anybody else to be able to reproduce our results.

Remember that the results of about 30% of all scientific papers cannot be reproduced by other scientists, and this comes to 50% of papers in biology, a very embarrassing situation. One great advantage of AMT publication is that it does not require paper printing, therefore cancelling a source of CO2 production. Only an interested reader would potentially print it. Therefore, with AMT we may reconcile CO2 economy and detailed description for better reproducibility of results. Finally, we note that the length of the paper did not discourage a fairly large number of scientists to download the paper when discussed in AMTD: The paper has been viewed HTML 175 times and the pdf downloaded 91 times (25 august 2019), about half from the US.

Response to specific comments:

P. 1, line 18: "i.e. OCO-2 .. " -> "e.g. OCO-2 .."; otherwise all instruments carrying out these measurements should be listed. Done.

Thank you for the correction, done. Google says indeed: "When you mean "for example," use e.g. It is an abbreviation for the Latin phrase exempli gratia. When you mean "that is," use "i.e." It is an abbreviation for the Latin phrase id est."

P. 1, line 38: "model underestimate" -> "model underestimates" Done Same line: "This is fully confirmed .." It is not entirely clear, what this refers so. Is it confirmed that the model underestimates by 15%? Or that the airglow intensity is mainly determined by the SZA. Please clarify. answer: We have added the sentence: ", and found that the nadir SCIAMACHY intensities are mostly dictated by SZA and larger than the model intensities by a factor âĹij1.13" P. 2, line 13: "Dioxyde" ->"dioxide"Done P. 2, line 14: "The atmospheric fraction .." I suggest being more precise here. Mention CO2 explictly and that the fraction is a mass ratio – that's what it is, right? answer: we have rewritten

the sentence: "The atmospheric fraction is the ratio of the atmospheric increase of CO2 mass to the mass of CO2 anthropogenic emission." P. 2, line 30: "The first satellites to be launched with the aim .." Why don't you list SCIAMACHY/Envisat? SCIAMACHY is certainly not as specialized to CO2 retrievals as the other instruments, but it was also built to measure CO2 and it should be listed here. The CO2 retrievals were also quite successful. answer: we have added : ENVISAT (ESA) with SCIAMACHY instrument,

P. 3, line 13: "mission" -> "missions" Done

Same line: "Airglow has a spectrum that is very similar .." This statement is not generally true for all airglow emissions occuring in the atmosphere and should be phrased more precisely. There are many other airglow emissions apart from O2, for which the statement is not valid. Rephrased: "Oxygen airglow at 1.27 $\mu$m has a spectrum..."

P. 3, Figure 1: the figure shows O3 photolysis as the only source of the 1.27 micron emission. Ozone photolysis is only one of several excitation mechanisms. I suggest stating in the caption that it is the main mechanism on the dayside. the following sentence was added to caption for Figure 1 : Ozone photolysis indicated in the figure is the main source of O2 airglow at 1.27 $\mu$m, but not the only one.

Caption Fig. 1, line 3: "are crossing" -> "is crossing" Done Same line: "which emission" -> "whose emission" Done P. 4, line 6: "confirms" -> "confirmed" Done P. 4, line 9: "mission" -> "missions" Done P. 4, line 12: "to determine from nadir viewing observations the CO2 vertical columns" Word order wrong. Please replace by: "to determine CO2 vertical columns and mixing ratios from nadir viewing observations" We have rewritten : "...to determine CO2 vertical columns and mixing ratios from nadir viewing observations (which needs associated O2 columns), Kuang et al. ... P. 4: "secular variation" -> "secular variations" Done P. 4, line 21: "plugged to a" -> "combined with a" Done P. 4, line 22: "We also note that the TCCON ground-based spectrometer array, observing the sun, uses this 1.27 $\mu$m band to derive the CO2/dry air mixing ratio" This is the third time this is mentioned and can be removed. Answer: Indeed, it is redundant.

However, here we explain why TCCON is selecting the 1.27 $\mu$m band rather than the A band. If we delete the sentence as proposed, then the following sentence would be difficult to understand. Therefore, we have re-arranged the sentence a little. P. 4, line 32: "intimately intricated." Grammar wrong, please correct. Done Replaced by "closely blended" P. 4, line 38: "Second, the transmittance Tr=exp(-_ ) saturates at high optical thicknesses _>1,while the emission does not." I don't understand this statement and think it's wrong. Emission is certainly also limited if the optical depth becomes quite large. What you probably mean is that extinction is not important for the emission, because the emission mainly occurs above 30 km. But this is not what the sentence states. answer: No we do not mean what you suggest. We rephrased for more clarity: "Second, the emission at 1.27 $\mu$m increases linearly with the column of O2* at all wavelengths (re-absorption by O2 is negligible at emission altitude), resulting in a constant relative shape of the emission spectrum, while the absorption spectrum is not linear: the transmittance Tr=exp($-\tau$) saturates at high optical thicknesses of O2 $\tau$>1,and the absorption spectral shape is not constant but depends on the air-mass factor"

P. 5, line 1: "on Fig." -> "in Fig." Done P. 5, line 3: "which positions" -> "whose positions" Done P. 5, line 5: "rending" ?? Do you mean "rendering"? Word order is also wrong: "rendering this proposal unpractical" yes, correction done as you suggest. Done P. 5, line 7: "contributes" -> "contribute" Done Caption Fig. 2, line 1: "The transmittance within an individual O2 line (red) is much larger than" "transmittance .. much larger" is not well phrased. The transmittance is zero in the center of the line. Please be more precise. Answer : we agree that the meaning of Fig.2 caption is not clear at all. The caption of figure has been rephrased. "Comparison at high spectral resolution of spectral shape of atmospheric O2 transmission (transmittance) and spectral shape of O2* emission. The FWHM of an individual O2 line (red) is much larger than the FWHM of its counterpart in emission (black line), allowing in principle to disentangle absorption from emission at selected wavelengths. The channels recommended..." P. 5, line 22: "compared" -> "compare"Done Same tense as in previous sentences P. 6, line 1: "while in Section 6 are detailed the accuracy and bias results of" Word order

incorrect Corrected. P. 6, line 4: "In section 7 are examined briefly" Word order wrong: "In section 7 some other cases . . . are examined"Done P. 6, line 20: suggest to define Rayleigh the first time it is used. Done P. 7, line 3: "strong solar light scattered component" -> "strong contribution of scattered solar radiation" P. 7, line 5: "From high altitude the O2 absorption will be a little bit reduced." This is incomprehensible? Do you mean that the absorption is weaker at higher altitudes? Answer: no. Rephrased for more clarity: "From a ground based observing station located at high altitude the O2 absorption will be a little bit reduced."

P. 7, line 26: please cite Bovensmann and/or Burrows here. For all the other instruments you provide a citation, but not for SCIAMACHY. Done P. 7, line 37: "On Fig. 3 (from Khomich et al., 2008) are represented the various electronic" "On Fig." -> "In Fig" and the word order is incorrect: "Fig. 3 presents the various .." Done P. 7, last line: "on Fig." -> "in Fig." Done P. 8, line 2: "on Fig." -> "in Fig." Done P. 8, equation (4): "C" -> "c" Done Same equation: the units are incorrect, i.e. the equation is not valid as is. Please correct. Corrected. P. 8, line 18: "Fraunhofer" -> "Fraunhofer's" Done P. 9, line 24: "solar effect" One can tell what you mean, but it is not well phrased, too unspecific. Replaced by Âń solar photolysis of various species" P. 9, line 31: "emitted photon" -> "emitted photons" Done P. 9, equation (5): the middle part of the equation is incorrect. During the day, there will essentially be a steady state, i.e. O2* is produced by O3 photolysis (mainly) and removed by emission, i.e. d[O2*]/dt = 0. Answer : You are prefectly right ; the middlepart of the equation (5) has been deleted. P. 10, title section 2.3.3: remote (remove ?) period from section title Done P. 10, line 15: "A –coefficients" -> "A-coefficients" Done P. 10, line 16: "second members" Unclear, what you mean. The sentence is rephrased for more clarity : "...second members of equation (6) for all allowed transitions Li, giving the rate of emission of the corresponding spectral line VER(Li)".

P. 10, line 34: please provide k_B in SI units Done kB =1.38065 x10-23 joule K-1 C5 P. P. 11, line 4: "on all" -> "over all" (2 occurrences) Done P. 11, line 11: "on Fig." ->

"in Fig." Done P. 11, line 12: "term by" -> "term and" Done P. 11, caption Fig. 4, line 3: "There are 5, 7, or 8 values (and transitions) for each black circle on the figure" Not clear to me, why there are 5, 7 or 8 values for each black circle. Please explain. Answer : This is a consequence of the fact that HITRAN contains only the transitions which have a strength above a certain threshold.. Added in the caption : Âń . . . in the figure, present in the HITRAN list, because weak lines (below a certain threshold) are not in HITRAN." Same line: "on the figure" -> "in the figure" Done P. 11, line 21: "On Fig. 5 are represented the various energy" "On" and word order incorrect: "Fig. 5 presents .." Done P. 12, Title section 2.3.4: add space at beginning Done P. 12, line 6: "sate" -> "state" Done Next line: "sum on" -> "sum over"Done P. 12, line 21: "We found that the total decay rate is A21tot=2.29 10-4 s-1." Above you determined the total decay rate to be 2.22 x 10-4 s-1. What does "We found" refer to? This is not clear. answer : rephrased for better clarity : We found from the HITRAN data that the total decay rate is A21tot=2.29 10-4 s-1, slightly different from 2.22 10-4 s-1 derived from the rounded value 75 mn of the lifetime quoted by Lafferty et al. (1998). We may compute the lifetime 1/ A21tot =4367 sâĹij 73 mn. Next line: "in average" -> "on average" Done P. 13, line 9: "on Fig." -> "in Fig." Done Figure 6: spell out "eps" in Figure caption Done. Caption has been modified for more clarity. P. 14, line 16: "on Fig." -> "in Fig." and word order needs to be adjusted Done P. 15, line 30: "to the study" -> "for the study" or "for studying" Done P. 15, line 32: "Osiris" -> "OSIRIS" Done C6 P. 15, line 34: "Gao et al." Please cite the main SABER paper by Russell. Done Same line: Please also cite a SCIAMACHY paper (Bovensmann and/or Burrows). Done. There was already a reference in the following paragraph. It almost seems as if the authors avoid citing SCIAMACHY papers. This is a certainly a wrong impression ! On the contrary we emphasize the importance of SCIAMACHY in the problem of O2* emission and make a heavy use of these excellent measurements. P. 16, line 12: "At each tangent point, the vertical resolution is 2.6 km" That's the FWHM of the FOV, the vertical resolution is worse. Corrrection done : sentence rephrased : Âń . . .the FWHM of the FOV is 2.6 km (with a somewhat coarser vertical resolution). . . P. 16,

lines 20 – 28: please show sample spectra and illustrate the correction procedure. answer: the explanation of correction procedure is now illustrated by four Figures put in an Appendix B for convenience, reproduced at the end of this document. Done P. 17, line 5: Onion peeling is prone to noise, particulary lower down and is usually not the method of choice, but OK .. P. 17, line 15: "radiuses" -> "radii" Done. P. 18, equation (21): this usually does not work well, but leads to unrealistic oscillations. Section 3.2.2: is the model atmosphere divided into several angular segments in order to describe the attenuation within a given atmospheric layer properly? This doesn't seem to be the case and this should be stated explicitly, i.e. the technique applied is only an approximative treatment of the self-absorption. Answer : In the standard onion-peeling technique, the atmosphere is divided into spherical shells where the density (or emissivity ) is assumed to be constant. Each LOS is divided into segments which belong to various spherical shells. In this way, the vertical inversion is reduced to a linear system of equation (our equation (21) with a unique solution. Of course, this is an approximation, but we cannot do better with the finite sampling of the LOS. Adding the attenuation by O2 does not introduce any additional approximation. P. 19, line 2: "each .. spectra" -> "each .. spectrum" Done. P. 19, line 17: "the more the lower latitude." Done. Why should it depend on latitude? Do you mean altitude? This part of the sentence is also incomplete. altitude, of course !!! rephrased : Âż.. they must underestimate their emissivities more and more with lower altitudes.." P. 19, line 38: "at (lat" -> "(at lat" Done. P. 20, line 12: "Absorption by O2 may be computed in the nadir viewing geometry, though attenuation in this geometry is small (2% for the Q branch, less outside of the Q branch)." This is only valid for z > 30 km, right? I suggest mentioning this explicitly. answer : yes. We added (altitude z>30 km) P. 20, line 20: "On Fig." -> "in Fig." and word order needs to be adjusted Done. P. 21, Caption Fig. 10, line 1: "form" -> "from" Done. Same Caption, line 4: "ADAPTEE" -> "ADAPTED" Done. P. 21, line 19: "On Fig." -> "in Fig." and word order needs to be adjusted Same on line 29. Done. P. 22, Figure 11: Figure is truncated at bottom Corrected with original figure not truncated Caption Fig. 11: "for ENVISAT orbit 20070101_1256." Done This is not

the orbit number. Please provide orbit # and date. answer : we rephrased the caption : . . . ENVISAT orbit 25293, starting 1th January 2007 at 01h 12mn.

P. 23, line 6: "The ratio of spectra measured /model, Sobs/ Smod" This phrase is sloppily, please be more precise. sentence rwritten as : Âń The ratio of measured spectra /model spectra, Sobs/ Smod . . ." P. 23, line 10: "validates completely" Well, there is roughly a systematic 10% difference in the right panel of Figure 13, i.e. I would not speak of "complete" validation.

We have rephrased : "This comparison validates completely the approach that we developed in Section 2, except that the overall level of the ratio is slightly below 1." and we explain the reason for this exception right away. Caption Fig. 12: "Ratios of measurements/model of limb spectra" Sloppy phrase, please be more precise. Done. P. 24, line 11: "(some limb scans do not reach low enough altitudes)." What does this mean? Does it refer to the MLT measurements? Please clarify. sentence repharsed : Âń . . . containing 12,400 limb scans in the normal mode which go down sufficiently for our purpose (some limb scans do not reach low enough altitudes to allow retrieval of the full VER profile above 30 km)."

P. 24, line 14: "aerosols" –> "aerosols" ??? aerosol P. 24, line 14: "and pollutes the SCIAMACHY measurements" C8 This sounds like this is an instrumental problem, which is certainly not the case. Most of the SCIAMACHY limb data procucts are based on scattered radiation. Scattering does not generally "pollute" the limb measurements. I also suggest using a different word. this sentence has been rephrased : (the useful signal for SCIAMACHY limb mode ozone retrieval) which dominates over the O2* radiance. P. 24, line 23: "On Fig." -> "in Fig." and word order needs to be adjusted Done. P. 25, line 7: "On Fig." -> "in Fig." and word order needs to be adjusted Done. P. 26, Fig. 15 16: "Brightness = f(SZA.." and "Color scale -> latitude" This should be mentioned in Figure caption, not overplotted onto the Figure. This seems like an unfinished figure from a presentation, unsuitable for a paper. Done. answer : The figure 16 has been redone, and one sentence was added in the caption : The

airglow brightness is mostly correlated with SZA. P. 28, line 21: "radiation, atmosphere + aerosols+ surface)." Please form a sentence. sentence rephrased : Ân ... of the solar radiation back-scattered by the gaseous atmosphere, aerosols and the surface). P. 28, line 28: equality sign in equation is subscript. Done P. 29, line 10: "Fig." -> "Figs." Done P. 31, line 27: "which is almost polar and descending" As it is "which" appears to refer to "month", which doesn't make sense. Please adjust. Done P. 31, line 29: "on the intensity" -> "for the intensity" Done P. 32, line 1: "which is incorrect" It is specified like that in the SCIAMACHY documentation, I think, i.e. it's not correct to state, that this is incorrect. It's not the natural choice, but it is as documented, I believe. Maybe I'm wrong. answer : You are right. The definition of SZA extracted from the document "SCIAMACHY Command Line Tool Software User's Manual (SUM) for SciaL1c" : "Solar zenith angles of the start, middle and end of the integration time at TOA". We find the same definition in the "Sciamachy Level-1b IODD" Therefore, we have rewritten this paragraph: "Note regarding the SZA of the SCIAMACHY data: We noted that the SZA value provided in the SCIAMACHY ESA products in limb viewing, as defined in the data product, is the SZA of one of the two points corresponding to the intersection between the LOS and TOA (Top of Atmosphere defined at 100 km altitude). But what we need is the SZA of the tangent point of the line of sight (LOS), which is different. Therefore, we systematically calculated the SZA at the tangent point of the SCIAMACHY LOS using an external tool (IDL routine). All results presented in this report are obtained using this recalculated SZA."

P. 32, Figure 19: Order of panels not specified. Which is which? Same Figure: the legends in the individual panels overlap. Overall, the quality of the figure not suitable for publication. Please improve. Also: It's essentially impossible to C9 separate the SCIA and the model symbols. Needs to be replotted.

The figures have been replotted for improvement. P. 32, line 15: which local time was used for the model data? answer : We do not use Ân local time Âż here. We use UT time (Universal Time) of the measurement and in the model. P. 33, Figure20: increase

spacing between legend lines. Done P. 34, Figure 21: increase spacing between legend lines. Done Caption Fig. 21: "Fig.20" -> "Fig. 20" Done P. 37, line 2: "At night, GOMOS ozone profiles show a strong ozone depletion around 80 km" I suggest using "minimum" rather than "depletion", because the phrase suggests that there is less ozone during night than during the day, which is not the case. answer : Done .we have changed the Âń depletion Âż by Âń minimum Âż.

P. 38, line 1: "on Fig." -> "in Fig." Done Same line: "observed on" -> "observed in" Done P. 38, Caption Fig. 25: you need two GOMOS measurements for occultations with two different stars, right? Were both on the same day? Done answer: yes, two different stars. The left occultation is star S005 observed on 5th August 2007 22h13mn (orbit number 28397) at SZA=38° and the right occultation is star S008 observed on 15 February 2007 16h36mn (orbit number 25946) at 94°. Although the high tmosphere is still illuminated, the photochemical situation is mabiguous, and this figure has been deleted, keeping only one occultation. The caption of Fig 25 (now figure 24) has been modified accordingly: Figure 24:Left: One example of comparison of a GOMOS vertical dayside ozone profile (black curve) vs. REPROBUS prediction (red curve). The GOMOS profile was observed on August 5 2007, at an SZA angle of 38°. Right: relative difference (GOMOS-REPROBUS)/ REPROBUS .

P. 39, line 1: "GOMOS ozone concentration vertical profiles show quite similar values below 60 km between day and night, and quite lower values of O3 at night above 60 km, a feature well understood from mesospheric chemistry." ?? Statement unclear. Do you mean the comparison of GOMOS with the model or the comparison of GOMOS night vs. day measurements? Nighttime O3 in the mesosphere is significantly larger than daytime O3 (photolysis during the day). The statement is not correct as is. answer : This was a typo in the submitted manuscript. Nighttime O3 above 60 km is indeed larger than during the day, and this is well observed by GOMOS. We changed "quite lower" for "larger" in the revised version.

P. 39, line 18: "11h30 ascending node or 13h30 descending" This is impossible. One

of the two times is incorrect (at least). Answer: yes, it is possible, because MicroCarb is not yet launched ! So, this is one or the other. We have rephrased : Âń . . .11h30 ascending node or alternately13h30 descending node helio-synchronous orbit (to be decided later)."

Next line: "consists in" -> "consists of" Done C10

P. 40, line 12: "lands" -> "land" Done P. 40, line 13: "over seas" -> "over lakes" ? (sea = ocean) Done P. 40, line 18: "e.g" -> "e.g." Done P. 40, line 22: "slope of albedo" ?? slope with respect to what? This is unclear : wavelength slope Same line: "for each bands" -> "for each band" Done Same line: "aerosols properties" -> "aerosol properties Âż Done P. 40, line 24: define "CAMS" : (Copernicus Atmosphere Monitoring Service) Next line: "Sentinel 2" Which instrument on Sentinel 2? answer : There is only one instrument on board Sentinel 2 : MSI (Multi Spectral Instrument). We have added the name of the instrument and the sentence : (from the Multi Spectral Instrument MSI, the unique instrument on-board Sentinel-2)

Same line: please add reference for "PlanetObserver" answer : we have added the reference : (https://www.planetobserver.com/products/planetdem/planetdem-30/) P. 41, last line: "scattered solar radiation by the surface" There is also scattering by the atmosphere we have added the sentence : Âń The contribution of atmospheric Rayleigh scattering is small at this wavelength and ignored in this exercise." P. 42, line 18: "to O2 absorption is a continuous function of the wavenumber" What exactly does that mean? The ratio will certainly be a function of wavenumber. answer : It means that it can be computed for any wavenumber, independently of the existence of any spectral line. We have modified the sentence : "...the ratio of $O_2^*$ emission to $O_2$ absorption is not a constant, but a changing continuous function..." P. 43, line 10: "then applied IT to" Done P. 43, line 19: "spectral resolution" -> "resolving power" Done P. 44, line 6: I suggest speaking of higher / lower temperature, not warmer / colder. Temperature cannot be warm or cold, strictly speaking. answer : for temperatures, we change to high or low ; for spectra or components, we keep warm and cold, it has some meaning

in the mesosphere. P. 44, line 14: "spectrum.," Done 2 lines below: "SCHIAMACHY"
-> "SCIAMACHY" Done

P. 44, line 27: "spectru" Done Next line: "spectel" I assume this is not a typo, please
explain. answer: The word spectel is commonly used to designate a spectral element,
which may be contituted of several pixels in 2D imaging spectrometers. we have added
: : Âż. . . each spectel (spectral element) is. . . P. 44, second line bottom-up: "resolution"
-> "resolving power" Done There are several more cases, where "resolution" is used
rather than the correct "resolving power". Please search for them all and correct them.
P. 48, caption Fig. 31, line 3: "g" in the term symbol should be subscript. Done P.
49, Caption Fig. 32: "SCHIAMACHY" -> "SCIAMACHY" Done P. 49, line 7: "spectral
resolution" (see comment above) Done P. 49, line 16: "spectra resolution" (see com-
ment above) Done P. 50, line 6: "resolution power" (see comment above) Done P. 50,
line 25: "slope on albedo" Unclear, specify. changed to Âń spectral slope of albedo
Âż P. 50, line 29: "Lmoy" we rephrased : ". . . a median intensity luminance Lmoy. . ."
Please define. If this is french, please change to english. Answer :Change refused.
Lmoy corresponds to a number of MicroCarb official documents and is a standard
name designating a reference luminosity case of about median intensity. Remenber
Âń bremstrhalung Âż, Âń gedanken experiment Âż. . . We have added to the explana-
tion of Lmoy : This reference luminance value Lmoy corresponds P. 50, line 48: "which
associated" -> "whose associated" Done P. 51, line 11: "which peak" -> "whose peak"
Done P. 51, line 13: "These two spectra are then normalized at the intensity of the air-
glow spectrum that is put inside the simulated spectrum to invert." I read this sentence
several times, but didn't get it. Please rephrase. We have added some rephrasing :
"These two spectra are then normalized to the intensity of the airglow spectrum that
is put inside the simulated spectrum which we wish to invert. The model spectrum
that we wish to best approximate this simulated spectrum will be a linear combination
of these two normalized spectra, with a sum of coefficients near unity, which is more
convenient for the description of the mesosphere." Figure 33: "VER moy" Done C12
Replace by english term. Figure 34: legends overlap with figures, please correct. Done

P. 53, line 22: "which intensity" -> "whose intensity" Done Same line: move "respectively" after the numbers. Done P. 54, caption Fig. 35, line 2: "value" -> "values" Done P. 55, line 1: "spectral resolution" (see comment above) Done P. 55, line 35: "non-ETL" -> "non-LTE" Done 2 lines below: "transition at 1.58 $\mu$m of the O2 (1Delta) around 1.58 $\mu$m" Done 2 lines below: "CO2 band" -> "CO2 bands" Done P. 56, line 7: "nw" ?? Next line: "nw" ?? answer : nw is the official abbreviation of nanowatt. P. 56, line 19: "spectral resolution" (see comment above) Done P. 57, line 10: "of 2007 SCIAMACHY 12,833 limb-scans" Word order wrong. changed to Âń 12,833 limb-scans acquired by SCIAMACHY in 2007..." Next line: "with though" -> "although with" Done P. 57, line 16: "the O2* airglow is well organized" What is this supposed to mean? Airglow is well organized? Rephrased : "... the intensity of the O2* airglow is well organized (with a weak horizontal variability), and quite predictable, P. 57, line 39: "on Fig." -> "in Fig." P. 57, last line: "and the reflected solar flux is far to be" This makes no sense. Answer : The sentence has been modified. Also, while for the shape of the airglow emission, all the spectral lines are proportional to each other, on the contrary the radiance factor (=$\pi$ B/solar flux cos(SZA), B brightness) is modulated by the O2 transmittance spectrum (Tr($\tau$)=exp(-$\tau$)) which is not linear for the strong lines with large $\tau$. P. 58, lines 10, 13, 20, 27, 30, 38: "resolution" (see comment above) Done C13 P. 58, line 32: "which broad" -> "whose broad" Done Same line: "on Fig." -> "in Fig." Done P. 58, line 34: "for a better constraining" -> "allow to constrain the .. absorption better" Rephrased : Âń ... would be measured and would allow to constrain the CIA absorption better.." P. 59, lines 1, 7, 9: "resolution" (see comment above) Done P. 59, line 4: "Mission space mission" Is the repetition intended? Answer : Yes, we have seen documents with the name CO2 Mission. Rephrased : The CO2 Mission (space mission CO2-M).. P. 67, line 27: "From the pressure and temperature are also calculated the total density an" Word order incorrect. Done P. 71, line 2: "the loss term .. becomes very long" ? Do you mean "large" rather than "long"? "Long" doesn't really make sense. answer : we have rephrased the sentence : Âń ... when the solar photolysis rate of O2 varies significantly over the natural lifetime ($\sim$75 mn) of the O2(a1$\Delta$) excited state, d[O2*]/dt is $\neq$

0, as it is the case at very large zenith angles with d[O2*]/dt < 0 at dusk and d[O2*]/dt >0 at dawn (Fig. 31).

P. 72, line 13: "present" -> "presents" Done P. 74, line 7: "On Fig. A6 (left) are represented the" Word order incorrect. Done Same comment on line 16 on the same page. Done P. 74, line 8: comma missing in "40 S 40N" Done

Appendix B: Processing of SCIAMACHY Level-1c radiance data Here we show some figures describing our processing of Level-1c SCIAMACHY radiance data, as explained in Section 3.1. , in order to get a "pure" radiance spectrum of the O2* airglow.

Figure B1. This high altitude spectrum recorded above 105 km contains some residual spectral (readout) patterns left from the calibration step and is subtracted from all measurements obtained at lower altitude in the same scan limb.

Figure B2. Spectra corrected from high altitude spectrum showing still two bad pixels at wavelength 1262.267 nm and 1282.128 nm. We replaced their value by the average of their two surrounding pixels to obtain spectra of Figure B3. The tangent altitude of the LOS is colour coding each spectrum.

Figure B3: Same as Figure B2, after correction of the two bad pixels. In addition to the O2* airglow, there is the radiance of solar light scattered by air and aerosols, increasing when tangent altitude is decreasing. In addition, the strong absorption of O2 in the 1.27 $\mu$m band becomes obvious at lowest altitudes.

Figure B4. Airglow spectra obtained from Figure 4 by subtracting a linear interpolation based on the two constant values of the continuum (one on each side) estimated from the median value of all points outside the O2* band. This correction is valid above $\sim$20 km of tangent altitude.

Please also note the supplement to this comment:
https://www.atmos-meas-tech-discuss.net/amt-2019-54/amt-2019-54-AC1-

supplement.pdf

---

## Author Comment (AC2) · 21 Sep 2019

Response to referee#4 O2 paper amt-2019-54.docx Interactive comment on "The use of O2 1.27$\mu$m absorption band revisited for GHG monitoring from space and application to MicroCarb" by Jean-Loup Bertaux et al. Anonymous Referee #4

General Response to general comments. We are very much grateful for the great amount of time spent by Referee #4 in reading carefully our manuscript and pointing out some mistakes that are now corrected. We write in blue our answers. General Comments: I believe that the authors intend to show in the submitted manuscript that

[Figure]

O2(1_g) airglow can be modeled with sufficient accuracy to use the 1.27 $\mu$m O2 absorption band to retrieve O2 columns for greenhouse gas studies. answer: No, we did not show that the subtraction of a model would provide sufficient accuracy to retrieve the O2 columns. On the contrary, we wrote at end of Section 8.1: "However, the degree of accuracy that is needed for the determination of Psurf for useful measurements of GHG gases is very large, about $\sim$ 0.1 hPa for the bias and 1 hPa for random error. According to our simulations, and if the airglow is ignored in the inversion but subtracted from a model, this airglow intensity model would have to be accurate to $\sim$1.5% (for a mean radiance with albedo= 0.2) to achieve the 1 hPa random error. Therefore, in most cases it is insufficient to rely entirely on a model to predict the actual airglow intensity to be subtracted from an observation. We need to disentangle in the observed spectrum itself the contribution of the airglow and the contribution of the solar scattered radiation. For this, we will rely on the fact that the spectrum shape of the O2* airglow is different from the O2 absorption spectrum. " Our purpose in this paper was also to fight wrong ideas about this 1.27 $\mu$m, like :"it is highly erratic and variable", "we do not fully understand the physical process of the emission", etc. We showed that the spectral shape predicted by our new model coincides with the SCIAMACHY observations, and that the model intensities are lower than SCIAMACHY by about 10-15%, suggesting that ozone is underestimated in our CTM model. But we show that with a good spectral resolving power, the airglow emission may be disentangled from the O2 absorption in nadir viewing.

Length of the paper: We recognize that this paper is long, but we still believe that its overall length is appropriate. At an early stage one reviewer suggested to split the paper in several papers but we have been quite reluctant to continue along this line (split or shorten substantially) for the following reasons. All parts of the paper are relevant to the same subject: is it possible to use the O2 1.27 $\mu$m absorption band for CO2 mixing ratio retrieval, in spite of the strong airglow contamination? The team that was assembled for this scientific research had to cover several scientific aspects: our understanding of this airglow, building a model for the intensity, and a model for the
spectral shape, validation with comparisons with SCIAMACHY/ENVISAT data, separation of airglow from absorption. One reader is not obliged to read carefully all sections, he can pick up what he is most interested in. We estimate that if we would split our paper into two papers, the overall total length of the two papers would be longer than the present version, because of unavoidable repetitions (each paper must be self-consistent, including references). It would require also twice more reviewers and Editor work. AMT stands for Atmospheric Measurements Techniques and therefore our paper is perfectly in scope with the profile of the publication. Our paper is long because it is deliberately rather detailed, because we wish to ease the possibility that anybody else to be able to reproduce our results. The spirit of AMT, with public discussions before final publications, is in line with the "open source" philosophy. Cancelling parts of the paper would jeopardize this philosophy. Remember that the results of about 30% of all scientific papers cannot be reproduced by other scientists, and this comes to 50% of papers in biology, a very embarrassing situation. One great advantage of AMT publication is that it does not require paper printing, therefore cancelling a source of CO2 production. Only an interested reader would potentially print it. Therefore, with AMT we may reconcile CO2 economy and detailed description for better reproducibility of results. In its present form, our paper is somewhat "self-consistent" on its subject. It will serve as a reference, not only for the MicroCarb project, but also on other future GHG monitoring space projects that may consider the use of the 1.27 $\mu$m band. Finally, we note that the length of the paper did not discourage a fairly large number of scientists to download the paper when discussed in AMTD: The paper has been viewed HTML 175 times and the pdf downloaded 91 times (25 august 2019), about half from the US. If the final version were cut significantly, it would introduce an advantage to those who uploaded the early version versus those seeing only the final version.

It seems that the authors have done a lot of very good work over the past 3 years or so, and a lot of it appears in this manuscript. The paper is very long at 75 pages, and I am not sure that all this detail needs to be in the paper, as some of it is in the published literature (O2 spectroscopy and non-LTE calculations, for example) and

references might suffice. I feel that the paper might be easier to follow and make a stronger case for the conclusion if content in the main body were limited to what is needed to support the conclusion, and use references or supplemental material otherwise. answer: to our knowledge, the detailed shape of the O2* airglow and the principles of its calculation have not be published elsewhere, except perhaps Sun et al. (2018) who have most likely achieved similar results independently, but did not explain all the details in the short format of their GRL paper. 1. The theoretical development (to obtain a theoretical spectrum of the O2* airglow emission) that we present here was done in 2017, and completely independently from the work of Sun et al. (2018). If we were following the suggestion to just quote the equations of Sun et al. (2018) and not present our own analysis, it would give to the reader the false impression that we have followed blindly the developments of Sun et al. 2018, which is not true. The fact that both groups have developed the same kind of theory (form the same theoretical approach based on what can be found in Simeckova et al. (2006)) re-inforce the credibility of this approach, which is very important for "hundreds of millions dollars space projects". 2. Our equations (14) and (15) give an original result: a formula giving the wavenumber variation over the whole band of the ratio $\varepsilon$/SS of airglow emission $\varepsilon$ to the line strength SS found in Hitran table. This formula is NOT in the paper of Sun; the referee#1 had to check by some manipulation of equations that such a relation COULD be retrieved form formulas in Sun et al. (2018). Doing this, referee#1 shows that both groups are using consistent descriptions of the physics involved, a satisfactory piece of information for which we must thank him. 3. In fact, in the present study, we have used the formulation of this ratio in order to build very simply a synthetic spectrum of the airglow emission, by using the LBLRTM code computing the local absorption, and multiplying by the function $\varepsilon(\nu)$/SS$(\nu)$ (14) and (15). This is a totally original method, and we wish that all the AMT readers to be able to reproduce it and use it. This is why all equations establishing $\varepsilon(\nu)$/SS$(\nu)$ must be kept in the present paper.

More specific suggestions follow below. I thought it might also help to include some discussion regarding how the O2 column retrievals using the 1.27 $\mu$m band will be

validated, given the very high accuracy ( 0.01%) that is required. Will they be compared against the O2 A-band retrievals? But if O2 A band retrievals are good enough to be the standard, then what is the benefit of switching to the 1.27 $\mu$m band, given the added complication of the airglow correction? How will one know if the new retrievals are better? Answer: we quote from our paper: "Kuang et al. (2002) recognized the virtues of the O2 band at 1.27 $\mu$m (nearest to the CO2 bands), but discarded its use because it is contaminated by the intense O2 airglow day side emission". Therefore, the O2 A band at 0.76 $\mu$m was taken "by default". The whole idea of this investigation is to revisit the rejection of the O2 band at 1.27 $\mu$m, recognized to be much better than the O2 A band in case of aerosols, but only if it can be corrected from the dayglow. The problem of validation is important, and is not addressed here. MicroCarb is using both bands of O2, therefore allowing useful comparisons. One criterium would be the retrieval of Psurf, which is known from meteo fields. Also the difference between the two retrievals of Psurf (from the two bands of O2) could be correlated with the quantity of aerosols. We suspect that this difference will increase with the quantity of aerosols, because in many instances with aerosols the O2 A band (0.76 $\mu$m) underestimates Psurf when compared to the meteo field. But this discussion is well beyond the scope of this paper.

Specific Comments: I found the papers by Zarboo et al. (2018), Sun et al. (2018), and Simeckova et al. (2006), all cited by the authors, to be particularly helpful, and I think that there are places where the present manuscript presents conclusions or material that is similar (although clearly independently derived), so there are opportunities to make use of references to shorten the text. Again, my intent in making this comment is to find a way to limit the material in the paper to what is required to support the conclusion. a) Introduction, sections 2.0, 2.1: I think this was about the right length, although I'm not sure that the discussion of observations of Venus and Mars add much to supporting the goal of the paper. b) 2.2,2.3: This section is around 7.5 pages, and includes a lot of standard spectroscopy and line-by-line radiative transfer calculation information, including the use of Einstein A coefficients for non-LTE situations. The

discussion could be shortened considerably by the use of references and limiting the text to what is unique. Answer: As said in the beginning, we are quite reluctant to cut out or reduce some parts of the paper, because we wish to have it self-consistent. I was a little uncomfortable with the way that LBLRTM is being used in section 2.3.7, as it would be cleaner to just start from scratch with fresh code and do it right, but I appreciate that this may not have been practical given the resources available and it seems to have worked. Answer: we agree on this remark and we plan to do a calculation from scratch in the future. However the use of LBLRTM was very convenient; this software is widely used in the community, and maintained properly. If calculations from scratch on some examples will give the same results as LBLRTM, we will select the most convenient way to proceed. c) Section 3, the use of SCIAMACHY data: The authors have done a lot of work here, but I would suggest including only those elements of this section that are directly relevant to section 6. This section is 11.5 pages, and it is not clear to me how the onion peeling retrieval of VER from limb scans is relevant to simultaneous nadir retrievals of O2 column and airglow from MicroCarb. Answer: The onion peeling vertical inversion of SCIAMACHY limb observations (accounting for self-absorption by O2) was crucial to determine the true nadir intensity of the O2* airglow that will be observed in nadir MicroCarb geometry, and this was needed to test the algorithms allowing to disentangle the O2* emission spectrum from the general radiance coming from the ground and the lower atmosphere.

d) Section 4, comparison between REPROBUS airglow model and SCIAMACHY observations: The conclusion seems to be that the model underpredicts ozone and so underpredicts airglow, and so is not suitable for estimating airglow instead of retrieving it. Not sure if this is worth 12.5 pages; perhaps this work could be summarized? Answer: This discrepancy between the REPROBUS CTM model and SCIAMACHY O2* data is a very important scientific result. The comparison of GOMOS ozone data with REPROBUS ozone suggests that the airglow discrepancy is due at least (but may be not only) to a deficit in the ozone predicted by REPROBUS at high altitudes. As a result, co-author Franck Lefèvre will completely renew his REPROBUS code with new

chemistry solving algorithms, a "work in progress" expected to be achieved in 2020, and of course well beyond the scope of this paper.

e) Sections 5 and 6: It seems to me that this is the heart of the paper, and other sections should be adjusted so that they contain just what is needed to support the material in these two sections. Answer: as said above, we are reluctant to cut other parts. We wish to keep all the informations available to the reader, necessary for the reproduction of our results. f) Section 7: This seems to be a literature review, and not directly relevant, except perhaps the comments regarding CO2 airglow and potential impact on MicroCarb retrievals. Answer: this is not a literature review, but a non-exhaustive list of some situations when nadir viewing observations of one particular molecule are contaminated by fluorescence of the same molecule. We agree that the case of CH4 and CO are not directly relevant to MicroCarb, but we wish to take this opportunity to draw the attention of other scientists to this problem that seems to have been mostly ignored in the past. O2 band A and CO2 are relevant to MicroCarb. We suspect that contamination of the O2 band A might be larger than estimated by Sioris, because the emission was not estimated below 50 km. Also, it was not measured below 50 km in the analysis of Zarboo et al. of special SCIAMACHY MLT limb mode. g) Section 8, Conclusion: might need adjusting if the revisions above are considered.

Technical Corrections: Some of these corrections may be OBE if the major changes identified above are considered: 1)page 17, eq. 19: I think that the expression under the sq root in the third line should be r20 − p21, NOT r21 − p20 Answer: yes, you are absolutely correct! In earlier versions of the paper it was correct; a typo was introduced when we switched to word equation style for these equations. In fact r21 − p20 is negative. This error is not in the code. 2) Figure 19: It was very difficult to distinguish the stars and diamonds in the plots. Answer: this figure was redrawn with triangles instead of diamonds. 3) page 40, line 21: 4ARTIC retrieves CO2 and H2O on 19 vertical layers: what is the typical number of degrees of freedom for these retrievals? Is the profile information actually meaningful, or is it really just a column retrieval? Answer:

according to some other studies performed by ACRI, the typical number of of degrees of freedom for these retrievals is about 2. Therefore, you are right when you question the approach of 4ARCTIC with 19 levels. However, many other investigations are also using this 19 levels approach. 4) page 41, line 20: "...Henyey Greenstein function with g currently fixed to 0.8." A reference might be helpful here, for the function and for the choice of g. answer: this value of 0.8 is often selected in the literature to describe preferential forward scattering, but may be adjusted later. This kind of topics will be addressed in future papers, and is not addressed here in a detailed fashion: beyond the scope of the present paper. We have added the following sentence: "...fixed to 0.8, a value used frequently in the literature to describe preferential forward scattering, but could be adapted if necessary."

5) page 44, line 4: drop the extra "(" in Ag((_); done line 13: what reference spectrum is used?; Answer: this reference spectrum may be any O2* computed airglow spectrum. This normalization is done in order that the sum of the two coefficients of a linear combination of the two spectra (warm and cold) are of the order of unity, only for convenience. line 14: drop the "." after "spectrum.,"; done line 27: change "spectru" to " spectrum"; done line 28: So random error only, no calibration error, channel crosstalk, etc? Answer: yes, only random error is considered in this exercise with SCIAMACHY nadir viewing data. The random error is estimated from the fluctuations of the data. 6) page 55, line 14: delete "contaminated" (redundant after "contamination"); line 35: change "ETL" to "LTE"?; done 7) page 59, line 11: delete "inclusion of a" (redundant with previous "inclusion in the") done 8) page 66, line 15: I think that O2(b1S) should be O2(b1$\Sigma$)? done Interactive comment on Atmos. Meas. Tech. Discuss., doi:10.5194/amt-2019-54, 2019.

Please also note the supplement to this comment:
https://www.atmos-meas-tech-discuss.net/amt-2019-54/amt-2019-54-AC2-supplement.pdf

---

## Author Comment (AC3) · 21 Sep 2019

Response to referee #1 O2 paper amt-2019-54.docx
General answer: We are very grateful to referee #1 for his (her) careful reading of the manuscript, deep effort to reproduce some spectroscopic results, pointing out some errors, and one particularly interesting suggestion about the use of nadir viewing SCIA-MACHY data that we have followed. As a result, we have made considerable rewriting

of Section 2.3, and accepted most of the suggestions of refree #1.In the following, answers are in arial and blue in the following. Anonymous Referee #1 The feasibility of using the 1.27 $\mu$m, or the "singlet Delta" O2 band for satellite nadir remote sensing of greenhouse gases has been investigated previously (Sun et al. 2018). However, this manuscript, to my knowledge, firstly presents this observation strategy within the framework of a planned satellite mission (Micro-Carb). Since such a satellite mission may cost hundreds of millions of euros/dollars and provide crucial data with global coverage for many years, this strategy may potentially have a far-reaching impact, and this work may become an import reference for future missions. Given the aforementioned significance, this manuscript at this stage may still be substantially improved in terms of both presentation quality and technical rigor. The following lists my general comments concerning the manuscript structure and specific comments con- cerning individual sentences, equations, and figures. 1 General comments 1.1 In general the manuscript has been clearly improved since the initial submission. That said, it still reads a bit fragmental with different sections apparently from different groups of authors. The O2 molecules in a1_g state, for example, was denoted as O_2, O2(a1_g), O_2 (a1_), O2(1_), and O2(a1_) throughout the manuscript. As to the paper structure, Section 5 describes the MicroCarb mission and its physicsbased retrieval, 4ARTIC. Then it was deviated to a retrieval of SCIAMACHY nadir airglow using a different tool called LATMOS breadboard (sections 6.1–6.2) before 4ARTIC was applied to test different airglow mitigation approaches in section 6.3. The authors may consider moving sections 6.1–6.2 after section 3, because the SCIAMACHY nadir/limb airglow comparison naturally follows the onion-peeling VER retrieval, and combining section 5 with section 6.3. Some parts of section 7 seem to be out of the scope of this study (e.g., the discussion of methane and CO fluorescence), and the remaining parts may be combined with section 2.1.  1.2 Section 2.3 could be clarified and largely removed from the manuscript. First, the main conclusion from this section, equations 14 and 15 that will be used to convert absorption spectra simulated by LBLRTM to airglow emission spectra, seems to have already been demonstrated by equations 3–5 in Sun

et al. (2018). Combining equations 4 and 5 in Sun et al. (2018): answer: the recommendation to clarify and to largely remove it from the manuscript is self-contradictory, and is anyway not acceptable for the following reasons: 1. The theoretical development (to obtain a theoretical spectrum of the O2* airglow emission) that we present here was done in 2017, and completely independently from the work of Sun et al. (2018). If we were following the suggestion to just quote the equations of Sun et al. (2018) and not present our own analysis, it would give to the reader the false impression that we have followed blindly the developments of Sun et al. 2018, which is not true. The fact that both groups have developed the same kind of theory (form the same theoretical approach based on what can be found in Simeckova et al. (2006)) re-inforce the credibility of this approach, which is very important for "hundreds of millions dollars space projects". 2. Our equations (14) and (15) give an original result: a formula giving the wavenumber variation over the whole band of the ratio $\varepsilon$/SS of airglow emission $\varepsilon$ to the line strength SS found in Hitran table. This formula is NOT in the paper of Sun; the referee#1 had to check by some manipulation of equations (in black, below) that such a relation COULD be retrieved form formulas in Sun et al. (2018). Doing this, referee#1 shows that both groups are using consistent descriptions of the physics involved, a satisfactory piece of information for which we must thank him. 3. In fact, in the present study, we have used the formulation of this ratio in order to build very simply a synthetic spectrum of the airglow emission, by using the LBLRTM code computing the local absorption, and multiplying by the function $\varepsilon(\nu)$/SS$(\nu)$ (14) and (15). This is a totally original method, and we wish that all the AMT readers to be able to reproduce it and use it. This is why all equations establishing $\varepsilon(\nu)$/SS$(\nu)$ must be kept in the present paper. 4. The present manuscript was given to Iouli Gordon, one of the co-author of Sun et al (2018) and HITRAN expert and producer. It was also discussed by the first author(JLB) in a face-to-face meeting, and Iouli Gordon had no objection on the theoretical aspects. We consider that it re-inforces the validation of our approach.

The equation above is identical to equations 14–15 in the manuscript. As such, most equations/figures in section 2.3.2–2.3.6 can be replaced by a simple reference to equations 3–5 in Sun et al. (2018). answer: no. See the answer above.

Second, the symbols are not consistently defined, making section 2.3 challenging to follow. For example, the upper state energy is E2 in equation 9, E2i in equation 10, and Ei in line 9, page 11. answer: some rewriting has been performed to clarify what is what.

The total partition sums for both absorption transition and airglow emission are denoted as Qtot(T), but separately as Qlotot(T) and Qup tot(T) in equation15. answer: the text has been revised for more clarity, which wad needed indeed. The summation index is i in equation 10, but appears to be J0 in equation 11. Thelocal emission rate spectrum is given by Em(_) (line 17, page 15) and Emn(_) (line 18, page 15) in section 2.3.7, but it is unclear what this quantity really is. answer:The additional letter n is not an index, but stands for "normalized", since the values have been normalized. Finally, it will be very useful to discuss the improvements over previous studies. I did not find an equation that calculates Qup tot(T) in Sun et al. 2018, which is given by equation 10 in the manuscript. However, I cannot reproduce the results Qtot(T = 296K) = 147.196 (line 10, page 11) using equation 10 and HITRAN 2016 data because it is unclear what "index i refers to each value of E2" (line 36, page 10) exactly means. answer: index i refers to all values of the quantum number J' of each rotational level. This is now indicated in the text. One possible reason for a discrepancy on Qtot up may be that you may have taken the expression in (10), while we have (for convenience only) subtracted E_0 form all values of E2i in equation (10), to get a new definition of Qtot up, which probably coincides with the Q (T;airglow) from Sun et al. There is a factor 4.5x10 16 of difference. The other possibility is that you took the head band value of Lischichina et al.,(which is for J' =0) while we took for E2_0 the energy of the first authorized level J'=2. The relative distributions of levels are not affected by a particular choice of E2_0. 1.3 One significant missing piece of this study is the fine-scale spatial variation of airglow, potentially due to gravity waves. The REPROBUS CTM (2Å∼2_) and SCIAMACHY limb data (400 Å∼ 960 km) used in the study are too coarse to

capture those fine-scale variations. The fine-scale airglow features may propagate to the retrieved XCO2 as artefacts from the disturbed mesospheric temperature (spectral shape of both emission and re-absorption), excited O2 molecule concentration (emission), and O2 molecule concentration (re-absorption). The authors may have missed one opportunity as the SCIAMACHY nadir data over ocean have much higher spatial resolution and may reveal some gravity wave features. Larger scatter is seen from SCIA nadir ocean compared to SCIA limb (Figure 30). Is that just retrieval random errors or real spatial variation? How do they look on a map? Answer: We fully agree with your suggestion and made a new study. As a result, we have added at end of section 6.2 the following text: " Following an interesting suggestion of anonymous Referee#1, we have tried to estimate from nadir viewing SCIAMACHY data the small-scale horizontal variations of O2* airglow that could be due to gravity waves and are not represented in REPROBUS CTM. This is not an easy task using the relatively low spectral resolution of SCIAMACHY data. At this resolution spectral features in airglow and O2 absorption spectra are highly correlated and the estimation of airglow is accurate only for very low values of reflected solar flux as illustrated on Figure 29, where a large dispersion of airglow is observed for high values of reflected solar radiance. There are not enough observations reaching a low level of solar flux to plot maps of airglow. In spite of these limitations, we made an attempt to estimate at least an upper limit for the small-scale variations of airglow. We selected all pairs of nadir observations with reflected solar flux < 2 mW/m2/nm/sr, solar zenith angle < 60° and distance < 110 km. With these strong criteria only 1% of the observations were selected. The average difference in airglow intensity between the pairs of observations was equal to 1.0%. We consider this value as an upper limit of the impact of gravity wave perturbation in airglow intensity. At this level the impact on the retrieval of Psurf and XCO2 will be very limited."

We think that the absence of significant horizontal fluctuations of the O2* airglow is due to the fact that the emission layer is rather thick, in contrast with the much thinner OH layer at night, where horizontal fluctuations gravity waves may be observed.

[Figure]

2 Specific comments Line 31, page 1: the statement that the theoretical synthetic spectrum is from "a new approach" is contradictory to the following statement that this approach "is very similar (likely identical) to the approach of Sun et al. (2018)" (lines 7–8, page 2). The current work should be more clearly distinguished from previous before claiming it is "new". Answer: for us, this approach was new, with respect to a first crude approximation. We have deleted the word "new" at this place. Note that our work was done independently , and before the publication of Sun. Line 32, page 2: may include OCO-3 and GOSAT-2. Done Line 40, page 4: why is pressure broadening proportional to the "square" of air density? Answer: Sorry for this mistaken statement and thank you for pointing it out. The number of collisions per cm3 is proportional to the square of the density, but the number of collisions per O2 molecule (which is relevant to pressure broadening) is proportional to the density. Sentence corrected. Figure 2 caption: how can "transmittance" be "larger" than emission? Did the authors mean "wider"? Answer: Yes, of course. We have rephrased the sentence and used "wider": "Comparison at high spectral resolution of spectral shape of atmospheric O2 transmission (transmittance) and spectral shape of O2* emission. The FWHM of an individual O2 line (red) is much wider than the FWHM of its counterpart in emission (black line), allowing in principle to disentangle absorption from emission at selected wavelengths." Line 35, page 6: equation 3 looks the same as the second equation in the Appendix A1.1.1. They should be consistent. answer: They are consistent, because O atoms are in their fundamental state O(3P). This precision is now included in the text. Line 32, page 7: Dave Crisp et al. should be just Crisp et al. Done Line 30, page 9: quenching is significant below 50 km, and the airglow peak is also below 50 km. Are there any consequences if the quenching is ignored? Answer: The quenching becomes important at lower altitudes. If it were neglected in the airglow, it would result in a significant overestimate of the airglow. Line 15, page 14: E0 = 7892.02 cm−1 is inaccurate. Refer to Leshchishina et al. (2010). Answer: The work of Leschichina et al. is published in 2010. Co-authors are Gordon and Rothman, responsible for the HITRAN data base. We assume that the HITRAN2016 data base contains the best

available constants. Probably Leschichina et al.(2010) and us are not talking of the same thing. We find in the HITRAN data base that the minimum energy level E1 for the upper state is 7892.01738, for V'=0 and J'=2, as clearly stated in our paper. We have changed our value of 7892.02 into the full digit value 7892.01738. Anyway, it is a multiplicative factor, and is not important in the relative distribution of the various rotational levels of the upper state. Line 40, page 16: " appears to be spectrally integrated here, but spectrally resolved in equation 13, and back to be spectrally resolved in line 28, page 18 again. answer: you are right. We have been more specific at line 40, page 16; while for line 28, page 18, there is the sentence: "In reality, $\varepsilon$, $\tau$(s), Tr(s) and B all depend on wavelength $\lambda$.", which should be enough for the reader to understand. Line 5, page 18, equation 21, and line 4, page 19: a bold face " is used, but definition is unclear. answer: the classical notation (in english) for vectors and matrices is to write them in bold face, and this is what was intended here. We now have put B in bold face in the line just before equation (21). $\varepsilon$ is already in bold face in this line. We have also change to bold face B(z) in the line 4, page 19 (now changed to line 32, p.19). Line 24, page 21: why does O2 absorption have to be neglected? It is important and all parameters should be readily available to include O2 absorption in the analysis. answer: The O2 absorption certainly does not need to be "neglected" and we did not say that. It just happens that at the time of this exercise, we did not include the absorption in the model, and we are not going to redo it for this paper. We were glad enough to see the improvement of similarities between SCIAMACHY observed spectra and our new model of the emission (compared to our "old" model) when the emission is properly computed (figure 13). We have rephrased here: "We tentatively assign this behaviour to the fact that we have not accounted for..." and also in the paragraph before figure 13: " we have not accounted for the O2 absorption..." Figure 11: this figure may be removed. The SCIAMACHY limb data are binned across track so it is unnecessary to show separate across-track positions. The globe distribution of SCIAMACHY orbits is well represented by Figure 15. answer: we think it is important to see the separation between the 4 FOV at the limb and the distribution of the data points along one single

orbit. Figure 12b and the left panel in Figure 13: these should be removed, see the following argument. answer: we think that it is important to show that, at first glance (figures 12 a and 12 b), the observed spectra and the model spectra are quite similar, both for the crude approximation model and the "true" model. But when the ratios of observed/model are plotted, the better fit with the "true" model is obvious. So we wish to keep all these figures to show the importance of not using any longer the crude model.

Line 8, page 23: This is not a "simplified" model; the equation for absorption cross section is not simpler than equation for airglow emission. This is an "inaccurate" or "inappropriate" model. answer: we have replaced "simplified model" by "crude model", which is consistent with what is written in the first paragraph of Section 2.3.

Figure 16: SZArecomputed appears in the figure but is only explained at the end of page 31. answer: Figure 16 was redrawn without the word "recomputed". Lines 5–6, page 27: VER has been defined and should be only defined at first appearance. answer: Volume Emission Rate is now deleted here. Section 4.1.1, page 27: the authors are suggested to better articulate the usefulness of the REPROBUS model. Its low resolution is not sufficient to capture the fine-scale variation; its model top (0.01 hPa) does not fully capture the airglow profile (what is the error from neglecting airglows above 0.01 hPa?); the model is not used in the following MicroCarb retrieval tests. answer: We have added the following sentence: "It should be noted that, as a result of some discrepancies revealed by this comparison, the REPROBUS model will be modified in the future for a better representation of mesospheric ozone. Although the retrieval of O2 column does not need a model, it is likely that the output of the improved REPROBUS model (O2* intensity) will be used as a prior information in the retrieval process." Lines 13–14, page 29: how much is the error due to neglecting the reabsorption, and what is its relative contribution to total error? answer: In the nadir viewing geometry, the absorption of the O2* by O2 above 30 km would decrease the brightness by 1-2 % and was neglected. Figure 18: it's maybe useful to mark the

subsolar point in the plot. answer: we have added in the Figure 18a,b,c caption the sentence: "Because of the particular time and date, the subsolar point is at latitude 23.5° N and on the meridian 90° East, which is the one plotted at the center of each figure". Line 23–24, page 31: may change to "...an observer placed above the tangent points..." as there are multiple tangent points that are close to each other. Line 41, page 31: the SZA is defined at "one of the two points corresponding to the intersection between the line of sight and TOA". Which one? answer: we have rephrased the sentence: "We noted that the SZA value provided in the SCIAMACHY ESA products in limb viewing, as defined in the data product, is the SZA value of one of the two points (the nearest to ENVISAT) corresponding to the intersection between the LOS and TOA (Top of Atmosphere defined at 100 km altitude). But what we need is the SZA of the tangent point of the line of sight (LOS), which is different." Figure 19: removing the lines between markers may improve the figure clarity. answer: Figure 19 has been redrawn with the lines between points removed. Line 7, page 33: it could probably be delivered in a more rigorous way than asserting: "must have been". answer: yes, it could. But we are rather embarrassed, because a search on Google and ADS shows very little return on the topic of SCIAMACHY in-flight radiance calibration in the range around 1.27 $\mu$m by comparison with MODIS or other well calibrated instruments. We note also that another referee who obviously knows very well both SCIAMACHY and english language (likely John Burrows) did not protest about our formulation, and we prefer to keep our wording. Figures 20–22: Figure 22 should be enough to present all results so Figures 20–21 can be removed. answer: we have deleted Figure 21 and kept Figure 20 for three exemples of single profiles. Line 2, page 36: modified to "five orders of magnitude of variation of ozone with alti- tude". Done. Figure 25: what is the difference between the left and right? What are the observation time/SZA? answer: left and right are just two different examples of day side ozone profiles. The SZA are 38° (left) and 94°(right). During this second occultation, the tangent point is still solar illuminated. However, the chemical situation of ozone may be ambiguous at that SZA angle, and therefore we prefer to delete this figure and to keep only the occultation at

SZA=38°. The exact date and SZA is now indicated in the caption.

Line 5, page 39: the claim that "the night side model bias is quite negligible below 60 km" contradicts the main conclusion of this section, i.e., the ozone deficit below 60 km of 10–20% in the REPROBUS model is the main reason of airglow underestimation. Moreover, is it possible to compare with other reanalysis/observation data, especially MERRA-2? answer: We agree that there was a little bit of confusion in the summary of the comparisons GOMOS/REPROBUS, in large part because we wrote "lower" when we meant "larger" in the following sentence, now corrected: "- GOMOS ozone concentration vertical profiles show quite similar values below 60 km between day and night, and larger lower values of O3 at night above 60 km, a feature well understood from mesospheric chemistry. - there is a known shortcoming of the chemistry of REPROBUS model affecting strongly night side predictions above 60 km, quite apparent with GOMOS ozone night side comparisons (too much ozone in REPROBUS). - Because the model O3 diurnal variation is small below 60 km (there, we are more confident in the model than in GOMOS dayside data to estimate the small ozone diurnal variation), the comparison GOMOS/REPROBUS on the night side showing a deficit (10-20 %) of the model versus GOMOS ozone below 60 km may be applied also to the day side. " The comparison with MERRA-2 output model is an interesting suggestion but beyond the scope of the present study. Line 8, page 41: change "'maximum height Zaero" to something like "'peak AOD height Zaero" to avoid confusion. answer: agreed! we changed to " the height Zaero of peak aerosol concentration" Lines 8–9, page 42: the water vapour lines are not present in this figure. answer: agreed. We changed the caption text. Figure 27: what are the vertical axes on the right? In addition, it is hard to see, but there is a red line very close to the horizontal axes. What is that? answer: The figure has been corrected and redrawn. Congratulation for your accute viewing. Lines 12–19, page 42: those are repetition of lines 36–40, page 4, but not exactly the same. The same argument does not need to be repeated twice in the article. answer: in page 4, we illustrate the difference of width between absorption and emission, illustrated by figure 2; while in page 42, we show with the figure 27 the
CIA. In addition, we write:" ..as noted before..." Figure 28: The "@1.27$\mu$m" may be removed in the vetical axis label. answer: it could be removed but for practical reasons it will not be removed. Line 18, page 49: The REPROBUS model is coarse and cannot capture small-scale intensity variations, if they exist. answer: we have modified the sentence: "They should be very similar, if the characteristics of spatial lengths of intensity variations are as large as found by REPROBUS model, larger than the 2x2° REPROBUS resolution. This comparison would provide an important "sanity check" of the retrieval of Psurf (or O2 column). " Line 4, page 50: what is the resolution of the "very high resolution" spectrum? used by 4AOP? answer: the spectral sampling used in 4AOP is adjustable by the user. In the exercise that were done, the sampling was 0.001 cm-1. Done. Lines 23–26, page 50: what are the a priori errors, especially for Psurf and airglow scaling factors? answer: On most inversion exercises, the a priori errors were large: H2O : 2145.16 ppm; Psurf : 5 hPa ; mean Albedo : 1 Scaling factor of Airglow : 1000 (totally relaxed) except in case#2 where it was taken at 0.2 (20%). What is the impact of a priori on posterior error? answer: we have not done studies of this type and it could be interesting to do them in the future.

Does the airglow impact retrieved XCO2? answer: if the airglow is not accounted for (ignored in the retrieval and direct model), then it induces an error of 70 hPa on Psurf and a correlated 7% error on XCO2.

Line 29, page 50: what is "Lmoy"? answer: the definition of Lmoy is given in the following line. We have modified the etx for better clarity: "We remind that the MicroCarb requirements on the Psurf retrieval for a median intensity luminance Lmoy scenario are 0.1 hPa in term of bias and 1 hPa in term of random error. This reference luminance value Lmoy corresponds to an observation with SZA=36° and albedo at 1.27$\mu$m = 0.2." Lines 10–16, page 51: the cold/warm spectra here differ from what has been defined in equations 29–30 (spectra simulated at 217/270 K). Does this make any difference? answer: no, it does not make any significant difference. Line 31, page 56: change "dayglow" to airglow. Done. Line 39, page 56: may remove the exclamation mark after

"contaminated". Done.

Lines 13–15, page 57: the comparison with day-side GOMOS ozone profiles appears to be inconclusive according to section 4.3.2 and may not support the statement here.

Some pzrts of section 4.3.2 have been rephrased; nd here we have modified the sentence, in such a way that there is consistency between 4.3.2 and the sentence her: "For the time being, we assign this deficit to be due at least partially (but possibly not totally) to an ozone deficit in the REPROBUS model,..."

Lines 31–32, page 57: the first sentence may be inaccurate because a few factors stated previously in the manuscript. The absorption is still saturated while airglow emission grows linearly, the absorption is much more pressure-broadened, and only CIA is present in absorption. All those factors still hold even the airglow spectral shape were identical to absorption. answer: We still think that our sentence is correct: "...if the dayglow spectrum of O2* were strictly identical to the O2 absorption spectrum,": strictly identical is implying CIA, same pressure broadening at the same pressure, etc... Line 5, page 58: the O2 a1_g band is significantly more complicated than just P, Q, and R branches. The same argument applies to discussion of transitions in sections 2.3.3–2.3.5, which is suggested to be removed. answer: The P,Q,R branches contains the most intense lines of the transition. One advantage to use the LBLRTM code based on the HITRAN absorption data base, and then multiply by a smooth function of wave number to get the emission, is that the HITRAN data base contains all the lines (within a certain wavelength interval) above a very small line intensity threshold, including for instance some of the lines V'=1, V"=1, and some lines of branches N,O, S,T. Lines 31–32, page 58: I've found CIA being considered in Figure 3 and Eq. S5 in Sun et al. (2018), so it is inaccurate to state that "their whole analysis was done without accounting for the CIA O2 absorption". answer: You are right. Therefore we have modified the text there: "Their whole analysis was done with already accounting for the CIA O2 absorption, whose broad size and smooth pattern is insensitive to spectral resolving power (Fig. 27). On the other hand, as can be seen in Fig. 27 with the same number of spectels as MicroCarb and a coarser spectral resolution (and sampling), the whole O2 band would be measured and would possibly allow to better constrain the CIA absorption and O2 column retrieval.. The larger spectral sampling gives additional photons per spectel, which may be traded-off for an increased spatial resolution. However, the high resolving power of MicroCarb is an asset for the exact knowledge of the instrumental spectral function which is important for the retrieval accuracy."

and further down, we modified also slightly: "We suggest though, on the basis of our analysis and the results of Sun et al. (2018), that when CIA is taken into account, a spectral resolving power of about 5,000 and a high SNR could possibly yield a sufficiently good accuracy..."

Please also note the supplement to this comment:
https://www.atmos-meas-tech-discuss.net/amt-2019-54/amt-2019-54-AC3-supplement.pdf

---

## Author Response (AR1)

Revised version, September 22, 2019.

Dear Editor,

Fisrt, we must convey our gratitude to the work performed by the Editor and referees, who all have scrutinized in detail our paper, in spite of its length. They pointed out, not only English language errors, but also a number of small errors that are now corrected. A fair number of short sentences have been added for clarification, at the request of referees. As a result, our revised paper is really improved w.r.t. the first submitted version.

Several figures have been redrawn at the request of referees, and one has been deleted. However, at the request of Referee #2, we had to produce new figures to explain the details of our data processing of SCIAMACHY Level 1c data. These new figures are displayed in one Appendix B. We think that these figures are not necessary for our paper; we are ready to follow the advice of the Editor if he thinks that this Appendix B may be removed form the final version. This appendix is also included in our response to RC3 from Referee #2.

We have not accepted the suggestions of Referee #1 to mostly remove Section 2.3, which is the heart of paper, for reasons detailed in our answer to this Referee#1. We copy one of them below:

1. The theoretical development (to obtain a theoretical spectrum of the  $O_2^*$  airglow emission) that we present here was done in 2017, and completely independently from the work of Sun et al. (2018). If we were following the suggestion to just quote the equations of Sun et al. (2018) and not present our own analysis, it would give to the reader the false impression that we have followed blindly the developments of Sun et al. 2018, which is not true. The fact that both groups have developed the same kind of theory (form the same theoretical approach based on what can be found in Simeckova et al. (2006)) re-inforce the credibility of this approach, which is very important for "hundreds of millions dollars space projects".

We think that following the suggestion of Referee#1 would give unduly a unique credit to a US team in this field, while we feel that AMT is basically a European inspired publication.

**Length of the paper:** We recognize that this paper is long, but we still believe that its overall length is appropriate. At an early stage one reviewer suggested to split the paper in several papers but we have been quite reluctant to continue along this line (split or shorten substantially) for the following reasons.

All parts of the paper are relevant to the same subject: is it possible to use the  $O_2$  1.27 µm absorption band for  $CO_2$  mixing ratio retrieval, in spite of the strong airglow contamination?

The team that was assembled for this scientific research had to cover several scientific aspects: our understanding of this airglow, building a model for the intensity, and a model for the spectral shape, validation with comparisons with

SCIAMACHY/ENVISAT data, separation of airglow from absorption. One reader is not obliged to read carefully all sections, he can pick up what he is most interested in. We estimate that if we would split our paper into two papers, the overall total length of the two papers would be longer than the present version, because of unavoidable repetitions (each paper must be self-consistent, including references). It would require also twice more reviewers and Editor work.

AMT stands for Atmospheric Measurements Techniques and therefore our paper is perfectly in scope with the profile of the publication.

Our paper is long because it is deliberately rather detailed, because we wish to ease the possibility that anybody else to be able to reproduce our results. The spirit of AMT, with public discussions before final publications, is in line with the "open source" philosophy. Cancelling parts of the paper would jeopardize this philosophy.

Remember that the results of about 30% of all scientific papers cannot be reproduced by other scientists, and this comes to 50% of papers in biology, a very embarrassing situation.

One great advantage of AMT publication is that it does not require paper printing, therefore cancelling a source of  $CO_2$  production. Only an interested reader would potentially print it. Therefore, with AMT we may reconcile  $CO_2$  economy and detailed description for better reproducibility of results.

In its present form, our paper is somewhat "self-consistent" on its subject. It will serve as a reference, not only for the MicroCarb project, but also on other future GHG monitoring space projects that may consider the use of the  $1.27 \,\mu m$  band.

Finally, we note that the length of the paper did not discourage a fairly large number of scientists to download the paper when discussed in AMTD: The paper has been viewed HTML 262 times and the pdf downloaded 100 times (21 September 2019), about half from the US. If the final version were cut significantly, it would introduce an advantage to those who uploaded the early version versus those seeing only the final version. It would also jeopardize the efforts of the referees that have scrutinized and corrected the whole text.

The MicroCarb project is in full development now, and we hope for many new results to come before and after launch. We would be glad to continue to publish in AMT and ACP, if the present experience with AMT comes to a satisfactory conclusion.

On behalf of the authors

Jean-Loup Bertaux

**2.3.3 Computing the distribution of O2\* molecules among the various energy levels**

In their 2006 paper, Simeckova et al. (2006) describe « the calculation of the statistical weights and the Einstein A -coefficients for the 39 molecules and their associated isotopologues/isotopomers currently present in the lineby-line portion of the HITRAN database ». This is all that is needed to calculate second members of equation (6) for all allowed transitions  $L_i$ , giving the rate of emission of the corresponding spectral line VER( $L_i$ ).

In an approximation of a two level system (upper m and lower n levels are denoted as 2 and 1 respectively) at LTE (Local Thermodynamic Equilibrium), we have the well-known equations linking the Einstein A-coefficients and B-coefficients

$$g_1 B_{12} = g_2 B_{21}$$
(7)
$$A_{21} = 8\pi h \nu^3 B_{21}$$
(8)

where  $A_{21}$ (spontaneous emission) is in s-1, and  $B_{12}$  (absorption) and  $B_{21}$  (stimulated emission) are in cm3 (J s2)-1, and  $g_1$  and  $g_2$  are the statistical weights of the levels 1 and 2, respectively.

We start from equation (17) of Simeckova et al. (2006) with molecules in the lower level 1 and the upper level 2 (much less numerous at atmospheric temperatures), to describe their relative distribution according to their energy level  $E_{1i}$  or  $E_{2i}$  and temperature T, the index i indicating a particular rovibrational level defined by J' and V'. If N is the total number of molecules per unit volume at the temperature T, the population  $N_{2i}$  of one of the energy level  $E_{2i}$  of the upper level 2 is equal to:

$$N_{2i} = \frac{g_{2i}N}{q_{tot}(T)} e^{-c_2 E_{2i}/T}$$
(9)

and a similar equation for  $N_{di}$  and the energies  $E_{di}$  of the lower level (equation (17) of Simeckova et al. 2006). Here,  $Q_{tot}(T)$  is the total internal partition sum of the absorbing gas at the temperature T,  $g_{2i}=2J'+1$ , and  $E_{2i}$  is the energy of the upper state in units of wavenumber (cm-1).  $c_2$  is the second radiation constant,  $c_2$ -*hc*/kB, where c is the speed of light, *h* is the Planck constant, and  $k_B = 1.38065 \times 10^{-23}$  joule K-1 is the Boltzmann constant. The total number of molecules per unit volume N=  $\Sigma N_{11} + \Sigma N_{2i}$ , and  $Q_{tot}(T)$  is the sum of  $Q_{tok}^{(0)}(T)$  and  $Q_{tok}^{(0)}(T)$ , respectively the internal partition sum of the lower level and the upper level. The index *i* refers to all possible values of J', starting at J'=2 (J'=0 and J'=1 do not exist). We have by definition:

$$Q_{tot}^{up}(T) = \sum_{i} g_{2i} \ e^{-c_2 E_{2i}/T}$$
(10)

We may find the value of  $Q_{tot}(T)$  in Table 1 of the paper of Simeckova et al. (2006). For instance,  $Q_{tot}(T=296 \text{ K})=215.77$  for the main oxygen isotopologue  ${}^{16}O{}^{16}O$ . The temperature 296 K is a reference temperature for the HITRAN database. For our purpose, we have to find  $Q_{tob}{}^{up}(T)$  for the upper level of the transition, from a summation described in equation (10). The summation must be not over all the transitions, but over all energy levels, because from a given energy level having a certain population  $N_{2i}$ , there are several transitions going down to the lower level with different  $A_{21}$ . Once we have  $Q_{tot}(T)$ , the total internal partition sum, then we may

| bertaux 30 sept. 19 15:32
bertaux 29 sept. 19 15:35
bertaux 30 sept. 19 15:31
bertaux 30 sept. 19 15:30
bertaux 29 sept. 19 15:35
bertaux 29 sept. 19 15:41
bertaux 29 sept. 19 15:41
bertaux 29 sept. 19 15:42
particular upper state rotational level with
quantum number J' and N 21 the number of
molecules populating this level.
bertaux 29 sept. 19 15:42
bertaux 29 sept. 19 15:43
bertaux 29 sept. 19 15:43
|--------------------------------------------------------------------------------------------------------------------------------------------------------------------------------------------------------------------------------------------------------------------------------------------------------------------------------------------------------------------------------------------------------------------------------------------------------------------------------------------------------------------------------------------------------------------------------------------------------------------------------------------------------------------------------------------------------------------------------------------------------------------------------|
bertaux 29 sept. 19 15:35
bertaux 30 sept. 19 15:31
bertaux 30 sept. 19 15:30
bertaux 29 sept. 19 15:35
bertaux 29 sept. 19 15:41
bertaux 29 sept. 19 15:41
bertaux 29 sept. 19 15:42
particular upper state rotational level with
quantum number J' and N 21 the number of
molecules populating this level.
bertaux 29 sept. 19 15:42
bertaux 29 sept. 19 15:43
bertaux 29 sept. 19 15:43
| bertaux 29 sept. 19 15:35
bertaux 30 sept. 19 15:31
bertaux 30 sept. 19 15:30
bertaux 29 sept. 19 15:35
bertaux 29 sept. 19 15:38
bertaux 29 sept. 19 15:41
bertaux 29 sept. 19 15:41
bertaux 29 sept. 19 15:42
particular upper state rotational level with
quantum number J 2 and N 21 ; the number of
molecules populating this level.
bertaux 29 sept. 19 15:42
bertaux 29 sept. 19 15:38
bertaux 29 sept. 19 15:38
bertaux 29 sept. 19 15:43
particular upper state rotational level with
quantum number J' and $N_{2i}$ the number of
molecules populating this level.bertaux 29 sept. 19 15:42Supprimé: Ibertaux 29 sept. 19 15:38Supprimé: ibertaux 29 sept. 19 15:38Supprimé: ibertaux 29 sept. 19 15:38Supprimé: ibertaux 29 sept. 19 15:43Supprimé: i                                                                                                    |
| bertaux 30 sept. 19 15:31
bertaux 30 sept. 19 15:30
bertaux 29 sept. 19 15:35
bertaux 29 sept. 19 15:38
bertaux 29 sept. 19 15:41
bertaux 29 sept. 19 15:41
bertaux 29 sept. 19 15:42
particular upper state rotational level with
quantum number J' and $N_{2i}$ the number of
molecules populating this level.
bertaux 29 sept. 19 15:42
bertaux 29 sept. 19 15:38
bertaux 29 sept. 19 15:43
particular upper state rotational level with
quantum number J' and $N_{2i}$ the number of
molecules populating this level.bertaux 29 sept. 19 15:42Supprimé: Ibertaux 29 sept. 19 15:38Supprimé: ibertaux 29 sept. 19 15:43Supprimé: i                                                                                                                                                                                                                         |
| bertaux 30 sept. 19 15:30
bertaux 29 sept. 19 15:35
bertaux 29 sept. 19 15:38
bertaux 29 sept. 19 15:41
bertaux 29 sept. 19 15:41
bertaux 29 sept. 19 15:42
particular upper state rotational level with
quantum number J' and N 2i the number of
molecules populating this level.
bertaux 29 sept. 19 15:42
bertaux 29 sept. 19 15:38
bertaux 29 sept. 19 15:43
bertaux 29 sept. 19 15:35
bertaux 29 sept. 19 15:38
bertaux 29 sept. 19 15:41
bertaux 29 sept. 19 15:41
bertaux 29 sept. 19 15:42
particular upper state rotational level with
quantum number J' and N 21 ; the number of
molecules populating this level.
bertaux 29 sept. 19 15:42
bertaux 29 sept. 19 15:38
bertaux 29 sept. 19 15:43
| bertaux 29 sept. 19 15:35
bertaux 29 sept. 19 15:38
bertaux 29 sept. 19 15:41
bertaux 29 sept. 19 15:41
bertaux 29 sept. 19 15:42
particular upper state rotational level with
quantum number J 2 and N 2i the number of
molecules populating this level.
bertaux 29 sept. 19 15:42
bertaux 29 sept. 19 15:38
bertaux 29 sept. 19 15:43
bertaux 29 sept. 19 15:38
bertaux 29 sept. 19 15:41
bertaux 29 sept. 19 15:41
bertaux 29 sept. 19 15:42
particular upper state rotational level with
quantum number J' and N 2 ; the number of
molecules populating this level.
bertaux 29 sept. 19 15:42
bertaux 29 sept. 19 15:38
bertaux 29 sept. 19 15:43
| bertaux 29 sept. 19 15:38
bertaux 29 sept. 19 15:41
bertaux 29 sept. 19 15:41
bertaux 29 sept. 19 15:42
particular upper state rotational level with
quantum number J' and N 2i the number of
molecules populating this level.
bertaux 29 sept. 19 15:42
bertaux 29 sept. 19 15:38
bertaux 29 sept. 19 15:43
| bertaux 29 sept. 19 15:41
bertaux 29 sept. 19 15:41
bertaux 29 sept. 19 15:42
particular upper state rotational level with
quantum number J 2 and N 2i the number of
molecules populating this level.
bertaux 29 sept. 19 15:42
bertaux 29 sept. 19 15:38
bertaux 29 sept. 19 15:43
bertaux 29 sept. 19 15:41
bertaux 29 sept. 19 15:42
particular upper state rotational level with
quantum number J' and N 2 ; the number of
molecules populating this level.
bertaux 29 sept. 19 15:42
bertaux 29 sept. 19 15:38
bertaux 29 sept. 19 15:43
| bertaux 29 sept. 19 15:41
bertaux 29 sept. 19 15:42
particular upper state rotational level with
quantum number J' and N 2i the number of
molecules populating this level.
bertaux 29 sept. 19 15:42
bertaux 29 sept. 19 15:38
bertaux 29 sept. 19 15:43
bertaux 29 sept. 19 15:42
particular upper state rotational level with
quantum number J' and $N_{2i}$ the number of
molecules populating this level.
bertaux 29 sept. 19 15:42
bertaux 29 sept. 19 15:38
bertaux 29 sept. 19 15:43
| bertaux 29 sept. 19 15:42
particular upper state rotational level with
quantum number J' and $N_{2i}$ the number of
molecules populating this level.
bertaux 29 sept. 19 15:42
bertaux 29 sept. 19 15:38
bertaux 29 sept. 19 15:43
particular upper state rotational level with
quantum number J' and $N_{2i}$ the number of
molecules populating this level.bertaux 29 sept. 19 15:42Supprimé: I
bertaux 29 sept. 19 15:38Supprimé: i
bertaux 29 sept. 19 15:43Supprimé:                                                                                                                                                                                                                                                                                                                                                                                                                                                                               |
| bertaux 29 sept. 19 15:42
bertaux 29 sept. 19 15:38
bertaux 29 sept. 19 15:43
bertaux 29 sept. 19 15:38
bertaux 29 sept. 19 15:43
| bertaux 29 sept. 19 15:38
bertaux 29 sept. 19 15:43
bertaux 29 sept. 19 15:43
| bertaux 29 sept. 19 15:43
|                                                                                                                                                                                                                                                                                                                                                                                                                                                                                                                                                                                                                                                                                                                                                                                |
| bertaux 29 sept. 19 15:45                                                                                                                                                                                                                                                                                                                                                                                                                                                                                                                                                                                                                                                                                                                                                      |
|                                                                                                                                                                                                                                                                                                                                                                                                                                                                                                                                                                                                                                                                                                                                                                                |
| bertaux 29 sept. 19 15:46                                                                                                                                                                                                                                                                                                                                                                                                                                                                                                                                                                                                                                                                                                                                                      |
| bertaux 29 sept. 19 15:46
| bertaux 29 sept. 19 15:46 Supprimé: like bertaux 29 sept. 19 15:46                                                                                                                                                                                                                                                                                                                                                                                                                                                                                                                                                                                                                                                                                                             |

compute all values of  $N_{2i}$ , for the required temperature, from the distribution of the excited molecules between the various energy levels from equation (9).

However,  $Q_{tok} = (T)$  is very small when referred to all molecules N. For convenience, we have replaced in equation (10) the values  $E_{2i}$  by  $E_{2i}$ -  $E_{20}$ , where  $E_{20}$  is the energy of the lowest energy populated level with J'=2. With this approach we found new values  $Q'_{101} = Q_{102} \exp(c_2 E_{20}/T)$  for the upper level;  $Q'_{10k} = (T=296 \text{ K})$  for 296 K, and  $Q_{tok} = (T=200 \text{ K})$  is 100.143. In Fig. 4 are represented both the exponential term and the statistical weight, product of the exponential term and 2J'+1. Only the V'=0 are kept here, because levels V'=1 are weakly populated, though they are present in the line list that are extracted from HITRAN line-by-line data base in our selected wavelength interval of interest (transition (1,1).

[revised manuscript text omitted]

---

## Author Response (AR2)

Letter to the Editor about paper amt-2019-54.docx
"The use of O2 1.27μm absorption band revisited for GHG monitoring from space and application to MicroCarb"
 by Jean-Loup Bertaux et al.

Revised version, November 8, 2019.

Dear Editor,

In response to your request to implement "Major Revisions" in our manuscript, we are re-submitting a new version Rev6 of our manuscript taking into account your remarks listed in your letter dated October 22.
We have cut the main text from 60 pages (before References) to 37 pages only, a cut of 38 %. The deleted material contains essential information for the reader that would wish to reproduce our results, but possibly not essential for all readers. We have therefore re-organized the text with material now put in several Appendices, which should streamline the narrative.

We have also made some modifications relevant to some detailed remarks that you did, in particular in the Spectroscopy Section and now Appendix B.

1.As you suggested we have added in the main text the following sentence:
"We should mention that one reviewer was able to show with some manipulations of equations that the same relationship (4) could be obtained from the equations contained in Sun et al. (2018). It clearly stands as a validation of our present work and shows that the two approaches are consistent."

2.You criticize the fact that we jump from energy levels to transitions back and forth. We admit that it seems not so logical, but it has a deep-rooted reason. Since we are not professional spectroscopists as you seem to be, we have conducted our calculations from the informations contained in the HITRAN database, which consists of all transitions of the $O_2$ molecules between two wavelength limits, one line per transition containing some informations about the upper energy level (necessary to compute $\exp(-E/kT)$ ) and the allowed values of J', as described in Simeckova et al. (2006)
Therefore, we had to be careful to extract the relevant informations allowing to compute $Q_{tot}(T)$ for example, for any temperature. It is not trivial (not very complicated either) to extract the informations for all energy levels from the HITRAN list of transitions, and we wanted to caution the reader.
We have deleted the sentence:
"because from a given energy level having a certain population $N_{2i}$, there are several transitions going down to the lower level with different $A_{21}$."
and replace the whole sentence by:
"Since the HITRAN database consists in a list of transitions, some caution must be used when using the HITRAN database, in order to extract a list of energy levels."
and in the caption of Figure now B2, we have deleted the sentence:

"There are 5, 7, or 8 values (and transitions) for each black circle in the figure, present in the HITRAN list, because of transitions selection rules and weak lines (below a certain threshold) are not in HITRAN. The V''=1, V'=1 transitions (1,1) are

not shown."
and replaced by:
Black circles: the population computed from informations contained in the HITRAN database of transitions. It allows to retrieve the allowed values of J'. The first point is for J'=2 with value 5=2J'+1, and exp(0)=1.

Hoping to have answered satisfactorily to your remarks

On behalf of the authors

Jean-Loup Bertaux

[revised manuscript text omitted]

This equation is the same as equation (B13) of the Appendix B, where the various constants are described, and T is the temperature of the atmosphere in which is produced the airglow.

We have used this formulation to transform an absorption spectrum by $O_2$ that can be easily computed with LBLRTM software (see details in the Appendix B) into a synthetic absorption emission spectrum. This method of construction of a synthetic emission spectrum was the basis of our work on three topics: a satisfactory comparison with the observed spectra of SCIAMACHY (see below); retrieving the airglow intensity from SCIAMACHY nadir data over low albedo regions; retrieving the surface pressure from simulations at high spectral resolution.

We should mention that one reviewer was able to show with some manipulations of equations that the same relationship (4) could be obtained from the equations contained in Sun et al. (2018). It clearly stands as a validation of our present work and shows that the two approaches are consistent.

**3. The use of SCIAMACHY data for the study of the $O_2$ ($^1\Delta$) emission**

Several space instruments have been used in the past for the study of the $O_2$ ($^1\Delta$) emission, mainly to retrieve the $O_3$ concentration: the Solar Mesosphere Explorer satellite (SME, Thomas et al., 1984); one infra-red radiometer aboard the satellite OHZORA, (Yamamoto et al., 1988); one infra-red imager a part of OSIRIS instrument on board ODIN (Llewellyn et al., 2004); the SABER broad band photometer on board TIMED NASA mission (Russell et al., 1999; Gao et al., 2011) and SCIAMACHY spectrometer on board ESA ENVISAT mission (Burrows et al., 1995) . We have used the SCIAMACHY data because of the spectral capability (resolution $\lambda/d\lambda{\sim}850$) and extensive produced data set during the ESA/ENVISAT mission.

**3.1 Description of SCIAMACHY investigation of $O_2$ ($^1\Delta$) emission**

SCIAMACHY is a multi-channel spectrometer dedicated to the study of Earth's atmosphere on board the European Space Agency Envisat satellite. The name is the acronym of SCanning Imaging Absorption SpectroMeter for Atmospheric CHartographY (Burrows et al., 1995, Bovensmann et al., 1999). It is an eight-channel grating spectrometer that measures scattered sunlight in limb and nadir geometries from 240 to 2,380 nm. In addition it was operated also in solar and lunar occultation. In this study, we have used both limb and nadir measurements covering the $O_2$ ($^1\Delta$) band (1,230–1,320 nm) in the spectral channel 6 (1,050–1,700 nm).

In a recent study to retrieve the volume emission rates of $O_2$ ($^1\Delta$) and $O_2$ ($^1\Sigma$) in the mesosphere and lower thermosphere, Zarboo et al. (2018) have used a special mode of SCIAMACHY: the MLT limb scan mode, dedicated to the study of the mesosphere and lower thermosphere in the region 50-150 km. This mode was used only twice a month from July 2008 until April 2012. In contrast, we have used the normal limb mode viewing geometry, where SCIAMACHY tangentially observes the atmosphere from the surface up to about 100 km with a vertical step of 3.3 km. At each tangent point, the FWHM of the FOV is 2.6 km (with a somewhat coarser vertical resolution), the horizontal along-track resolution is about 400 km, and the horizontal cross-track

resolution is 240 km. To improve the signal-to-noise ratio, the four cross-track spectra at the same elevation step are co-added, reducing cross-track resolution to 960 km (the swath width).

To generate data for our study, we used the SCIAMACHY dataset level 1b version 8.02 that we converted into level 1c radiometrically calibrated radiances (in physical unit) by using the SCIAMACHY command line tool

5    SciaL1c version 3.2. Before deriving the $O_2(^1\Delta)$ VER profiles, we had to perform a few corrections on the level 1c radiance spectra, as illustrated in Appendix C. First we subtracted the average of the 4 spectra measured above 105 km tangent height (generally around 150 km or 250 km) as a dark spectrum from the measured spectra at all of the other tangent heights. This high altitude spectrum contains some residual spectral (readout) patterns left from the calibration step. All spectra contain two bad pixels at wavelength 1262.267 nm and

10    1282.128 nm. In order to correct these two pixels we replaced their value by the average of their two surrounding pixels. When the tangent altitude of the LOS decreases, there is an increasing background signal due to the Rayleigh and/or aerosol scattering outside the $O_2$ band. We corrected the spectra from this signal by removing a straight line computed as a linear interpolation between the two "surrounding" average backgrounds (estimated from the median value of all points to avoid outliers) in the [1235-1245] nm domain and in the [1295-

15    1305] nm domain. The spectra after correction are ready to be used for the retrieval of the SCIAMACHY $O_2(^1\Delta)$ volume emission rate (VER), as described in Appendix C. An onion-peeling method, modified to account ofr the re-absoprtion of $O_2$, allows to retreive the VER vertical distribution from any limb scan. Then the VER is integrated vertically, yielding the $O_2(^1\Delta)$ intensity that would be observed at nadir for an observer located at the tangent point of the limb scan.

20    In Fig. 3 the nadir radiances (equivalent to intensities or brightnesses) derived from a series of SCIAMACHY limb scans along one particular orbit are plotted as a function of Solar Zenith Angle (SZA), when different atmospheric models are used. For each model, there are two branches, corresponding to North and South along the dayside polar orbit of ENVISAT (the North branch is in winter, while the South branch is in summer for this orbit). We see that the choice of the atmospheric model in the computation of the $O_2$ absorption has a small

25    (~3%) but noticeable impact (on the brightness seen at nadir). We have also plotted the prediction of the REPROBUS model, as described in Section 4 and subsection 4.2.1. It should be noted that the choice of the "adapted climatology" makes it possible to reduce the separation between the two branches and thus to be closer to the separation between the two branches obtained with the REPROBUS model.

[Figure]

Figure 3. Computed O₂* radiances in nadir viewing geometry, derived from SCIAMACHY limb radiances, as a function of SZA for orbit 20070101_1256 when the O₂ absorption is computed with various choices of atmospheric models: Climatology US_STANDARD (black), SUBARTIC_WINTER (blue), SUBARCTIC_SUMMER (green) and ADAPTED_CLIMATO (red) (see Appendix C for details). There is a slight dependence of the nadir intensity on the choice of atmospheric model. The purple dashed curve (with filled circles) corresponds to the REPROBUS v02 model.

**3.3 Computation of synthetic spectra and comparison with SCHIAMACHY observed spectra**

Once we have the vertical profile of VER corresponding to a given SCIAMACHY limb scan, we can compute the spectrum of the local emissivity (in absolute units of photons/ (cm³ s¹ sr nm)) with the theoretical approach developed in Appendix B. Then we may integrate the spectra with Abel's integral along horizontal LOS tangent at the limb, for a direct comparison with the actually observed SCIAMACHY spectra. In this particular exercise, we did not account for the O₂ absorption for simplicity, and for this reason we restricted our comparison to altitudes >60 km. The spectral resolution of SCIAMACHY was used to smooth the high resolution spectra (line by line) obtained from the approach described in Appendix B.

In Fig. 4 are represented the locations of the tangent points of SCIAMACHY limb scans for a particular orbit of ENVISAT. In Fig.5a are represented the observed spectra, binned by altitudes (60-70 km, 70-80 km, and 80-90 km), along with our model spectra computed for the same scans and binned in the same way, for a particular limb scan (points in green in Fig. 4). The agreement is basically very good, both in shape and intensity. We note that the model is slightly brighter than the data, and the relative difference is larger for the bin 60-70 km than for the other bins. We tentatively assign this behaviour to the fact that we have not accounted for the O₂ absorption along the LOS in the model, more important at 60-70 km than higher.

Figure 5b is the same as Fig.5a, with the crude model in which the spectral shape of the O₂* emission is identical

bertaux 31 oct. 19 16:35

bertaux 7 nov. 19 17:19

bertaux 7 nov. 19 17:20

bertaux 31 oct. 19 17:12

bertaux 31 oct. 19 17:12

bertaux 31 oct. 19 17:13

bertaux 31 oct. 19 17:13

bertaux 31 oct. 19 17:13

to the $O_2$ absorption. In this case, the R branch is systematically overestimated by this crude model.

[Figure]

**Figure 4. Geographic positions of SCIAMACHY LOS tangent points at the limb (by groups of four) for ENVISAT orbit 25293, starting 1th January 2007 at 01h 12mn. The green points are the locations of the limb spectra which are compared with our theoretical derivation.**

bertaux 31 oct. 19 16:35

[Figure]

**Figure 5a. SCIAMACHY limb spectra (solid lines, absolute units are photons/(s cm$^2$ nm sr)), binned by altitudes (60-70 km, 70-80 km, and 80-90 km), along with our model spectra computed for the same scans and binned in the same way.**

bertaux 31 oct. 19 16:36

[Figure]

**Figure 5b. Same as Fig. 5a, but with the crude model in which the shape of the emission of O₂\* is identical to the absorption by O₂. This crude model shows an excess of emission in the R branch (left) and a deficit in the P branch.**

The ratio of measured spectra /model spectra, Sobs/ Smod were averaged together for all scans of that particular orbit within the same three altitude bins. They are represented on Fig. 6, both for the crude model (absorption = emission, left), and for our "true" model of emission (right). It is clear that the crude model does not represent well the observed spectra, while the model with the true emission agrees quite well with the data. This comparison validates completely the approach that we developed in Section 2 and Appendix B, except that the overall level of the ratio is slightly below 1 (right panel). Again we assign this behaviour to the fact that we have not accounted for the O₂ absorption along the LOS in the model, and it can be seen that the ratios are nearer 1 for larger altitudes. Below 1255 nm and above 1285 nm, the intensity of the spectra is very small and thus we attribute the noisy shape of the ratio spectra to low SNR.

[Figure]

**Figure 6: Ratios of measured spectra/model spectra of limb spectra, averaged over a whole ENVISAT orbit, and binned by altitudes (60-70 km, 70-80 km, and 80-90 km). Left: crude model in which the shape of the emission of O₂\* is identical to the absorption by O₂. Right: same ratios with our new model described in Appendix B. The ratios are**

bertaux 31 oct. 19 16:36

bertaux 31 oct. 19 16:36

bertaux 31 oct. 19 17:13

bertaux 31 oct. 19 16:36

bertaux 7 nov. 19 17:25

bertaux 31 oct. 19 17:14

[revised manuscript text omitted]

---

## Author Response (AR3)

**Associate Editor Decision: Reconsider after major revisions** (07 Feb 2020) by
Christof Janssen
Comments to the Author:
Dear Authors,

While most of the referees have given a very positive evaluation and the scientific quality of the manuscript is out of question, one referee has still expressed serious concerns about the presentation quality of your manuscript. Given that I also have stumbled over many details (see my individual remarks further below), I request that the paper gets revised once more before being published. The paper also concludes on results that are only presented in the Appendix. Please include the parts that are essential for your conclusions and stay as concise as possible. I agree with referee 3 that Appendix A can be suppressed altogether.

The manuscript is still hard to read at times and it is therefore recommended that you improve on the readability of the paper.

When revising, please take into account the remarks of all referees.
**Answers are written in blue, bold face.**

p 2 l 16 : The atmospheric fraction is the ratio of the atmospheric increase of CO2 mass to the mass of CO2 anthropogenic emission **a new sentence of explanation is added just before this one:** We know how much $CO_2$ is produced each year by human activity, but it does not correspond to the measured yearly increase of $CO_2$ in the atmosphere.

p 2 l 33 : CEOS 2018 ??? **UNCHANGED. The explanation is in the list of references.**

p 2 l 35 : build -> built **done**

p 2 l 35 : give name of the chinese space agency instead of country **Chinese Academy of Sciences**

p 3 l 15 : the acronym TCCON is not defined **done, (Total Carbon Column Observing Network)**

p 4 l 10 : carry -> record **done**

p 5 l 13 : phrase is redundant with figure caption **UNCHANGED. The caption of a figure must explain what is on the figure, because some quick readers just look at the figures and need to understand by only reading the captions. It is usual practice to have some redundancy between the main text and the captions.**

p 6 l 1 : Delete We have made use ... **done**

p 6 : Number appendices in order of appearance. Describe the structure of the paper first and Appendices later. **done**

p 7 l 11 : fundamental -> electronic ground **done**

p 7 l 37 : form -> from **done**

p 8 l 5 : what is an absorption emission spectrum ? **done: "absorption" is deleted.**

p 8 l 17 : as a part of the OSIRIS instead of a part of OSIRIS **done**

p 8 l 21 : spectral capability (resolution l/dl~850) -> resolving power (L/dl~850) of the **done**

p 9 l 2 : reducing cross-track -> reducing the cross-track **done**

p 10 l 11 : change cm3 -> cm2, s1 -> s **done partially. The emissivity is per unit volume; and $s^1$ was correct, but now changed to s.**

p 10 Fig. 4 The figure should go into Appendix. It is neither discussed nor does is seem to be particularly important to what is developed in the paper. **done; displaced to**

**Appendix F.**

p 12 l 11 : nearer 1 for larger -> closer to 1 for higher **done**

p 19 figure 11: There are no panel numbers on the Figure (a, b, c). please denote by top middle and bottom panel. Corect the text accordingly. **done**

p 22 figure 13: The figure does not comply with publication standards. **will be done if required by printer**

p 23 l 11 : Delete second part of title. This is explained in the text. **done**

p 24 l 14 : Delete phrase : The wavelength unit pm ... **done**

p 25 l 26++ : Use _a instead of _aero. Avoid using _aer and _aero interchangeably.

**Partially don. We have kept $z_{aer}$ and $w_{aer}$ to be consistent with notations of Butz et al 2009.**

p 27 figure 16 : panels are interchanged. top is high albedo, bottom is low albedo. **done**
Avoid cryptic notation that is not used in the publication (Lmax, Mmin)
**Additional explanation in the caption makes Lmax and Lmin no longer cryptic.**

p 27 l 12 : Tr is used as a symbol for transmission, whereas it is T later on (section 6.2.1) **Tr changed into $T_{atm}$**

p 27 l 16 : replace changing continuous by continuous **done**

p 28 l 24 : The header (LATMOS inversion breadboard) is not very informative about what is done in this section. It should be chosen differently **done: new header: Algorithm used in the LATMOS inversion breadboard**

p 29 l 21 : symbol TH2O is not explained **Done**

p 29 l 12 : please explain why water enters into the modeling. So far the article has been concerned with O2 and all simulations have been done without H2O. **Done, explanation added, see below.**

p 29 l 22 : intensity is not the same as an albedo. **done: (proportional to the albedo)**

p 29 l 20 : why does K5 appear as exponent in eq (11) ? **this is due to $T_{atm}(\tau)=exp(-\tau)$. Text added: $T_{H_2O}$ is the atmospheric transmission of water vapour for a reference column of $H_2O$, and is elevated to the power $K_5$ to compute the transmission of another column of water vapour, where $K_5$ is the ratio of $H_2O$ column / reference column.**

p 30 figure 17: Avoid SCIA as an abbreviation for SCIAMACHY. Note that you have already used CIA as abbreviation for collision induced absorption. You have used SCIAMACHY everywhere else. **done**

p 30 l 11 : intensity -> intensities **Done**

p 31 l 14,15 : g should be a subscript. **Done** Use same notation of singlet oxygen everywhere. **not done; greek letters are not allowed in many documents (this one, on the web site of AMT, in particular), and we found convenient to use the $O_2^*$ notation to designate this excited state of the molecule. We added the sentence (page 7, 1st paragraph:**
"**Throughout this paper, we use for convenience indifferently $O_2^*$ or $O_2(^1\Delta)$ to designate either the molecule in its excited electronic state ($a^1\Delta_g$), or the light emission when the molecule de-excites spontaneously. The context allows understanding what is meant when it is used.** "

p 31 l 12: values haigher -> higher values **Done**

p 33 figure 20: One can hardly distinguish dark violet points (morning) from black points (Reprobus) **done , changed to blue for better colour contrast.**

p 34 l 19 : so far, you have been concerned with the determination of the O2 column.

Psurf shuld only be introduced as an auxiliary, not as the main parameter. **done**

p 35 l 8 : similarly use O2 column instead of psurf. **done**

p 35 l 18: This airglow is mainly due to photolysis of ozone in the mesosphere, letting an O2 molecule in an excited state O2* which spontaneously de-excite. -> This airglow is mainly due to the spontaneous relaxation of excited oxygen (O2 Delta) that is formed in the photolysis of ozone. **done**

p 35 l 23: absolute amount seems to be in contradiction to contamination. replace "absolute amount of ... contamination with "airglow radiance". **done**

p 36 l 2 : In summary, we have found that the intensity of the O2* airglow is well organized. Probably you mean something like well behaved. The word organized is very confusing here. **done**

p 36 l 5 : what is a meteo field ? meteorological or meteorology field ? done; replaced by **meteorological .**

p 36 l 7+: This is a new result and should not suddenly appear in the conclusion section without having been presented as a result before. **Answer: Agreed. A new sub-section 6.3 was added, summarizing the results of Appendix D (new numbering after deletion of Appendix A).**

p 36 l 38: It is not clear which simulations you refer to : "This allowed performing some simulation exercises with good confidence about their ability to represent reality, showing that with the resolving power of 25,000 of the MicroCarb instrument, it is indeed possible to disentangle the airglow emission from the O2 absorption in the O2 band at 1.27 μm." **done: added: (see below section 6.3)**

p 37 l 4 : These are new results that have not been discussed in the previous part of the document, but rather in Appendix E. It seems to be necessary to include parts of Appendix E in the main section. **Answer: Agreed. A new sub-section 6.3 was added, summarizing the results of Appendix D (new numbering after deletion of Appendix A).**

Appendix : Check references to equations in text. There are plain numbers in the text whereas equations are numerated by the Appendix Number + Equation Number. **done**

p 65 l 12 : ratio emission/absorption -> emission/absorption ratio

**done...emission/absorption strength ratio.**

p 65 l 23 : Q_tot^up is not defined. **It is defined in line 3 of page 60.**

Submitted on 30 Nov 2019
Anonymous Referee #1

**Anonymous during peer-review: Yes** No
**Anonymous in acknowledgements of published article: Yes** No

**Recommendation to the editor**

| | |
|---|---|
| **1) Scientific significance**
Does the manuscript represent a substantial contribution to scientific progress within the scope of this journal (substantial new concepts, ideas, methods, or data)? | **Excellent** Good Fair Poor |
| **2) Scientific quality**
Are the scientific approaches and applied methods valid? Are the results discussed in an appropriate and balanced way (consideration of related work, including appropriate references)? Note that papers do not necessarily need to be long to be scientifically sound. | Excellent **Good** Fair Poor |
| **3) Presentation quality**
Are the scientific results and conclusions presented in a clear, concise, and well structured way (number and quality of figures/tables, appropriate use of English language)? | Excellent **Good** Fair Poor |

For final publication, the manuscript should be

**accepted as is**

**accepted subject to technical corrections**

accepted subject to **minor revisions**

reconsidered after **major revisions**

    I am willing to review the revised paper.

    I am **not** willing to review the revised paper.

**rejected**

**Suggestions for revision or reasons for rejection (will be published if the paper is accepted for final publication)**

Some extra modification might be necessary to better distinguish the main text and the appendixes. Specific comments:
Title: consider replacing "GHG" with "greenhouse gases". **Done**
Page 1, lines 32 and 38: these two "fully" may not be necessary. "Validated" and "confirmed" are assertive enough. **done**
Page 2, line 5: please quantify "a great accuracy". **It is quantified in the following sentence**

Page 2, lines 9-10: consider changing "very similar (likely identical) to" to "consistent with". **done**

Page 3, lines 1-2: not only temperature profiles are needed as ancillary information. Suggest "atmospheric profiles". **done**

Page 3, line 3: "molecule" to "molecules". **done**

Pages 3-4: the last two paragraphs in page 3 seem largely redundant, and many points are repeated in page 4. The authors are encouraged to reorganize and consolidate these background-introductory paragraphs. **Corrected**

Page 6, first paragraph: it appear really disorganized in current form. Try simplify it: section 2 – airglow spectral model; section 3 – VER retrieval; section 4 – airglow VER vs. REPROBUS model; section 5 – application to MicroCarb; section 6 – airglow retrieval from nadir spectra. **answer: rewritten along the suggested lines for more clarity.**

Page 6, title of section 2: may remove "(0,0)" as airglow is also in (1,1) band. answer :done

Page 7, lines 2-3: the sentence starting with "once it is produced" may read better if moved downwards to after "… of 30 MegaRayleigh." answer: change done.

Pages 7-8, section 2.3: the terms of equation 4 should be clearly defined in the main text, so the readers can understand the context without going to the appendix. answer :done

Section 3: please make the notation of excited O2 molecules consistent (O2*, O2(1Delta), or O2(a1Delta)). **Answer. We prefer to keep $O_2^*$, $O_2$ (1Delta) indifferently to designate the emission process, and $O_2$ (a1Delta g)) to designate the particular electronic state. One reason is that it is impossible to put a greek letter in some documents, like the present document, while $O_2^*$ is OK. We added the sentence (page 7, 1st paragraph:** "**Throughout this paper, we use for convenience indifferently $O_2^*$ or $O_2(^1\Delta)$ to designate either the molecule in its excited electronic state ($a^1\Delta_g$), or the light emission when the molecule de-excites spontaneously. The context allows understanding what is meant when it is used.** "

Page 8, lines 16-22: this paragraph seems redundant with section 2.2. **done**

Page 9, line 22: please provide information on how "atmospheric models" were used in the VER retrieval, and why different model profiles gave different VER inversion. **answer: done with inclusion of : (the density profile modifies the re-absorption by $O_2$).**

Page 9, line 27: please explain "adapted climatology" in the main text. answer: done by adding explanation …"adapted climatology" (for which we take for each measurement the most appropriate in latitude and season of the three considered atmospheric models),

Page 11, Figure 5a: I'm not sure how there could be an offset between the reconstructed airglow spectrum and the SCIAMACHY limb spectrum, with non-zero mean, because the VER profiles were retrieved from SCIAMACHY. answer: we assume that you are refering to Figure 6b (now 5b after deletion of one figure). The main point of the figures 5a and 5b is to show how better is the fit to SCIAMACHY spectra of the "new" model w.r.t. the "crude" model, by dividing SCIAMACHY spectra by a forward model. On figure 6b, we see that the "new" model is slightly below the data, with an offset (from 1) which is decreasing with increasing altitude. This trend is assigned to the fact that, in this simple particular model computation, the re-absorption by $O_2$ is not accounted for. The fact that there is still some offset (a few %) at high altitudes (80-90 km) may be interpreted as the fact that even at this altidue range, the O2 absorption may be still significant at this level. However, it might be assigned also to some small numerical shortcomings in the vertical inversion/computation of the synthetic spectrum/integration along the LOS, in particular when dealing with the uppermost layer of an onion-peeling inversion scheme. We are not

going to investigate this point in details, out of the scope of the present paper which is already very long.

Still, we are thanking you for the suggestion, and such a verification could be implemented with future limb-viewing of MicroCarb. Also, the vertical inversion could be done separetely for each spectral line for a better representation of reality.

Page 12, line 9: given the aforementioned concern, I'm not sure this validates the airglow spectral model "completely". The residuals from the right panel Figure 6 are not perfect; something systematic is still there. answer: **"completely"** is deleted

Page 14, lines 15-16: instead of deferring it, why not briefly summarizing the linkage here? **answer: end of last paragraph modifed as suggested.**

Page 16, line 11: just "except over the oceans…". done

Figure 11: I don't see the subsolar point plotted at the centre of the figure. In fact, the date plotted is the northern solstice, but the plot appears to be showing the southern solstice, if the subsolar point is at the centre. done with a red spot on the bottom panel (same position on all panels).

Page 22, lines 2-11: this paragraph may read better if moved to the end of this subsection, after Figure 13/14. **answer: there is some new re-arrangement of the text of section 4.2.2 as suggested .**

Page 22, lines 10-11: please briefly summarize how ozone distribution may contribute the discrepencies. answer: **added: "One obvious possibility is that there would be actually more ozone in the upper stratosphere than predicted by the REPROBUS model, since the $O_2$\* emissvity is proportionnal to the ozone concentration at high altitudes (optically thin medium).**

Figure 13 and 14 show similar things. Consider removing one.answer: we prefer to keep fig.13 and 14? Fig.13 is showing three examples.  Fig 14 contains mainy examples overplotted.

Page 26, lines 5-9: this paragraph looks introductory. Similar information already appears in the beginning. **answer: slightly rephrased for consistency.**

The OCO-2 bias correction is a new point, but it might be contradictory, as we don't know if OCO-2 had adopted the 1.27 micron band, what would happen to its biases. **answer: if OCO-2 had adopted the 1.27 micron band, we suspect that the biases would be smaller, because the aerosols spectral signature would be better represented.**

Section 6 title: it is not "a synthetic spectrum". Suggest "Disentangling O2\* airglow from O2 absorption in nadir viewing spectra" **answer: change done**

Figure 16: please provide the source of CIA cross section. Is that also HITRAN 2016? answer :yes, the $O_2$ CIA is now in HITRAN.
added: now included in HITRAN 2016 (Gordon et al. 2017)

Figure 20: it would be helpful to emphasize here that this day/night hysteresis is not observable from limb due to sampling sparsity. Only nadir ocean data can provide such spatial sampling density. AH?

Page 35, line 27: change "contaminated" to something like "influenced by atmospheric scattering" as "contaminated" is used for airglow two lines above.
Answer: "Contaminated" has been deleted two lines above at the request of the editor.
we added: … not contaminated by atmosphere/aerosols scattering of solar radiation..

Page 37, lines 16-18: it is difficult to understand where the 1.4 ppm XCO2 error comes from. Please clarify.

Answer: **This 1.4 ppm is simply the error on the O2 column= 0.35%, x 410 ppm of CO2.**
(0.35%x410 ppm) added.

Answer: **This 1.4 ppm is simply the error on the O2 column= 0.35%, x 410 ppm of CO2.**
(0.35%x410 ppm) added.

**Report #2**

Submitted on 02 Dec 2019
Referee #4: David G. Johnson, david.g.johnson@nasa.gov

**Anonymous during peer-review:** Yes **No**

**Anonymous in acknowledgements of published article:** Yes **No**

**Recommendation to the editor**

| | |
|---|---|
| **1) Scientific significance**
Does the manuscript represent a substantial contribution to scientific progress within the scope of this journal (substantial new concepts, ideas, methods, or data)? | **Excellent** Good Fair Poor |
| **2) Scientific quality**
Are the scientific approaches and applied methods valid? Are the results discussed in an appropriate and balanced way (consideration of related work, including appropriate references)? Note that papers do not necessarily need to be long to be scientifically sound. | **Excellent** Good Fair Poor |
| **3) Presentation quality**
Are the scientific results and conclusions presented in a clear, concise, and well structured way (number and quality of figures/tables, appropriate use of English language)? | Excellent **Good** Fair Poor |

For final publication, the manuscript should be

**accepted as is**

**accepted subject to technical corrections**

accepted subject to **minor revisions**

reconsidered after **major revisions**

    I am willing to review the revised paper.

    I am **not** willing to review the revised paper.

**rejected**

**Suggestions for revision or reasons for rejection (will be published if the paper is accepted for final publication)**

General Comments:
I did find that the paper was easier to read after the latest revision, although I noted a number of references that were not updated after the material was reorganized. The ones that I caught are noted below. I was also surprised that after moving some material to appendices the total page count increased from 75 to 90, although this was perhaps a result of inefficient formatting. I noticed that the appendices seemed to have a lot of half pages.

I do not understand why the old section 6.3 (Surface pressure retrieval on simulated nadir

spectra contaminated by O2* airglow) was moved to Appendix E. Despite the disarming title, this seems to me to be one of the most important sections of the paper. This section presents the simulated MicroCarb retrievals that are quoted in both the abstract and the conclusion, and by moving it to the appendix the body of the paper contains no detailed description of where these important results come from. **Answer: Agreed. A new short sub-section 6.3 was added, summarizing the results of Appendix D (new numbering after deletion of Appendix A).**

Specific Comments:
Page 27, Figure 16 caption: I think that the description for the top panel matches the bottom plot and the description for bottom panel matches the top plot. **Correction done**
Page 28, line 12: Refers to subsection 6.3. 6.3 has been moved to Appendix E. So either change reference to Appendix E (**done**), or move Appendix E back to subsection 6.3 (preferred?)
**Answer:  A new short sub-section 6.3 was added, summarizing the results of Appendix D (new numbering after deletion of Appendix A).**

Page 37, first paragraph of section 7.3: Shouldn't the authors at least quote the result - 0.11hPa/0.88hPa vs the requirement of 0.1hPa/1.0hPa from Appendix E, Table 2? Isn't this result the key point of the paper? **Yes, done. A new short sub-section 6.3 was added, summarizing the results of Appendix D.**

Technical Corrections:
Page 1, line 17: "derived form" should be "derived from" **done**
Page 2, line 20: "fuelled" should be "fueled" **done**
line 34: "lost at launched" should be "lost at launch" **done**
Page 4, line 26: "smoothness is comforted" should maybe be "smoothness is confirmed"? **done**
page 7, line 37: "using some equations form" should be "using some equations from" **done**
page 9, line 16: "account ofr" should be "account for" **done**
page 56, line 25: "quadrupo:" should be "quadrupole" **done**
page 57, line 11: "form the ground" should be "from the ground" **done**
page 59, line 8: "equation (6)" should be "equation (B3)"? **done**
page 62, line 13: "rovibrationnal" should be "rovibrational" **done**
page 65, line 18: "equation (14) and put it in equation (13)" should be "equation (B11) and put it in equation (B10)"? **done**
page 66, caption for Figure B5: the font size seems to change after "217 K,"? **done**
page 71, line 8: Should reference to Section 2 be changed to a reference to Appendix B? **done**
page 72, line 5: same question about the reference to Section 2. **done**

**Report #3**

Submitted on 13 Jan 2020
Anonymous Referee #5

**Anonymous during peer-review: Yes** No
**Anonymous in acknowledgements of published article: Yes** No

**Recommendation to the editor**

| | |
|---|---|
| **1) Scientific significance**
Does the manuscript represent a substantial contribution to scientific progress within the scope of this journal (substantial new concepts, ideas, methods, or data)? | Excellent **Good** Fair Poor |
| **2) Scientific quality**
Are the scientific approaches and applied methods valid? Are the results discussed in an appropriate and balanced way (consideration of related work, including appropriate references)? Note that papers do not necessarily need to be long to be scientifically sound. | **Excellent** Good Fair Poor |
| **3) Presentation quality**
Are the scientific results and conclusions presented in a clear, concise, and well structured way (number and quality of figures/tables, appropriate use of English language)? | Excellent Good Fair **Poor** |

For final publication, the manuscript should be

**accepted as is**

accepted subject to **technical corrections**

accepted subject to **minor revisions**

**reconsidered after major revisions**

    **I am willing to review the revised paper.**

    I am **not** willing to review the revised paper.

**rejected**

**nswer**

The manuscript "The use of O2 1.27 micron absorption band revisited for GHG monitoring from space and application to MicroCarb" is a detailed and thorough inspection of the oxygen airglow phenomenon as observed by SCIAMACHY on ENVISAT, however, it does not adequately project these results into the purported conclusions regarding the architectural choices for the MicroCarb instrument.

The title includes "...application to MicroCarb", the abstract and introduction discuss uncertainty in retrievals between A-band and delta band with respect to confounding aerosol signatures as well as atmospheric emission. The question posed (somewhat

indirectly) by the authors is "Should GHG instrumentation use the 1-Delta band for airmass determination?". The answer to the question is presented as a competition between errors induced by atmospheric emission vs. errors induced by aerosol extinction (and extrapolation of that effect). This is a reasonable premise for a manuscript, however it ignores other factors such as (1) the relative signal to noise ratio (SNR) of practical instrumentation (2) the errors due to instrument trades in spectral resolution (3) biases induced by spectroscopic errors. Some of (2) is mentioned in this manuscript, but it is not quantitative. The cursory mentions of SNR (1) are made for existing instruments and not put in context with MicroCarb SNR, which is not described. As for (3), the authors are utilizing HITRAN, which is a world-class standard, but is NOT what is used for OCO and GOSAT missions, (ABSCO), nor for TCCON (GFIT), which find that retrievals approaching 1 ppm accuracy require (among other things) a dedicated spectroscopic formulation that goes beyond HITRAN.

**Answer: HITRAN is used as a first cut for simulations. The Geisa data base will be used also. The problem of GHG is sufficiently important for the whole humanity that any mission needs to capitalize on previous work, and make publicly available new techniques and methods. This is the spirit of the present paper. And the MicroCarb analyzing team will make use of the best available spectroscopic data, ABSCO, GFIT, or other, including the speed effect etc…**
**The MicroCarb instrument is described in Section 5; the SNR is indicated for each channel, and it was used in the simulations with 4ARCTIC software, as described in Appendix D and summarized in (new) section 6.3.**

Before providing some detailed comments, I will make a primary recommendation - re-write this manuscript as two independent manuscripts. The first manuscript should focus on the airglow retrievals and model intercomparisons of the SCIAMACHY data. The second manuscript should build on the airglow model validation of the first paper and then present a fully quantitative intercomparison of the methods for airmass retrieval from the differing O2 near-infrared bands. This first manuscript is essentially complete and represents a subset of the information currently reported. The second manuscript, if meant to answer the question about the use of 1-Delta band for MicroCarb, will need to address much of the detailed comments below. If the authors choose to retain a single manuscript, my 'presentation quality' rating of 'poor' will be difficult to overcome because the effort as presented here is imbalanced and does not provide a cohesive conclusion.

**Answer: in the present manuscript, we do not pretend to have completely demonstrated that the use of the O2 IR band, by improving the representation of the effect of aerosols in the CO2 bands, will improve the retrieval of CO2. Such a demonstration can be done only with actual data, and this is the role of MicroCarb to bring for the first time such data. We are just showing that the airglow spectrum may be disentangled from the solar back scattered radiation with sufficient accuracy to be useful for GHG retrieval.**
**Therefore, your rating of "poor" is not relevant to the aim of our paper. The three other referees have not used such a rating.**

****detailed comments*****
major:

A forward model of aerosol effects is formulated in Eqs 5-7, but not used to make any quantitative statements. For application to GeoCarb, I expected the relative contributions

of aerosol loading in the 0.76 and 1.27 micron regions to be presented. I also am concerned that the retrieval of aerosols will depend critically on the bandwidth of the instrumentation, as alluded to in the dicussion about reducing GeoCarb's resolution to cover the whole 1.27 micron band. These factors surrounding aerosol effects must be quantified and then weighed vs. other sources of error such as the airglow uncertainty.

**Answer: We are not considering GeoCarb in the present paper, but only MicroCarb and possibly CO2-M. And the effect of the aerosols have not been quantified by simulations: this effort is in progress but well beyond the scope of this paper which already very long.**

The airglow model is presented to be independent of meteorological and chemical perturbations, which is a solid argument for its use in the simple subtractive approach. The authors dismiss issues at high solar zenith angle (SZA) in two different ways (1) the simplistic time fields do not capture the rapidly changing conditions near the terminator (2) the night-time kinetics are not important during daytime measurements. I find two issues with this approach. First, there is still $CO_2$ to be measured at high SZA (i.e. latitude) - so what is the practical cut-off? Second, the rapidly changing area of the SZA dependence is your best chance to observe/quantify what will otherwise be a bias. Another way to frame these points is that a detailed model of the high SZA dependence would allow for airmass retrievals to extend to high SZA/latitude where airglow bias would shrink considerably, and perhaps reveal other contributing biases. Since the presented airglow model assumes steady state and large time zones, this region will be dominated by model biases - at the very least, the MicroCarb mission should understand the extent of where these biases will effect retrievals.

**Answer: your suggestions for the analysis of the biases as a function of SZA are very welcome. And you are right about high latitudes being observed with large SZA. Also, MicroCarb will have the capability to observe at the limb the "pure" airglow emission, and MicroCarb science team will be able to characterize the detailed shape of the spectrum in limb viewing. One sentence was added in Section 5 describing MicroCarb: "Observations at the limb are foreseen, dedicated to record the pure airglow emission for spectral characterization and better simulation in the forward model that will be used for the retrieval of $O_2$ column in nadir viewing. »**

The proposed configuration of MicroCarb is presented as retrieving two bands of oxygen 'for safety'. Does this presume that the A-band retrieval is sufficient? Wouldn't a two-band retrieval (which is implied to be a good way to retrieve aerosol) be best? It is unclear what 'for safety' means, some kind of reduced risk presumably, but the manuscript provides no quantitative comparisons of A-band vs. delta-band vs. combined dual band retrievals.

**Answer: Clearly, we think that A-band retrieval is not sufficient, based on previous space missions. There are hints that areosols may be better coped with using the 1.27 µm band, and Sun et al 2018 are on the same line. A recently new CH4 mission in the US (MethaneSat) is also going to use the O2 IR band.**
**It should be recognized however that it is only with actual data that anybody could prove that there is a real improvement for CO2 (and other GHG) retrieval, using the O2 IR band. The launch of Microcarb will allow testing several retrieval methods, A band only, B band only, A+B bands… We have not even yet completed the simulations with aerosols, well beyond the scope of this paper. All future space missions may benefit from the exploration of the use of O2 IR band.**
**We have deleted "safety" and put … "for reference and comparison with previous**

**space missions. »**

The entire Appendix A is review material and is un-necessary for supporting the publication. **answer: This appendix is deleted.**

minor:

in the absract third paragraph (why are there 3 paragraphs!) the text implies CO2 can be retrieved with 'only' the 1.27 micron band measurement/ sentence modified :
**with the $O_2$ 1.27 μm band only as the air proxy (without the A band).**

in the first sentence of the introduction the main driver of climate change is attributed to CO2. I recommend qualifying this as 'human-induced climate change' since geologic climate shifts are driven by solar and orbital effects as well as GHG loading from geological sources. **agreed, done.**

page 3, line 15, "...modelling may be more accurate." this was stated as fact in the abstract. As a central theme of this paper, it seems to be inconsistently referred to in many ways. changed to « **modelling is more accurate »**

page 8 line 5, "...synthetic absorption emission spectrum." this is an odd thing to state **done : the word « absorption » is deleted** .

FIgure 6. If I understand the context, the emission=absorption case is what is described in the text and used in the model, but this figure implies a massive error, I think this is just my confusion, but it shouldn't so be hard to follow. Either way, this is a fundamental effect that seems trivial to spend a section explaining. **Sorry for the confusion. A new paragraph was written at the end of Section 2, explaining the difference between our « crude model » (used in an early phase of the project) and the new model based on equation (4).**

Page 16 line 9-12, "This is why it is not practical..." seems like a death-sentence for MicroCarb, I understand the spectral resolution is better, but the listed issue "back-scattered solar radiation" is not quantified for MicroCarb (vs. SCIAMACHY) in this document **The "back-scattered solar radiation" is the main signal on which is imprinted the absorptions of CO2 and O2. MicroCarb and SCIAMACHY both see the same addition of the airglow spectrum and the back-scattered solar radiation. We have shown in the present paper that, with the resolving power 850 of SCIAMACHY, we can estimate the airglow only over the oceans (with not enough accuracy to derive the mixing ratio of CO2), but we have shown in Appendix D (new numbering) that with the 25,000 resolving power of MicroCarb, we may achieve the dsired accuracy for $X_{CO2}$.**

Figure 10, which is reproduced from another source, is totally un-necessary for readers to understand the physics. **answer : Not all scientists interested in GHG issues are knowing very well the spectroscopy of the O2 molecule, and this figure summarizes visually what is included in the chemistry model computing the number density of the**

**O2\* molecule.We wish to keep this figure.**

Page 18 lines 19-20 "At a given solar zenith angle..." this point is central to the thesis of the effort, it should be mentioned in the abstract and the conclusion. **answer: In principle, it is not that central, since our simulations have shown that we can disentangle the airglow intensity for each nadir observed spectrum. However, if the nadir intensity varies smoothly with SZA, it will most likely ease the retrieval. Therefore, we have added a sentence in the abstract and in the conclusion. Note that we tried to estimate also the observed degree of variability of the airglow intensity from SCIAMACHY data which could be due to gravity waves, not simulated in the REPROBUS model (end of sub-section 6.2.2).**

page 22 lines 4-5, the radiometric properties of SCIAMACHY surely deserve a literature reference. **Answer : references must be relevant to the particular version of the SCIAMACHY data that we treated. We found and added one reference by Morstad et al. 2012.**

Figure 13. The descrepancies between measurement and model are indicated to be minor throughout the discussion, however, this plot suggests significant (maybe 30%) of the airglow (at higher altitudes) is unmodeled. Perhaps this error is not propogated into MicroCarb results, but no proof has been given. **Answer : you are right about your estimate, looking at figure 14 bottom. We have written the discrepancy numbers in the text below figure 14. This discrepancy is probably a shortcoming of the present version of Reprobus. It has no real impact on MicroCarb simulations, for which vertical distributions coming either from SCIAMACHY inversions or REPROBUS modeling were done. Some new efforts to improve REPROBUS are in progress now.**

FIgure 15. In the text the 1-Delta band is stated to be in the spectral and brightness range of the CO2 bands, but these plots seem to indicate that only the A-band has dynamic range that encompasses both CO2 bands, with the 1-Delta band falling in-between, is this a quirk of the selected simulation? My belief is that a 1-Delta only retrieval, especially with the limited bandwidth of MicroCarb, would not effectively span the radiative dynamic range (contrast) that provides high precision. **Answer : to be franck, at this stage we think that nobody completely understand the problems associated with retrievals with only the A band, though we suspect strongly the wavelength dependency of aerosols (general case up to now). MicroCarb will give the opportunity, for the first time, to compare retrievals either with the A band only, or with the 1.27 μm only, or with a combination of both bands.**

Page 29 line 25-26 "...uncertainty is assumed to be proportional to square root of signal." Such a weigthing scheme reduces the impact of a large dynamic range, it also seems unjustified unless the noise in the instrument is dominated by its source (the sun), usually remote sensing instruments are dominated by detection noise which is spectrally flat. **Answer : We disagree with this last statement. For remote sensing instrument measuring the scattered solar radiation, the main source of noise is photon noise, much more than the detector readout noise. Therefore, the choice of the error proportionnal to the square root of the signal is justified (Poisson law) in the simulations made with the LATMOS breadboard. In the simulations made with 4ARCTIC described in Appendix D, they are made with the instrument curve of the**

**noise vs signal, as given by engineers.**

Page 29 line 38 following on the last comment, the uncertainty may not be calculated from the SNR, but it IS used to weight the fit, which does something, also the "dispersion of the results" could be either instrument noise, model error, or retrieval failure, some description of the relative impacts of these factors is needed to understand the value given:
**answer : we found out on figure 17 that the dispersion of the retrieved airglow intensity results is high (large error bar on retrieved value, which can be qualified as a failure of the retrieval) , except when the global intensity is very low (low albedo,high SZA).  We have rephrased accordingly:**
**"The dispersion depends very much on the intensity of the backscattered solar flux as shown later on Figure 17."**

Page 35 line 14-17, "It is widely recognized..." here the main question is answered as being a known, however, this manuscript did not convince me and no reference is given.
**Answer : we fully agree that the full demonstration , by simulation, that this O2 IR band is better than the A band in the case of aerosols is not present in this paper. This demonstration is well beyond the scope of this paper, as it requires heavy efforts of simulation with 4ARCTIC or other software. We note that Kuang et al (2002) « recognized the virtues of the [IR] O2 band » as we write page 4.**

**We have re-written : "It is widely recognized..., it should be in principle…"**

Page 36 lines 9-12, Here are some quantitative statements, presumably the nadir GHG observation that is imagined here is that of MicroCarb? This list of quantitative results is in the conclusion, and the necessary retrieval method implied by them is described in the following section of the conclusion. This is a very odd structure for a manuscript.
**Answer : Agreed. This was the infortunate result of some re-arrangment of the paper done at the request of the editor. Now, a Section 6.3 has been put back in the main tetx, summarizing the retrieval exercises detailed in Appendix  D (before, Appendix E).**

Page 37 line 9-10 Here we are in the conclusions still, and another result is provided fro the first time, albeit very qualitatively "...accuracy almost compliant...". The number to comply with was never stated, the model result not given and the superiority over A-band results is simply stated as fact.
**Answer : The new Section 6.3 is written explaining these numbers.**

Page 37 lines 20-25 This imples MicroCarb could/should do better with a resolution/bandwidth trade, which contradicts much of the premise of the rest of the mansucript that seems to confirm MicroCarbs configuration **Answer : this paper is not only about MicroCarb, but more generally about the use of the $O_2$ 1.27 µm band. At the beginning of MicroCarb, engineering studies showed that the knowledge of the ILSF (Instrument Line Spectral Function) was essential for a good rendering of the CO2 column, and dictated the 25,000 resolving power. At a later stage, and as a result of our analysis (presented int his paper), the MicroCarb project decided to include the 1.27 µm in the final design, at the expense of dropping a $CH_4$ channel. The reason for this excruciating choice was to improve as much as possible the CO2 measurements quality, because the main objective of MicroCarb is to quantify the**

**natural $CO_2$ fluxes as accurately as possible.**

Page 37 lines 25-29 Here, at the end, another potential instrument is described, something called CO2-M, and a recommendation is given about including the 1.27 micron band. Presumably this is a follow on possibility? If anything, the effort here frames a capability to model and predict good instrument design choices, but what is given here is hardly more that a statement about necessary resolving power, and the study claims to provide justification for the entire band. When it comes to instrument design trades, resolving power is only one of many factors. **Answer : the CO2-M project (a series of satellites to monitor CO2) is decided and financed by European Community and operated by ESA for UC. Will the design of $CO_2$-M be frozen for the next 30 years, without the possibility to include the $O_2$ 1.27 μm band ? even if MicroCarb shows that it is a good choice ?**

Appendix B page 59 line 9, reference to equation 6 is wrong, equation 6 is about aerosols **Answered : corrected , now A3. (Appendix A is deleted).**

Appendix B page 60, the discussion about partition sums references HITRAN quite a bit, but the partition sums in HITRAN are literature traceable to Gamache and the TIPS code/tables, the references are given in HITRAN. The 'care' needed to extract energy levels from HITRAN should be alleviated with HITRANonline, which now book-keeps energy levels directly, the energy levels for oxygen are traceable to Yu et al. If one needs to calculate upper state energies, the transition energy is added to the lower state energy. **Answer : This is what we have done indeed. We have used HITRANonline, used the lower energy levels which are in these tables, and added the energy of the transition to get the energy of the upper level.**

Figure B2. THe labeling must not be correct, 'statistical weight' would be a straight line vs. J', what is plotted is probably population of the level. This figure is completely basic and un-necessary. **Answer: you are right. Labeling has been corrected. Of course, for a spectroscopist, there is nothing new in this figure. But not all scientists concerned with climate change and GHG are good spectroscopists, and we prefer to keep this figure here.**

Figure B5. The axes label and caption seem to imply a ratio is plotted, however the values and units are not consistent with a ratio. **Answer : this is a ratio of two quantities atached to one molecule that have not the same dimensions, and therefore must have a unit.  The emission rate is in photons per molecule s$^{-1}$, and the line strength in Hitran tables is given for one molecule in cm$^2$ cm$^{-1}$. Therefore, the unit of the ratio should read : photons cm$^{-2}$ s$^{-1}$/ cm$^{-1}$, being understood that cm$^{-1}$ is a wavenumber unit, This is what is done on the figure.
The cm$^{-1}$, wavenumber unit,  should certainly not be confused with a length unit cm or area unit cm$^2$.**
page 67 line 24, what is E_mn(lambda)? **Answer : Emn(lambda) it is the same as Em(lambda), after normalization to the VER, as explained in the text.**

Page 86 lines 20-23 I find it surprising that the observations "...suggest a fast increase of the emission with decreasing altitude around 50 km..." the chart in Fig 13 shows a decrease in emission below 50 km. **Answer : here we are talking of the emission in the A band**

**around 0.76 µm, while Figure 13 is for the 1.27 µm band. The fast increase with decreasing altitudes may be found on figure 5 of Zarboo et al 2018, left, as indicated in the text.**

[revised manuscript text omitted]

---

## Author Response (AR4)

**Author's response to the Associate Editor Decision (1st May, 2020).**

All requested changes have been made and included in word revision mode to remain visible in the following revised version in pdf. Separately, the corrections have been accepted in the revision mode to produce a revised version of the manuscript.

**Associate Editor Decision: Publish subject to technical corrections** (26 Apr 2020) by Christof Janssen
Comments to the Author:

Dear Authors,

Thank you very much for completely addressing all issues that have been brought up in the previous round of refereeing. All referees agree that your paper is original and should be published in Atmospheric Measurement Techniques and I congratulate you for this accomplishment.

However, in rereading the latest version of your article, I still have found some technical issues that must imperatively be addressed before the article can go into the publication phase.

Please correct these :

p 1 Lines 6 - 12. It seems that addresses 1 and 6 are effectively the same. If so, please remove affiliation 6.

p 7 l 6&7. Please delete ", or the light emission when the molecule deexcites spontaneously." The phrase is not compatible with common rules. In the chemical literature (ACS style book (p. 262)) the star designates an electronically excited state and not the process of de-excitation involving such a state. Note that both EGU and AGU embrace recommendations on notation from the chemical societies, and from ACS in particular. Note also that you mostly write O2* emission or O2* airglow instead of simply O2* when you refer to the process of photon emission via de-excitation of O2*.

p 7. The symbol for the rayleigh is R. If used with SI prefixes, megarayleigh should be MR and kilorayleigh kR.

p 8 l 12 : Replace "joule "by "J"

p 9 l 1,3 : remove "," in the wavelength specification. The use of a thousand separator is inconsistent with usage elsewhere. It is thus confusing.

p 9 l 24 : Please remove square brackets to designate wavelength domains.

p 16 - 21 : 3 or 4 different referees have all criticised the confusing use of different notations for O2 in its first electronic excited state (O2*, O2($\Delta$), O2(1$\Delta$), O2(a1$\Delta$), O2(a1$\Delta$g)). While the use of these different notations is not incorrect in the first place, it is a possible source of confusion for the general reader if different notations are used in an arbitrary fashion. Since AMT imposes the consistent use of symbols/notation, I request to replace all instances of O2(a^1$\Delta$_g)) on pages 15 - 21 by O2*, except for appearances in equations, Figures or on p 16 l 35 (which refers to Fig 9). In addition, O2(a1$\Delta$) should be replaced throughout the whole manuscript by O2(1$\Delta$).

This would remove one notation and bring the text consistent with your explanatory remark on the use of notation on the top of p 7, where O2(a1Δ) is not listed.

p 21 Fig 12: The curve labelling outside the legend box is not readable. Since figures are already given in the legend box, the labels should be suppressed altogether. Also change the x-scale so that SCIAMACHY data for SZA = 33.3° are fully shown. The color bar on the RHS is completely superficial given the much more accessible information provided by the legend box. Please remove.

p 27 l 15: "Igor Pro" is a commercial software. It should be cited using the correct name and you should possibly give the version number that you have used. Since it is a company who owns copyright of the product, its name should also be mentioned in the text or given in the reference list.

p 50 Fig A3: The wavelength label "1.27 µm" corresponds to a transition, not to an energy; please remove from graph.

p 74 l 5 : Remove "s" at end of line

[revised manuscript text omitted]

bertaux 1 mai. 20 17:14